# A self-adaptive hardware with resistive switching synapses for experience-based neurocomputing

S. Bianchi [1,5,6], I. Muñoz-Martin [1,5,6], E. Covi [1,2], A. Bricalli[3], G. Piccolboni[3], A. Regev[3], G. Molas [3], J. F. Nodin[4], F. Andrieu[4] & D. Ielmini [1] ✉

Neurobiological systems continually interact with the surrounding environment to refine their behaviour toward the best possible reward. Achieving such learning by experience is one of the main challenges of artificial intelligence, but currently it is hindered by the lack of hardware capable of plastic adaptation. Here, we propose a bio-inspired recurrent neural network, mastered by a digital system on chip with resistive-switching synaptic arrays of memory devices, which exploits homeostatic Hebbian learning for improved efficiency. All the results are discussed experimentally and theoretically, proposing a conceptual framework for benchmarking the main outcomes in terms of accuracy and resilience. To test the proposed architecture for reinforcement learning tasks, we study the autonomous exploration of continually evolving environments and verify the results for the Mars rover navigation. We also show that, compared to conventional deep learning techniques, our in-memory hardware has the potential to achieve a significant boost in speed and power-saving.

In the last decades, artificial intelligence (AI) has drawn inspiration from the biological world, where humans and animals interact with one another and the surrounding environment to improve the efficiency of routine tasks[1]. This continuous and mutual interplay enables a constant boost of the abilities, the knowledge, and the complexity of the organisms, which become increasingly resilient to the daily life[2]. Currently, achieving efficient adaptation to the continually evolving situations of life is a major objective of the AI community, whose principal aim is to build machines able to infer concepts and to make decisions[3].

The experience-based knowledge, where agents evolve by trial-and-error episodes throughout their entire life, is an interdisciplinary subject of biology, computer science and neuroscience known as "reinforcement learning"[4]. During the last decades there have been several studies to contextualize the framework of reinforcement learning. For instance, the Markov Decision Process introduces a numerical framework under the hypothesis that the state probability and the reinforcement learning operations are known and accessible[5,6]. Such decision-making procedure introduces a probability function $P(s, a, s')$ which weights the value $V(s)$ of a certain position "$s$" for moving toward another state "$s'$". In equation:

$$V(s) = \max_a(R(s, a) + \alpha \sum_{s'} P(s, a, s')V(s')),\qquad(1)$$

The solution of the Markov process is a policy method which defines, if the model of the environment is known, the most convenient action to take at every available state[6]. However, in biology, organisms do not often have a model of the environment a priori, and they have to handle their own policies relying on the current occurrences by direct interaction with the surroundings. In this context, the Q-learning theory is a model-free algorithm used to assess the quality

[1]Dipartimento di Elettronica, Informazione e Bioingegneria, Politecnico di Milano and IUNET, Milano 20133, Italy. [2]NaMLab gGmbH, Dresden, Germany. [3]Weebit Nano, Hod Hasharon, Israel. [4]Univ. Grenoble Alpes, CEA, Leti, F-38000, Grenoble, France. [5]Present address: Infineon Technologies, Villach, Austria. [6]These authors contributed equally: S. Bianchi, I. Muñoz-Martin. ✉e-mail: daniele.ielmini@polimi.it

of an action in a particular state[7]. In formula:

$$Q(s) = R(s, a) + \alpha \sum_{s'} P(s, a, s') \max_{a'} Q(s', a'), \qquad (2)$$

where $\max_{a'} Q(s', a')$ is the maximum of all the possible $V(s)$. Consequently, a quality of a certain position $Q(s)$ is dependent on the quality of the nearest states $s'$. The Q-values can also map the value of each position with respect to the environmental modulations in time, thus defining the so called "temporal difference (TD)" framework[8]:

$$TD_t(s, a) = \beta(Q_t(s, a) - Q_{t-1}(s, a)), \qquad (3)$$

where $\beta$ is the inverse of the learning rate of the current $Q$ value with respect to the previous $Q$ value ($Q_{t-1}(a, s)$). These models can map the behaviour of the agent developing a decision-based policy by exploiting the interaction with the environment and taking a decision whose effect, in turn, constitutes part of the experience of the agent[9–11]. All these intuitions have been demonstrated in "Dyna", where learning methods were used for managing the planning results and for developing a cause-consequence model of the agent's actions[12].

To study the spatial learning and memory, several experiments were carried out in the field of behavioural neuroscience, such as the water maze exploration[13,14]. In particular, the Morris Maze navigation has been investigated by neuroscientists to study the effect of cognitive diseases related to the spatial learning[15]. Such studies also modelled the physiological basis of reward-based behaviours using Hebbian learning and spiking neurons[16]. In this context, it has been observed that when a penalty/reward event occurs, humans and animals release in brain dopamine, a pleasure-related neurotransmitter which become the reinforcement variable for the elaboration of the experience[13].

All these findings have been sources of inspiration for building intelligent hardware computing elements. In particular, in the last years, recurrent synaptic connections have been addressed as key elements for reproducing reward-based decision-making demonstrators[17] using both CMOS-based platforms[18] and non-volatile memories[19]. CMOS technology is the most mature approach for the AI hardware design, highlighted by the results achieved by deep learning with AlphaGo[20,21]. However, the first hardware setup of AlphaGo required 1920 central processing units (CPUs) and 280 graphics processing units (GPUs), with a peak power of half a megawatt[22]. Such power requirement is far from what is observed in brain-computation for mainly two reasons: (i) the slow and energy-hungry training procedures of deep learning techniques, for instance the "backpropagation"[21]; (ii) the communication delay between the processing units and the dynamic random-access memory (DRAM), also known as "Von Neumann bottleneck", while biological computation happens in-situ, i.e. in the same place where the information is stored[23].

For this reason, memristors, such as resistive switching devices (RRAMs) and phase change memories (PCMs), appear interesting for emulating the stochastic neuro-plausible computing, thanks to the reduced area, 3D stacking capability in the backend-of-the-line, increased parallelism, and analogue storage[24–27]. A key advantage of networks based on these emerging devices is the fast computation exploited by vector-matrix multiplications which can intrinsically perform in-situ multiplication and summation via Ohm's and Kirchhoff's laws[28,29]. Memristor arrays have shown enhancements in speed and energy for both in-memory supervised learning[30–32] and unsupervised learning[33–36]. Furthermore, they are the best candidates for neurocomputing, boosting algorithms such as the spike-timing dependent plasticity (STDP)[35,37] and the homeostatic mechanisms to stabilize the divergent growth of the weights under pure Hebbian learning[38,39]. Such features offer key abilities for the implementation of resilient bio-inspired systems but, generally, are not as accurate as

standard deep learning approaches, which, on the other hand, lack plasticity. These dichotomies of artificial neural networks with respect to the biological word was summed up since the early years of investigation in AI with the sentence "stability-plasticity dilemma"[40].

In this work, we propose a neuromorphic hardware based on Silicon Oxide (SiO$_x$) RRAM devices able to join state-of-the-art accuracy and bio-inspired plasticity for autonomous and resilient navigation at low-power. The network relies on bio-inspired algorithms, such as STDP and plastic homeostasis, to adjust the parameters along a temporal sequence, as in recurrent neural networks (RNNs)[41]. The RRAM devices are used for both Hebbian learning processes (integration, fire, potentiation/depression of the synapses) and to map the recurrent internal state of each neuron. In particular, the multilevel capability of the devices is used to modulate the neuronal threshold, acting as homeostatic boundary of the firing activities[42]. To test the resilience of the hardware, a two-dimensional dynamic maze showing environmental changes in time is experimentally configured in a field-programmable-gate-array (FPGA), thus mimicking biology[16] and deep-learning software-based approaches[43–45]. The bio-inspired hardware described in this work is also tested for complex cases such as the Mars rover navigation, thus investigating the properties of the system in terms of scalability and reconfigurability. The network starts from stochastic trials, it progressively maps the configuration of the environment, it becomes a master of the problem trial after trial, and it finally finds the optimum path towards the objective. Furthermore, we benchmark our work with respect to deep learning techniques, finally demonstrating that our solution overcomes the standard approaches used for autonomous navigation. In this context, we also present a theoretical framework which highlights the main benefits of the RRAM-based in-situ computation such as the high efficiency, resilience, low power consumption and accuracy. In the Supplementary Discussion of this manuscript, we also provide a further appendix on the numerical modelling of the bio-inspired approach to reinforcement learning and a more technical insight about the experimental setup.

## Results
### RRAM synaptic devices
The network relies on the resistive switching memory, RRAM, which consists of two electrodes in TiN separated by a thin layer of Silicon Oxide (SiO$_x$), Fig. 1a. Set and reset processes of the RRAM cause an increase or decrease of the resistance of the device, respectively: a gap appears during reset, responsible for the resistance increase to the high resistive state (HRS), while a filamentary growth emerges during set, responsible for resistance decrease to a low resistive state (LRS)[46]. The application of bipolar voltage pulses to the Top Electrode (TE), as in Fig. 1b, causes the switching of the devices from LRS to HRS and vice versa. On the other hand, by applying a sufficiently low $V_{READ}$ signal (50-150 mV, i.e., smaller than $V_{SET}$ and $|V_{STOP}|$) it is possible to measure a read current that is proportional to the resistive state of the device. When the RRAM is used as synaptic element in a neural network, this current is referred to as "post-synaptic current". Note that the RRAM devices show a wide resistive window (more than one order of magnitude), Fig. 1c. Furthermore, by varying the compliance current $I_C$ acting on the gate of the selector, it is possible to program a wide set of low multi-resistive values, thus enabling the use of the RRAM devices for mapping the plastic behaviour of the synaptic elements, Fig. 1d–f[32]. A similar tendency is evident even modulating $V_{STOP}$, here highlighting a multilevel range of high resistive values, Fig. 1g[47]. As visible in Fig. 1e, f for the LRS, and in Fig. 1g for the HRS, the definition of a resistive weight is always affected by a statistical uncertainty. Such variation is one of the main problems in memory-based deep neural networks since the computation strongly relies on the precision of the synaptic weights[48–51]. On the other hand, biological organisms draw their capability from the inherent parallelism, stochasticity, and resilience of neuronal and synaptic computation. Introducing bio-inspired

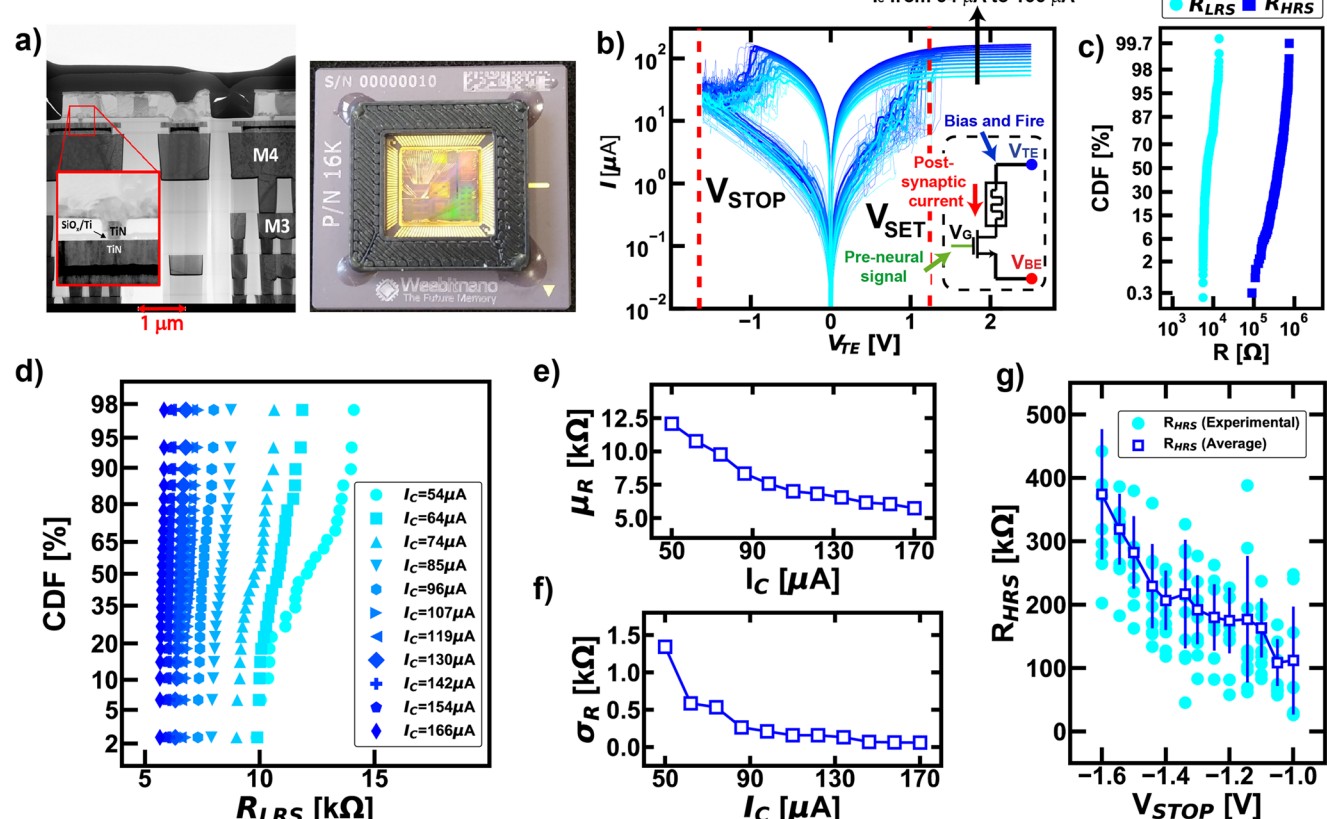

**Fig. 1 | Electrical characterization of the RRAM synaptic devices. a** Scanning Electron Microscope image of the SiO$_x$ RRAM devices and sample photo of the packaged RRAM arrays used in this work. **b** I-V characteristics of the 1T1R RRAM devices (device-to-device measurements) at fixed V$_{STOP}$ as a function of the compliance current I$_C$ in order to study the switching mechanism of the synapses under different operative conditions. Note that the compliance current is directly managed by acting on the gate voltage V$_G$ of the selector of the cell, which is an nmos transistor. By sending pre-neuronal spikes at the gate of the selector and biasing the top electrode of the RRAM synaptic element, a post-synaptic current is generated and used for the post-neuronal computation. During the fire events, a programming signal is superimposed to the bias of the top electrode in order to set or reset the memory device. **c** Typical low-resistive (LRS) and high-resistive (HRS) distributions using I$_C$ = 74 μA and V$_{STOP}$ = −1.5 V. **d** Multilevel LRS at increasing I$_C$ with the average resistive value μ$_R$ (**e**) and the corresponding standard deviation σ$_R$ (**f**): note that the precision of the synaptic weight is dependent on the module of the programming current (higher power, higher precision of the synaptic weight). **g** Modulation of the HRS as a function of V$_{STOP}$ sweep with the extracted σ error bar: the higher, in module, the stop voltage, the higher the resistance that is obtained.

dynamics into neural networks would thus improve robustness and reliability of artificial intelligent systems.

## The event-based architecture

A resilient hardware should be reactive to the events that occur in its surrounding to experience every event in terms of penalties and rewards, Fig. 2a. Considering the autonomous navigation, it is possible to describe the succession of decision-making situations by means of bio-inspired instances. For example, the movement between two positions can be modelled by the firing activities of two neurons (PRE- and POST-) connected by an RRAM synaptic element, accordingly to the STDP procedure. The post-synaptic current, which depends on the state of the RRAM, Fig. 1b, is integrated and then compared to the internal threshold of the post-neuron, Fig. 2a. If the internal threshold is overcome, a programming signal arises, and it directly potentiates the synaptic element by means of a feedback to the top electrode of the synapse connecting the current position with the firing neuron. The synaptic signal can be also depressed after spiking events of random neurons selected by means of linear-feedback shift registers (LFSRs): in this case, a "refractory period" of 1μs is considered, as in biology[36]. Every neuron can be equipped with a further synaptic device (named "state") which directly affects the firing threshold V$_{TH}$ during the learning activity, Fig. 2a. Thus, synaptic and internal RRAM devices constitute a proper framework for the definition of the navigation problem.

To introduce the high-level functionality of the system, we propose in Fig. 2b the flowchart of the reinforcement learning algorithm for autonomous navigation. Note that the system can start from pure random initial conditions. However, it is preferable to prepare high resistive internal states for the V$_{TH}$ state matrix, |V$_{STOP}$| = 1.1 V in Fig. 1g, and moderately low resistive synapses, I$_C$ = 54 μA in Fig. 1d. This choice makes the neuronal integration faster. Note that some random positions are selected to provide initial stochasticity to the system, hence finally getting bimodal distributions for the RRAM matrices. When the exploration starts, the post-synaptic currents are integrated by the nearest neurons, identified by the cardinal positions, eventually leading to firing activities and thus to movements of the agent[52]. This behaviour is similar to what is observed in bio-inspired winner-take-all (WTA) networks, where the output neurons compete with each other to specialize on different tasks[53,54].

The bio-inspired behaviour is mapped in hardware by means of a digital system on chip (SoC) with a microcontroller and an FPGA embedded, managing several RRAM arrays and output neurons. The RRAM synaptic arrays shown in Fig. 2c are used for mainly three reasons: (i) to connect the SoC with the CMOS neurons taking advantage of the synaptic matrix vector multiplication (MVM)[23]; (ii) to track the position of the agent at each instant[54]; (iii) to implement bio-inspired homeostatic STDP[55]. At each step of exploration, the FPGA collects the current position (i, j) of the agent and stimulates its nearest neurons by sending pulses at the gate of the corresponding (i, j) 1T1R RRAM

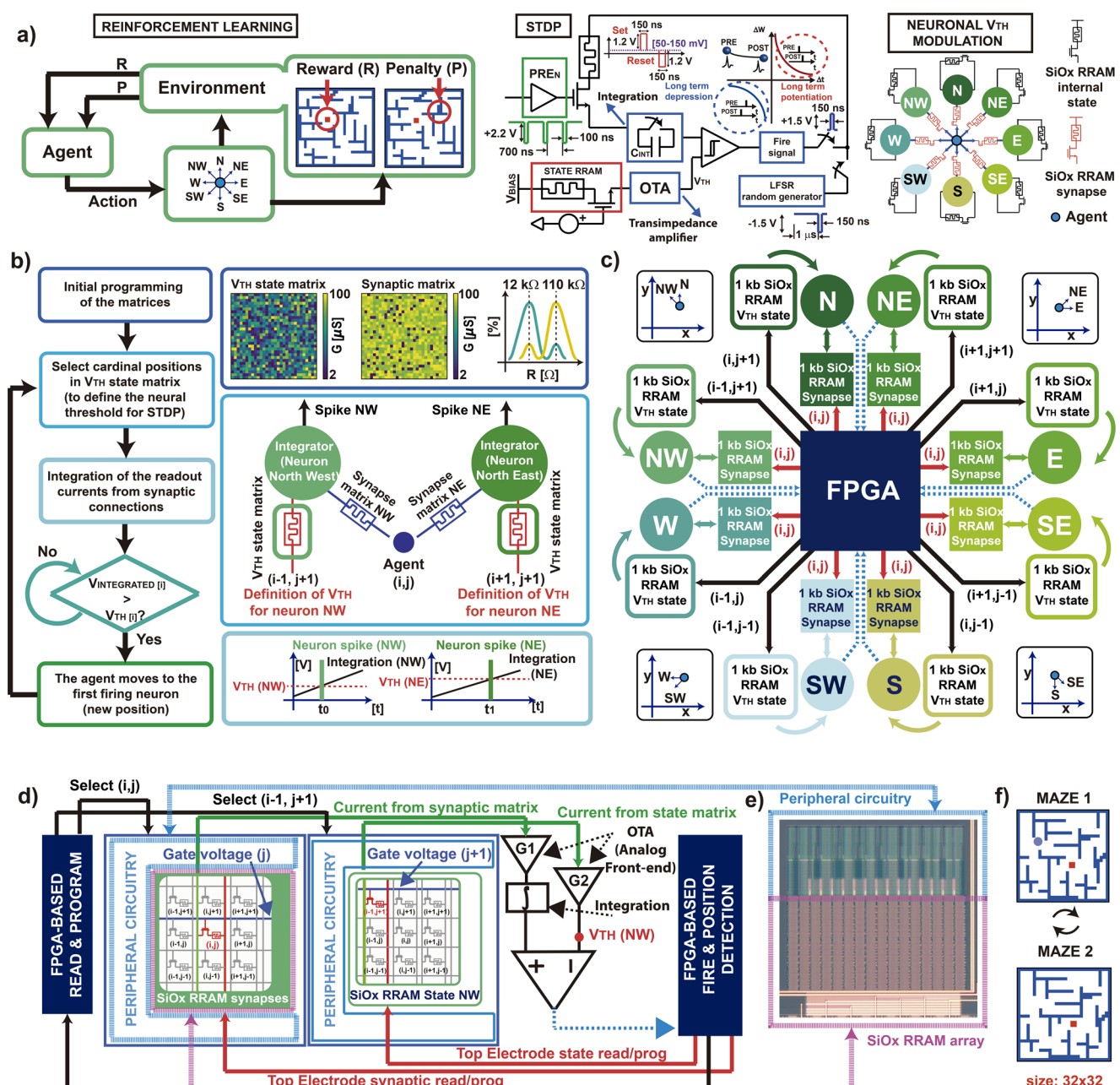

**Fig. 2 | Flow-chart of the reinforcement learning procedure implemented in hardware. a** Representation of high-level reinforcement learning for autonomous navigation considering 8 main directions of movement: an agent (e.g., a robot) interacts with the environment by means of decision-making events which eventually lead to penalties or rewards that modulate the next actions. The direction of movement between two positions is ruled by the STDP. The pre-neuronal signal (current position of the agent) excites the gate of the selector of the synaptic RRAM element by sending a sequence of rectangular pulses while the TE of the synapse is biased at a read voltage (between 50 and 150 mV). The consequent post-synaptic current is integrated in the post-neuron and compared with the internal threshold (ruled by a further "state" device) eventually inducing fire activities which potentiate the synaptic element and mark the direction of movement. Note also that LFSR

registers can select random neurons for sending stochastic depression signals. **b** High-level description of the bio-inspired reinforcement learning procedure implemented in hardware. Note that, for best operation, the initial combination of the RRAM matrices is bimodal. **c** Block scheme of the hardware, with the "synaptic" and "internal states" RRAM arrays, the FPGA and the 8 neurons that stand for the 8 cardinal directions. **d** Example of the operative condition of the firing neuron NW with respect to the synaptic and internal state arrays. The internal threshold is modulated by the resistive state of the internal RRAM device which changes as a function of the fire activity: an analogue front-end is also necessary for a correct definition of the post-synaptic currents. **e** Top view of the memory array and of the integrated circuital periphery for the management of the memory addresses. **f** Example of a dynamic maze to test the systems for reinforcement learning tasks.

synapses (refer to Fig. 2c for the block scheme and to Fig. 2d for the description related to a specific firing example). Once a neuron, e.g. neuron north (NW), reaches its internal threshold $V_{TH}$, the corresponding synapse (i, j) is set to LRS for the STDP mechanism[36,54,56]. At the same time, the neuron sends the fire signal back to the FPGA, represented by the dashed lines in Fig. 2c, d. Once the FPGA stores the

signal, it consequently selects the address of the firing neuron, in this example position (i−1, j + 1) of neuron NW (north-west), and partially sets its internal RRAM state. This procedure directly affects the internal threshold of that neuron, since the slight decrease of resistance of the internal state (modulation of the firing activity) prevents to reach the threshold of the neuron itself in the next integration phase. Note that

in the case of a spatial movement, each position can be ideally reached from all the geographical coordinates (north, south, north-east,…); for this reason, the modulation of the threshold is performed at every (i−1, j + 1) state position, i.e. for every internal state array. The partial set of the internal state device keeps track of the history of that position, thus configuring, for every explored point, a dynamic behaviour in time. Furthermore, note that the movement of the agent might also be backward: this is assured by the distribution of the synaptic arrays as a function of the direction, Fig. 2c.

Moreover, note that the parallelization of the synaptic arrays and the replication of each internal state for every direction towards a point, Fig. 2c, d, allows the use of only 8 CMOS neurons. This is due to the fact only the threshold modulator, i.e., the RRAM device selected by the decoders in the circuital periphery, changes for every explored position, while the hardware of the neurons remains the same. Such choice goes along with the efficient hardware architecture we fabricated for the RRAM arrays, and it is relevant in terms of Giga-operations per second (GOPS/mm$^2$), since the RRAM devices are built in the backend of the line. Thus, reducing the number of CMOS neurons is a key point to enable the scalability of the hardware in terms of power and area consumption.

In order to provide a concrete case study, we describe now the exploration of a dynamic maze whose walls dynamically move in time, Fig. 2f. The goal of the network is to find a final reward, the red square, by successive trials, each one limited in time.

## In-memory computing for autonomous navigation

The maze can be pre-designed by an external user into the FPGA. This architectural choice allows to test the system without building a physical agent (e.g. a robot) in a real maze. The environment is configured as a matrix of 32 × 32 positions, thus requiring for this test case a total of 16,384 RRAM devices (8192 for the internal states and 8192 for the synaptic connections), as indicated in the block scheme of Fig. 2c.

The current position in the maze is defined by the address of the internal states that the FPGA operates in a particular moment, e.g., position (i, j) = (2, 2) in Fig. 3a. At every position occupied by the agent, the signals sent by the FPGA via the synaptic connections cause the integration and eventually the fire of one of the 8 post-neurons. The fire event of a neuron (N, in the case of Fig. 3b) causes (i) the potentiation (resistance decrease) of the connecting synapse, Fig. 3c, and (ii) the inhibition of all the other neurons by discharging the signals stored in the integration blocks, Fig. 2a, as it happens in WTA networks[54]. Once this procedure is completed, a further signal is sent back to the FPGA, which now moves the position of the agent to that indicated by the firing neuron, i.e., (i, j) = (2, 3), Fig. 3d.

Consequently, the RRAM elements of all the eight internal states related to that address are partially set, Fig. 3e, thus increasing the internal threshold of that specific position (for clarity, only one $V_{TH}$ increase referred to the spiking position is shown). The control of the internal threshold is fundamental for the management of penalties and rewards. For instance, when the agent hits a wall, it will then try to find the escape path along other directions. Conversely, if the final reward is found, the agent is likely to remember the last occupied positions to ease the successive trials towards the solution.

In order to better clarify the role of the recurrent state, Fig. 3f–h describe the evolution in time of an internal state under different situations. Figure 3f shows the modulation of the internal state for an ordinary (i.e. without reward or penalty) position. Every time the neuron fires, it increases its internal threshold, thus reducing its firing excitability and promoting the exploration. When the agent touches a boundary (a wall) it receives a penalty, which increases the internal threshold of that neuron for the successive trials, Fig. 3g, and reset to HRS the corresponding synaptic connections, thus mimicking the sensorial receptors of a mouse swimming in a water maze[16]. On the other hand, when the agent finds the final reward, the FPGA incrementally

reduces the internal thresholds of the 10 last positions, Fig. 3h. On the other hand, note that the RRAM resistance value of the internal threshold can reach an upper limit, Fig. 3i, thus building a boundary condition to the learning activity based on homeostatic STDP.

A synapse is set to LRS when a PRE-neuron fires before a POST-neuron to make the agent moving ahead, Fig. 2a. However, accordingly to STDP-based Hebbian learning[54], we also insert random spiking activity at low frequency by means of LFSRs. Once the LFSRs generate the coordinates of random positions, the hardware system sends "reset" (negative) programming pulses to the top electrode of the selected devices, superimposing the programming signal to the bias voltage, Fig. 2a[37]. If this happens in parallel with the excitement of the synaptic gate, a reset occurs, thus providing stochasticity to the network and random depression, Fig. 3j[56]. Note also that a "digitalized STDP" (binary potentiation/depression) is enough for providing efficient operation without accuracy loss[36]. Furthermore, the synaptic connections are potentiated toward the direction of movement and depressed when a penalty occurs, Fig. 3k, thus inhibiting the movement towards inconvenient directions in the next trials. The synaptic connections and the internal states of the ordinary positions are re-initialized at every trial, which ends after reaching the time limit or when the reward is found, as in biological experiments[16]. Note that if the environmental configuration changes and the previous rewarded path is inhibited by a new wall, the threshold increases more slowly, since the internal state starts from a higher resistive value, Fig. 3l.

## Exploration, optimization and recall

The experiments follow the same procedure used in the case of the Morris Maze in biology: the agent has a limited time to explore the environment under successive trials[16]. Once a trial starts, the sequence of firing neurons maps the movement of the agent in the environment, Fig. 4a. The exploration is configured as successive random walks which progressively develop a model of the environment. When the solution is found, it is remembered and improved in time, until the environment changes and another escape path must be found. In this situation, the agent gets a penalty in unexpected positions (refer to snapshot number 4 in Fig. 4a, where the lighter colour indicates the longest time spent by the agent in those points). However, when the system comes back to the previous configuration, it easily recovers the first solution.

Once the global reward is found, the system incrementally reduces the internal threshold $V_{TH}$ of the last 10 positions by resetting the internal state resistances of every "state" array, speeding up the overall response of the network. This is evident in Fig. 4b, where the time needed to reach the reward is measured as a function of the number of trials and then averaged over 50 experiments. At the beginning, the system cannot find the solution and the time spent in the maze is the maximum available. Then, if the solution is found, the system progressively decreases the computing time. When the maze changes shape, the network starts to fail; however, after an exploration period, it successfully gets to the target again. Once the system is brought back to the first configuration, the previous solution is retrieved faster than before. This is due to the "recall property", which is related to the residual memory of the internal states and to the intrinsic recurrent structure[19]. Supplementary Movie 1 illustrates the experimental setup and the hardware demonstration of the exploration of the dynamic environment via reinforcement learning. Supplementary Code 1 provides the simulation code for the maze exploration via reinforcement learning.

The overall average energy per trial required by the system was extrapolated by simulation studies of the hardware setup. Considering an average case of 50 repetitions of the same exploration, as in Fig. 4c, the major energetic contribution is related to the use of the SoC. On the other hand, the switching activity of the arrays is considerably high when the network has to modify its internal structure to get self-adaptation to the environment, thus causing a non-negligible increase of the power consumption. The energetic efficiency improves with the

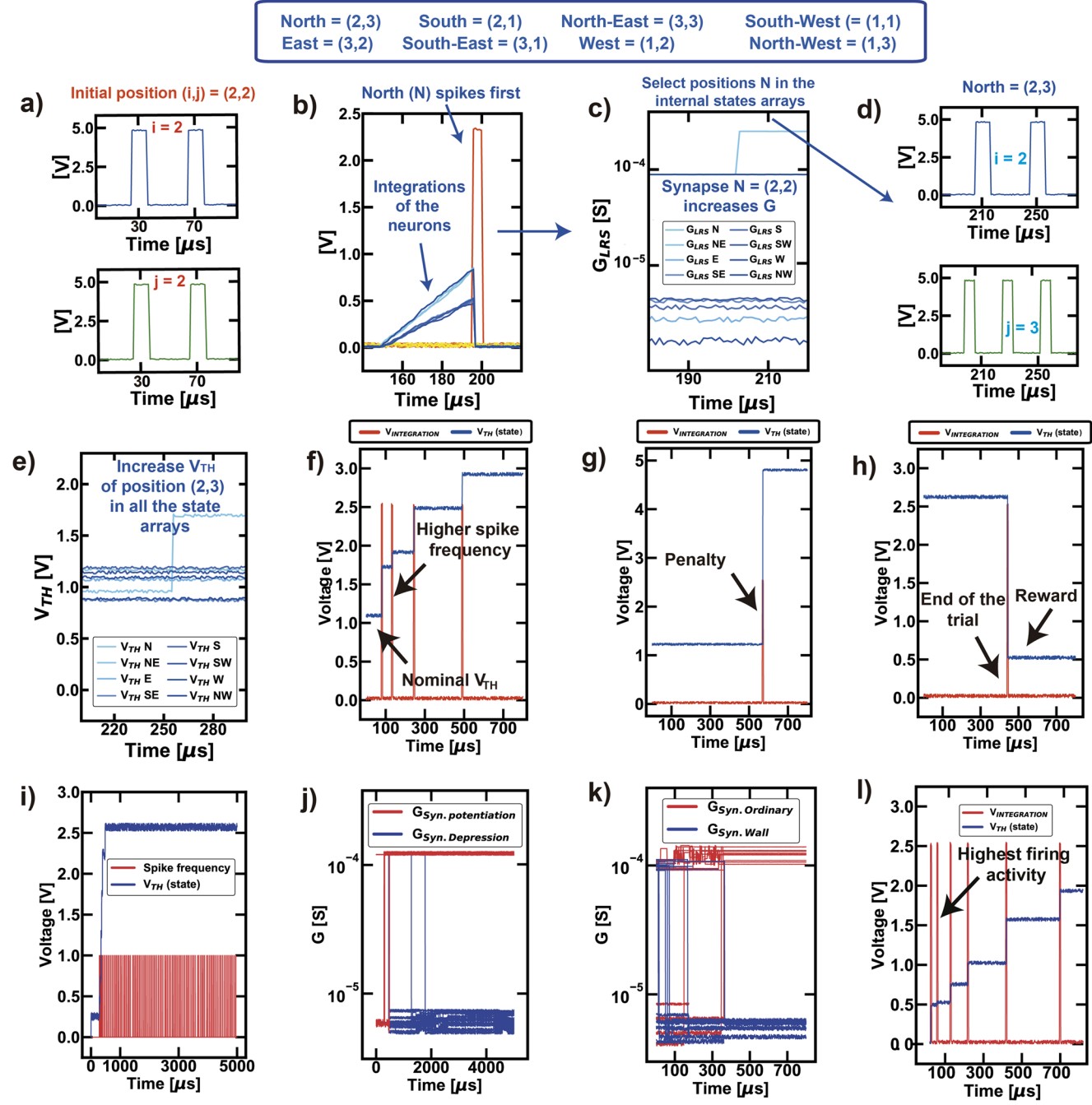

**Fig. 3 | Step-by-step description of the main signals ruling the autonomous navigation. a** The FPGA records the current position of the agent and **b** triggers the gate voltage signal of the synaptic devices to start the integration phase of the nearest neurons. Once a neuron fires, all the integration signals are discharged by switching on a transistor in parallel to the capacitor used for integration. After the fire event, the corresponding synaptic connection is brought high (**c**) and the current position of the agent is updated (**d**); the threshold of the new internal state rises as a consequence of the internal state partial set (**e**); the procedure (**a–e**) is repeated at every movement of the agent. **f** If a position (**i, j**) is accessed consequent times, it plastically adapts the corresponding internal thresholds causing a gradual increase of the threshold $V_{TH}$; the neuronal threshold plastic adaptation is also used

to map the penalties, by increasing the corresponding $V_{TH}$ (**g**), and the rewards, by decreasing the corresponding $V_{TH}$ (**h**). Note that the gradual increase of the neuronal threshold is bounded to the effective multilevel capability of the RRAM devices (**i**). During the ordinary movement, the synaptic connections from one position to another are potentiated or depressed for the STDP mechanism (**j**), while, on the other hand, the penalty positions always undergo depression, due to reinforcement learning (**k**). Note that the synaptic connections are always potentiated if the agent does not come back. If rewarded positions run into a penalty due to the dynamic evolution of the environment, the corresponding internal thresholds rise slower than the ordinary positions, due to the firing history and the different fire excitability (**l**).

increasing accuracy of the system to get to the reward since the integration time of each step decreases.

Figure 4d shows the modelling of the environment, after several trials starting from different initial points, as a function of the internal states, considering the corresponding threshold associated to each

position. Note that near the changing walls the averaged threshold is lower, due to the presence of two escape paths depending on the current configuration of the environment. In any case, the overall firing activity of the neurons is higher for the positions near the reward, Fig. 4e.

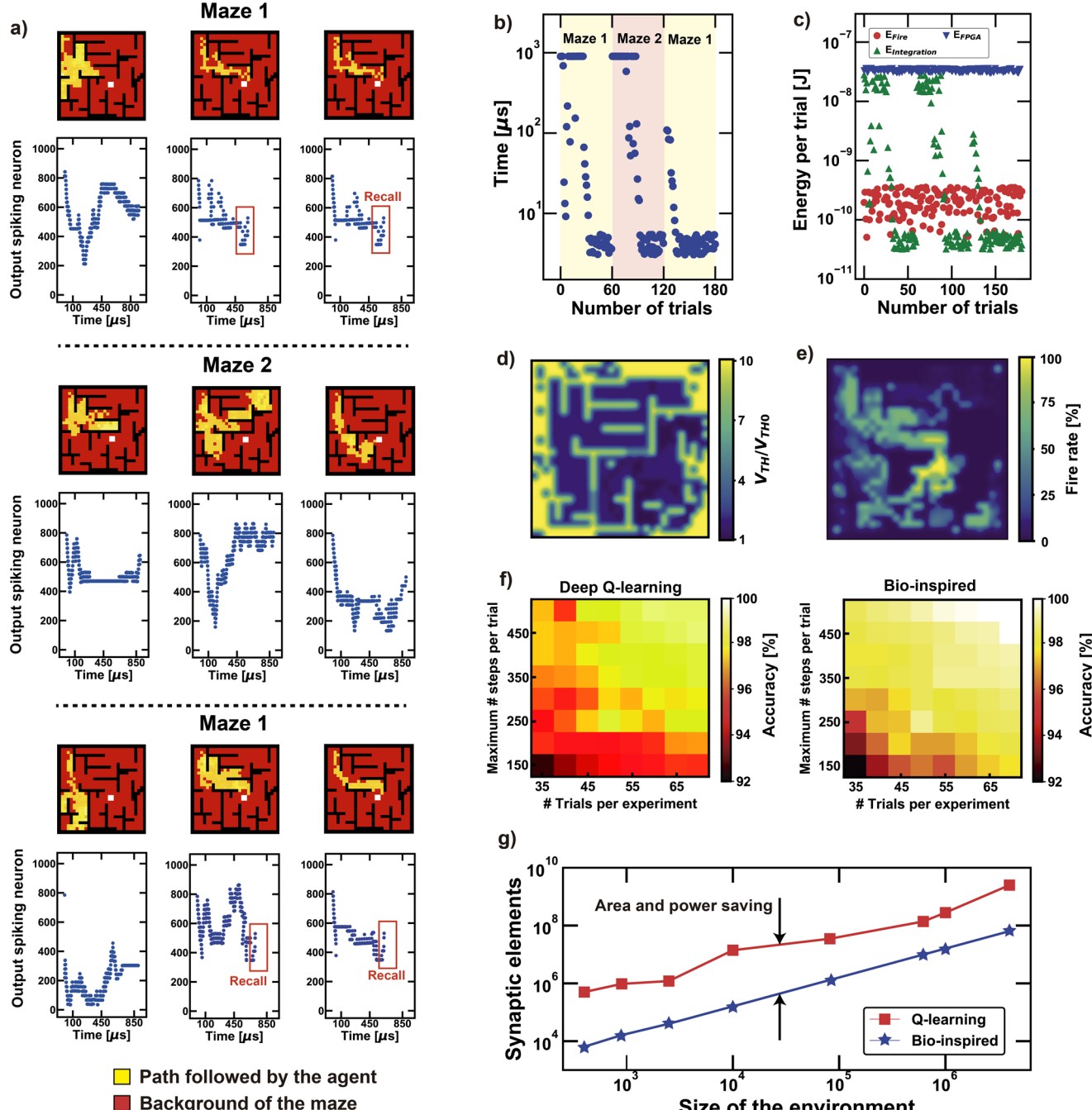

**Fig. 4 | Recall property in dynamics environments and power efficiency of the system. a** Experimental results for 9 successive trials of a maze which changes topological configuration every 3 trials. The system explores the environment to find the reward and it recalls the first solution once the previous configuration is proposed. **b** The time to get the solution improves from trial to trial along with the optimization of the policy. However, note that, once the maze changes shape, the reward time increases accordingly since a new solution must be found. When the maze comes back to the previous situation the first solution is recalled. **c** Energy consumption tendency for each core of the system. **d** Once the initial point is changed from trial to trial, the energy consumption stays high, but a policy map of the whole environment is retrieved. **e** Map of the firing rate of the neurons, showing

that the highest values are, on average, in the nearby zone of the final reward. **f** Colour maps of the accuracy for standard Python-based deep Q-learning and the proposed bio-inspired approach under the same benchmarking condition. Note that the bio-inspired hardware assures better accuracy results for every combination of explorative parameters (number of trials per experiment and number of steps per single trial, i.e., exploration time). **g** Comparison in terms of memory computing elements between the deep Q-learning procedure and the bio-inspired solution at increasing sizes of the environment to explore. Note that the power consumption is also furtherly improved in the bio-inspired solution thanks to the use of RRAM memory devices built in the back end of the line, which avoids the von Neumann bottleneck typical of standard computing platforms.

To propose a fair benchmark with respect to the state-of-the-art, we have studied the efficiency of standard approaches under the same environmental configuration ("maze 1") depicted in Fig. 4a. The standard approach of the conventional free-model reinforcement learning was developed using the Python framework (https://pypi.org/project/

pyqlearning/). As shown in Fig. 4f, the bio-inspired solution overcomes the state-of-the-art free-learning algorithm in terms of accuracy. The better results are due to the more plastic and resilient algorithm to find the solution, which leads to a faster convergence to the optimum result.

Furthermore, the bio-inspired approach shows a far better management of the computing resources with respect to standard solutions, Fig. 4g. In fact, by calculating the number of memory elements that are needed for carrying out an exploration at a certain average accuracy (99%), it comes out that our solution is 10 times cheaper (the number of computing elements is directly proportional to the area/power consumption). Related to this topic, note that the most efficient supervised CNN (convolutional neural network) necessary in standard deep reinforcement networks for achieving the 99% target, requires more than 5 million of parameters with almost 1 billion of multiply and accumulate operations[57]. On the other hand, the bio-inspired approach proposes a better computing architecture (for the matrix-vector multiplication, Fig. 2c) and exploitation of the resources (plasticity is assured thanks to the STDP).

We report more information about the benchmark with respect to the state-of-the-art in the section "Discussion" and in the supplementary information, where we also propose an appendix for the theoretical comparison in terms of resilience.

## Mars rover navigation

In the last few years, there have been several works which have started to apply reinforcement learning algorithms to terrain images of Mars, to test the artificial networks based on learning by reinforcement in harsh natural landscapes[58,59]. In particular, the terrain images of Mars were taken from HiRISE, which collects an entire dataset of high-resolution Martian images[60].

We performed the Mars Rover Navigation test step-by-step. The first regarded the definition of the environment from readapted HiRISE pictures: we elaborated the images, and we individuated slopes and descents to build a proper description of the environment in the SoC; secondly, we defined the experimental setup; thirdly, we performed the experimental measurements. Finally, we used the experimental measurements for performing Monte Carlo simulations and assessing performance studies.

Figure 5a shows an example of readapted satellite image with dimension $128 \times 128$. The goal of the rover is to find the path to get to the target without fatal errors, i.e., the fall into a crater or the roll-over while climbing a rock. The penalty mechanism, in this case, is referred to the slope of the frame of the rover, which cannot overcome 15 degrees. Figure 5b shows the trajectories of the agents during various attempts of the rover to get to the reward. Note that, if the rover has a single-shot trial of exploration, the optimum path is not assured due to the lack of a model of the environment. However, once the rover experiences random walks in the Martian environment, it progressively maps the morphology of the territory, highlighting the forbidden locations (i.e. craters and hills) by remembering the received penalties, Fig. 5c. Note also that the capability of adaptation to the environment enables a mapping in time of hills and craters in case of morphological changes.

The dynamic self-adaptation of the neural network depends on the required time to get to the solution: firstly, the system creates a model of the environment experiencing penalties; secondly, it finds the solution and tries to progressively optimize the time to get to the reward. The time evolution of the system depends on the iterative procedure of integration and fire that is performed for every position P occupied by the agent. Thus, given a starting point, successive trials of exploration lead to the definition of preferential paths toward the reward, as indicated in Fig. 5d for the number of movements needed to reach the reward as a function of the number of trials.

## Reconfigurability of the hardware

The Mars Rover navigation proposes a case study which is more demanding in terms of power and area consumption, since it deals with a larger environment where reliability, resilience and accuracy play a key role. Similar types of exploration are relevant for several tasks. For instance, robots from "Boston Dynamics" have been used in archaeological areas to inspect hard-to-access sections of the ruins, to collect data and to alert people for safety and structural problems whenever some unexpected changes are detected (https://www.washingtonpost.com/world/2022/03/31/pompeii-robot-dog-patrol-boston-dynamics/). In this section, we are going to discuss the scalability of the bio-inspired hardware in the framework of the Mars Rover navigation, investigating the best management of the computing resources and demonstrating that the proposed recurrent neural network can infer abstract strategies. A further appendix related to this topic, "Additional insights over the scalability topic", is also proposed in the supplementary information.

In order to demonstrate the scalability of our system, we compare the exploration of a new environment using two different approaches, namely (i) the step-to-step mapping described in Figs. 3 and 4 and (ii) the optimized exploration using transfer learning from previous trials.

This latter approach is based on two steps which enable the re-use of previous information. During the first step, Fig. 5e, small sections of the old policy map are dissected in order to record random shapes. The record is simply driven by the integrated current of all the RRAM devices included in the region of the memory under consideration, choosing only those sections which are far enough from the maximum and minimum boundaries (i.e. all LRS devices and all HRS devices). Once this procedure is iterated for different forms, the set of shapes is recorded in the FPGA and stochastically used as penalty function (red squares) during the exploration of the new environment, Fig. 5f. Such approach improves the efficiency results, Fig. 5g, and it avoids the physical device-position mapping (a single address is enough to abstract a region of space when the agent touches a penalty).

Note that the system is flexible because it can be easily reconfigured during operation. For instance, it would be also possible to re-write old, allocated memory arrays within the same trial in order to dynamically improve the RRAM memory efficiency over time. Such reconfigurability, which enables the use of the same hardware for different autonomous navigation tasks, goes in the direction of providing hardware-based computation while retaining the flexibility of a software approach, as it is done in reconfigurable FPGAs[61].

Furthermore, the RRAM-based computation is performed "in situ", thus offering a far better management of the computing resources. In this context, the memory array can be continually exploited by the bio-inspired computation until the complete memory resource is fully allocated. Then, the direction of movement, the number of steps and some further information (penalty/rewards) can be saved in separated registers (which could also be RRAM-based) as pure coordinates. Thus, the RRAM array is practically ready again to perform further explorative trials, abstracting the previous maps of exploration and referencing the stored coordinates to the effective number of "refreshes" of the RRAM memory arrays.

## Theoretical modelling for in-memory reinforcement learning

In order to study by a theoretical point of view the benefits introduced by the in-memory bio-inspired approach in terms of energetic efficiency, resilience and accuracy, consider again the Q function, Eq. II. We analyse now the same equation accordingly to the main outcomes related to the hardware presented in this work.

- The reward function $R(s, a)$ of the hardware is mapped by the homeostatic reaction described in Fig. 2a, b and in Fig. 3i: the environment gives penalties and rewards which directly affect the quality "Q" of a position "s" by acting on the "state" RRAM devices. If a penalty occurs, or a reward is found, the firing neuron threshold is modulated, Fig. 3g, h. Note that the firing neuron is the one which overcomes its threshold first, i.e., $(I_{out} - I_{th}) > 0$, in which $I_{out}$ is the post-synaptic current, and $I_{th}$ is the equivalent homeostatic threshold of that neuron. Thus, keeping constant the read voltage and being the neuronal current dependent on the synaptic elements, the reward function can be

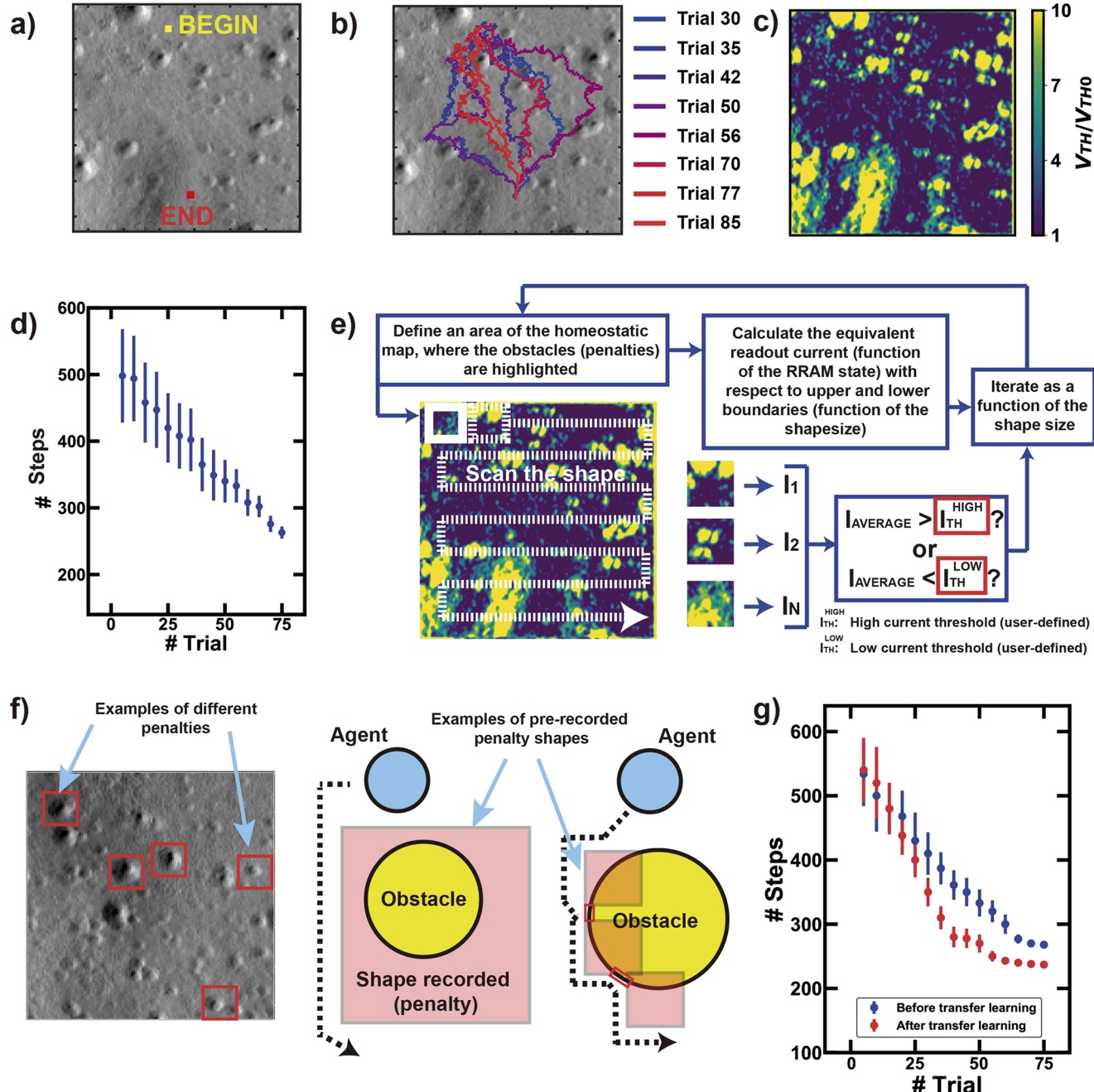

**Fig. 5 | Reconfigurability and scalability of the hardware under the Mars Rover navigation test. a** Custom environment of Mars readapted from HiRISE, with highlighted the initial (start) and the final (global reward) points selected for the test of the algorithm. **b** The agent explores the environment in 100 trials, eventually finding the target: note that the successful trials improve the strategy step-by-step to get faster to the solution. **c** Various trials of exploration lead to the creation of a complete policy map of the whole environment, with higher equivalent threshold of the positions which received penalties. **d** Time improvement of the exploration path: after selecting a starting point, the policy map drives the system to optimize the number of steps to get to the final reward. **e** Iterative selection of small sections of the previous policy map by reading the integrated current of the state array to record generic shapes of the penalty-related objects. Note that this procedure can

be iterated as a function of the shape size by choosing proper boundaries to avoid misleading cases (e.g., sections of the policy maps in which no shapes are detected). **f** Exploration of a new environment taking into consideration different sizes of penalty shapes: once the agent receives a penalty, it is possible to inhibit a generic pre-recorded area of the environment, thus avoiding a memory-position mapping. Furthermore, the memory array can be continually exploited by the bio-inspired computation until the complete memory resource is fully allocated. Then, the direction of movement and the further information can be saved in peripheral registers as pure coordinates. **g** Study over 100 experiments of the time improvement of the exploration path comparing the free-policy with the optimized policy using recorded penalty shapes, eventually highlighting the benefits of transfer learning from previous explorations.

written as a function of the conductance values only, i.e., $R(G_{syn}, G_{state})$.

- The learning factor is not required in the neuromorphic approach, since it is a parameter related to the deep learning

procedure. However, we introduce here a generic fitting factor $\beta$ for the modulation of the quality factor equation.

- The probability function "$P$", which describes the probability of the quality of a certain position $s$ for moving towards another

state $s'$, is dependent on the homeostatic-based STDP mechanism. Thus, it depends on the synaptic evolution of the synaptic connection between state $s$ and $s'$, Fig. 3j, in formula, $P = P\left(G_{syn}(s, s'), G_{state}(s), \frac{\partial G_{syn}(s,s')}{\partial t}\right)$[36].

- The value $\max_{a'}Q(s', a')$ is the maximum of all the possible values $Q(s')$ among the possible states $s'$ the agent could explore after an action $a'$. This feature is mapped by means of the synaptic-based movement, since the Q factor of each position is modelled over time by the plastic modulation of the synapses, Fig. 4e. Thus, we can rewrite this contribution as $\max_{s'}Q\left(G_{syn}(s', s), G_{state}(s'), \frac{\partial G_{syn}(s',s)}{\partial t}\right)$, where the inverted $s$ parameters stand for the possibility, depending on $R(s')$, of going on exploring or coming back to the previous state, Fig. 2a.

Thus, the neuromorphic Q-learning equation, that we now call $Q_N$, can be re-written using this formula:

$$Q_N(s) = R\left(G_{syn}(s, s'), G_{state}(s)\right)$$
$$+ \beta \sum_{s'} P\left(G_{syn}(s, s'), G_{state}(s), \frac{\partial G_{syn}(s,s')}{\partial t}\right)\max_{s'}Q\left(G_{syn}(s', s), G_{state}(s'), \frac{\partial G_{syn}(s',s)}{\partial t}\right). \tag{4}$$

Following the same procedure, it is also possible to describe the TD($\lambda$), Eq. III:

$$TD_{N,t}(a, s) = \rho\left(Q_{N,t}\left(G_{syn}, G_{state}\right) - Q_{N,t-1}\left(G_{syn}, G_{state}\right)\right), \tag{5}$$

where $\rho$ is a fitting parameter. Note that we have re-written all the reinforcement learning equations in terms of memory-based circuital parameters, but it is also possible to furtherly develop the theoretical study by highlighting the mathematical relationships behind the pure functional representations of Equations IV-V. In the supplementary material we provide the appendix "Additional insights over the theoretical modelling of bio-inspired networks for reinforcement learning" to link the reinforcement variables to the physical parameters of the circuit.

All these outcomes highlight the advantages of the bio-inspired approach since (i) time and power consuming data transfers between CPU and DRAM are avoided and (ii) the system can rely on pure synaptic adaptation to carry out accurate computation. By a theoretical point of view, this is the most relevant achievement introduced by this work since it highlights the intrinsic benefit of neuromorphic in situ-computation with respect to the state-of-the-art.

## Discussion

Deep learning techniques using standard Von Neumann processors enable accurate autonomous navigation but require great power consumption and long time for making training algorithms effective, Fig. 4f, g (https://pypi.org/project/pyqlearning/). In particular, the environmental information is often sparse, noisy and delayed, while training procedures are supervised and require direct association between inputs and targets during the backpropagation. Hence, complex models of convolutional neural networks are needed to numerically find the best combination of parameters for the deep reinforcement computation[62], (https://pypi.org/project/pyqlearning/). Thus, the standard approaches to reinforcement learning enable free-policy learning by reinforcement, but this is paid in terms of lower accuracy with respect to the same environmental configuration, Fig. 4f, and in higher cost of the resources when a specific performance is targeted, Fig. 4g. Furthermore, standard processors require data transmission back and forth the DRAM (Von Neumann bottleneck) while in-memory computing assures a local processing of the information where it is stored, Fig. 1a[22,23,63].

Note also that deep Q-learning techniques suffer from unstable learning under some conditions of bias overestimation which requires a

mutual training of a multi-layer-perceptron (MLP) network and a correct setting of the learning rate[64]. This could affect the effectiveness of the training algorithm when the system must map the environment autonomously, requiring a network of several layers with the Adam optimizer applied for stochastic optimization[65]. All these features assure high accuracy in, at least, 1000 episodes for each trial. Contrarily, the bio-inspired learning procedure relies on training-free in-situ hardware computation. This approach improves a lot the time efficiency, Fig. 4b, and the energy consumption, Fig. 4c, while keeping high the accuracy, Fig. 4f. Furthermore, the STDP does not require dedicated methods for stochastic optimization, and it assures an optimum behaviour also when the configuration of the space to explore is not constant[36]. Related to this context, in the supplementary appendix "Comparison of the resilient properties between bio-inspired and deep learning approaches", we report a theoretical study over the adaptation capabilities of the neuromorphic solution with respect to the standard Python-based approach (https://pypi.org/project/pyqlearning/).

The Markov decision process, Q-learning, TD($\lambda$) and deep learning are not the only topics to which the scientific community refers to for modelling and designing reinforcement learning algorithms. For instance, the multi-bandit problem is often taken as benchmark. The multi-armed bandit problem deals with an agent that attempts to make decisions as a consequence of previous experiences but, at the same time, it needs to acquire new knowledge for the next decision-making events. To cope with this framework, several works have proposed the use of RNNs for enhancing the re-use of past information[66] and for building "meta-learners", i.e., systems trained on a distribution of similar tasks featuring a generalization capability when novel goals are targeted[67,68]. However, even considering these meta-approaches, several CNN-based training algorithms are anyway necessary to provide the system with an optimum policy map for the required navigation task, thus falling again in the power and time bottleneck.

In conclusion, we proposed an event-based hardware based on RRAM devices capable of self-adaptation to get efficient neuro-computing in reinforcement learning tasks. We studied the experimental behaviour of the network highlighting the resilient capability of the autonomous navigation under various environmental difficulties, such as obstacles and dynamic modifications of the maze. We also proposed a study of the hardware reconfigurability of the system under the Mars rover navigation test. Finally, we introduced a theoretical framework for bio-inspired reinforcement learning highlighting the main outcomes of RRAM-based computation with respect to the state-of-the-art. This work highlights the relevance of bio-inspired approaches for artificial intelligence and underlines the computational benefits of non-volatile memories for autonomous hardware systems.

## Methods
### SiO$_x$ RRAM arrays
The RRAM devices are deposited in the backend-of-the-line (BEOL) on top of the 4th metal layer of 130 nm-technology CMOS wafers. First, a TiN bottom electrode (BE) is created as an inert electrode. Afterwards, an optimized resistive switching layer of SiO$_x$ is deposited, followed by a Ti layer (playing the role of oxygen scavenging layer) and a TiN layer. The memory dots are obtained by etching. Then, a passivation layer is deposited. Finally, the top electrode (TE) contact is opened, and the 5th metal line is processed to complete the integration process. Note that every state array is separated from the synaptic array and each of them has a dedicated direct-memory-access (DMA) circuit addressable by proper pad connections (refer to Supplementary Fig. 4).

In particular, each of the integrated circuits used for the experiments proposed in this manuscript enables the use of 96 kb bonded devices. Given the high number of available arrays, further memory elements could be easily accessed by providing more bonding wires to the package (until 1 Mb). Note also that the maximum dimension of the array that can be accessed using only one DMA is around 16 kb

(128 × 128), while the smallest fully connected array addressable by the hosting board and the experimental setup shown in Supplementary Fig. 3 has a dimension of 8 × 8. Finally, note that the high reconfigurability of these arrays gives the possibility of choosing different top-level architectures for taking advantage of different features of the devices, depending on the application and target.

The devices can be accessed via two digital signals, one for the row and the other for the column, respectively. All the devices were electroformed by applying an increasing amplitude voltage sweep relying on an automated setup with a parameter analyser (HP4156C). In order to characterize the devices and study the main features in terms of resistive window and multilevel capability, a "write and verify" algorithm was used, constituted by a series of alternative application of write and read pulses. To provide both set and reset switching activities, the polarity of the cells was switched accordingly to the transition to obtain: during set, a positive polarity was applied to the top electrode of the cell; during reset a positive polarity was applied to the bottom electrode of the cell. Conversely, the device was also tested by applying positive pulses at the top electrode during set and negative pulses during reset. During reset the maximum gate voltage is applied to the gate of the transistor to get the lowest possible ohmic resistance. The data were collected and studied using Matlab or the libraries "Matplotlib", "Pandas", "Seaborn" in Python environment. As illustrated in Fig. 1b, the switching behaviour of the devices allows to obtain a $V_{SET}$ from 1 to 1.35 V. Thus, in the operation setup, a sufficiently high voltage $V_{SET}$ was always used in order to guarantee a good switching behaviour; the same was assured for the reset transition. However, note that significantly high standard deviation affects the HRS multilevel capability of the devices, Fig. 1g. On the other hand, the multilevel low resistive capability of the devices as a function of the compliance current $I_C$ is more stable, even if the distributions of Fig. 1d present overlaps between one another. However, this is not important for the overall computation of the network, since bio-inspired computing does not require a precise definition of the weights. Note that in the supplementary information we provide a further appendix, "Additional insights over the theoretical modelling of bio-inspired networks for reinforcement learning", which investigates the mathematical connection between the reinforcement equations IV and V with the physical parameters ruling the resistive state of the RRAM devices.

## Simulation setup

In order to design the system, a high-level environmental simulation was implemented in Matlab. The simulation was initially carried out with an ideal definition of the weights for then inserting the values coming from the characterization of the devices.

The algorithm has three main sections, and it deals with a simplified behavioural description:

1. The first part deals with the creation of the maze, the definition of the constants (such as the initial position, the rate of maze modulation in time, the maximum number of trials and epochs) and the initialization of the variables (such as the current calculation, the current integration, and the cardinal points).
2. The second part regards the loading of the experimental data. The agent is connected to the cardinal points by $SiO_x$ RRAM synapses that are potentiated (set) or depressed (reset) by Hebbian learning. Each cardinal point is represented by a spiking neuron that integrates the current coming from the synapses. The first neuron that reaches the threshold voltage (defined by a specific $SiO_x$-RRAM state device) induces a spike that fully sets the corresponding excitatory synapses; at the same time, the state device is partially set, thus causing a gradual increase of the neuronal threshold. If the agent founds a wall (penalty) or the final goal (reward) the corresponding state positions are remembered from trial to trial in order to boost the learning by reinforcement.
3. The third section of the code is referred to the calculation of the movements. Once the initial position is set, the agent can go

towards eight possible directions, which are mapped by the corresponding RRAM synaptic elements. After the definition of the synaptic elements, all the nearest positions of the maze are scanned by the code, as the integration plus fire events make the agent move. If the agent moves toward a wall, the corresponding synapse is re-programmed to HRS while the corresponding internal state is re-programmed to LRS, thus reducing the neuronal spiking excitability of that position. If the agent moves toward the reward of the maze, the corresponding internal states of the last 10 run position are reprogrammed to HRS, thus lowering the thresholds and easing the reward path. Note that, for the ordinary positions of the maze, the STDP-based Hebbian learning is taken into consideration: the integrated current (variable "Current") is the product of the read voltage ($V_{COM}$) e.g., 100 mV, times the corresponding synaptic conductance. The current is integrated (variable "Integration") and the first spiking neuron determines the direction along which the agent is moving. This direction defines a new initial position from which the previous calculations are repeated.

### Pseudocode of the reinforcement learning algorithm

**Algorithm 1.** Behavioral code for reinforcement learning

```
1:   define MAZE,
2:   define INITIAL POSITION (I,J),
3:   define CURRENT,
4:   define INTEGRATION,
5:   define CARDINAL POSITIONS,
6:   define CARDINAL POINTS = 8,
7:   define Other auxiliary variables
8:   Load MONTE CARLO DATA: R_LRS, R_HRS
9:   for Trial ≤ Trial_MAX do
10:      FOUND == 0
11:      procedure SYNAPSES MATRIX(R_LRS, R_HRS)
12:          for i ≤ i_MAX do
13:              for j ≤ j_MAX do
14:                  if (i,j) ≠ save CARDINAL POSITIONS then
15:                      G_SYN(i,j) ← random 1/R_LRS or 1/R_HRS
16:          return G_SYN;
17:      procedure STATES MATRIX(R_LRS at low I_C or R_HRS at low V_STOP)
18:          for i ≤ i_MAX do
19:              for j ≤ j_MAX do
20:                  if (i,j) ≠ save CARDINAL POSITIONS then
21:                      G_STA(i,j) ← random(1/R_LRS from low I_C)
22:          return G_STA;
23:      for Epoch ≤ Epoch_MAX do
24:          Calculate CARDINAL positions from INITIAL position
25:          if CARDINAL positions == MAZE(WALL) then
26:              G_STA(CARDINAL positions(i,:)) ← 1/R_LRS
27:              G_SYN(CARDINAL positions(i,:)) ← 1/R_HRS
28:              save CARDINAL positions(WALL)
29:          Calculate CURRENT from G_SYN
30:          Calculate INTEGRATION from CURRENT, CARDINAL positions
31:          for i ≤ CARDINAL POINTS do
32:              if INTEGRATION(i) ≥ G_STA(CARDINAL positions(i,:)) then
33:                  save CARDINAL positions(i,:)
34:                  if CARDINAL positions(i,:) == MAZE(REWARD) then
35:                      FOUND = 1
36:                      break
37:                  else
38:                      INITIAL position ← CARDINAL positions(i,:)
39:                      G_SYN(CARDINAL positions(i,:)) ← 1/R_LRS
40:                      G_STA(CARDINAL positions(i,:)) ← 1/R_LRS incremental I_C
41:          if FOUND == 1 then
42:              10 latest positions ← save CARDINAL POSITIONS
43:              G_SYN, 10 latest positions ← 1/R_LRS at highest I_C;
44:              G_STA, 10 latest positions ← 1/R_HRS at highest V_STOP;
45:              break
46:          Maintain data related to WALL and REWARD (also 10 latest positions)
47:          save new CARDINAL positions
```

**Table 1 | Accuracy of the network for finding the escape path starting from a fixed initial point**

| Dependence of the accuracy on the binary programming of the devices | | | | |
|---|---|---|---|---|
| V$_{STOP}$ (V) I$_C$ (mA) | −1 V | −1.2 V | −1.4 V | −1.6 V |
| 50 μA | 69.4% | 89.8% | 96.6% | 98.1% |
| 70 μA | 73.1% | 91.0% | 97.7% | 98.7% |
| 100 μA | 75.3% | 93.4% | 99.0% | 99.3% |
| 130 μA | 77.7% | 94.6% | 99.5% | 99.8% |
| 160 μA | 78.6% | 95.0% | 99.7% | 99.8% |

Considering ideal multilevel synaptic elements, the better the programming phase of the binary synapses, the higher the accuracy (number of successes per fixed number of trials, 100).

Note that two key procedures are defined in the pseudocode, namely (i) "Synapses matrix" and (ii) "States matrix".

1. The synaptic matrix is initialized to high conductance values using I$_C$ = 54 μA. However, it is also possible to select the LFSR registers in order to choose random positions and to provide random HRS with |V$_{STOP}$| = 1.1 V. Thus, the final distribution for the synaptic matrix is bimodal, as reported in Fig. 2b. Those synapses that have undergone a penalty or received a reward keep the conductance obtained during the previous trials. This initialization procedure is made for assuring faster response of the explorative algorithm, but further initialization approaches would be anyway acceptable while keeping the same accuracy.
2. The States matrix is generally initialized at lower conductance values with |V$_{STOP}$| = 1.1 V (plus local high conductance values of random positions selected by the LFSR registers and programmed at I$_C$ = 54 μA). Note that the state devices are gradually increased in conductance as the devices are gradually set at higher I$_C$. All the resistance values are taken from the distributions of experimental data.

To perform an analysis regarding the time and the accuracy at varying I$_C$ and V$_{STOP}$, the combination of different devices for the "Synapses matrix" was studied. Considering ideal multilevel synaptic definition for the internal states, the efficiency of the algorithm improves when the synaptic devices have low-dispersed LRS and HRS, as reported in Table 1.

## Data availability
The experimental data generated in this study have been deposited in this GitHub repository: https://github.com/Bianchi27/A-self-adaptive-hardware-with-resistive-switching-synapses-for-experience-based-neurocomputing.git. Further data that support the findings of this study are available from the corresponding author upon request.

## Code availability
The computer codes are available accessing this GitHub repository: https://github.com/Bianchi27/A-self-adaptive-hardware-with-resistive-switching-synapses-for-experience-based-neurocomputing.git.

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

## Acknowledgements

This article has received fundings from the European Union's Horizon 2020 research and innovation program (grant agreement No. 899559).

## Author contributions

S.B. and I.M.M. have contributed equally to the conceptual planning, the design and implementation of the system, the extraction and the interpretation of the results, the figures realization and the text writing. E.C., A.B. and G.P. have contributed to the development of the experimental setup and to the experimental measurements. A.R. has provided the memory devices and led the project with respect to the memory arrays. G.M., J.F.N., and F.A. have contributed to the physical realization of the memory arrays. D.I. has supervised the planning and the design of this project.

## Competing interests

The authors declare no competing interests.
