## [Peer Review File · Nature Communications]

A self-adaptive hardware with resistive switching synapses for experience-based neurocomputingREVIEWER COMMENTS

Reviewer #1 (Remarks to the Author):

The manuscript reports a recurrent neural network using a digital system on chip integrated with in-memory resistive switching random-access memory array for experience based neurocomputing. Authors confirmed the feasibility of concept by experimentally demonstrate the autonomous exploration of a maze while shows the simulations for the Mars rover navigation. The proposed hardware is capable to process Hebbian learning and homeostatic plasticity. However, prior to the publication, the author needs to address and clarify the following issues. Other than below points, this manuscript clearly demonstrated the concept and confirmed the feasibility of their process.

1. Authors need to clearly explain about the major advantages of the Hebbian learning-based approach compared to conventional reinforcement algorithms for the maze problem such as Q-learning and multi-armed bandit problems. Can the authors compare the maze-solving performance based on various approaches to support the author's proposed bio-inspired hardware can outperform the performance or is comparable to these widely used software algorithms?

2. The author needs to specify which learning methods of the Hebbian learning has been employed in their neurocomputing system, such as long- or short-term potentiation, STDP, SRDP, or LIF. Also, the corresponding Hebbian responses of the memristor cells should also be experimentally demonstrated.

3. The author implemented a hardware-based RNN reinforcement learning (RL) platform. However, it is not clear that how the conventional software RNN and RL algorithms have been implemented using the RRAM crossbar. Schematic representation of the software RNN/RL in conjunction with the hardware implementation could highlight the hardware interpretation more clearly, similarly in Fig. 1a,b in ref1. ref1) Milano, G., Pedretti, G., Montano, K. et al. In materia reservoir computing with a fully memristive architecture based on self-organizing nanowire networks. Nat. Mater. (2021).
<https://doi.org/10.1038/s41563-021-01099-9>

4. The working principle and the system structure are not easily understandable in 'The event-based architecture' section and Fig. 2. The explanation of the working principle is dispersed and repeated in the manuscript too broadly, from Result to Discussion sections. The related questions are:
-What is the total dimension (col×row) of the fabricated RRAM system and how is it divided into the building blocks such as the 'Synapse' and 'State'?
-Fig. 2a, 2b are not sufficient to demonstrate the working principle. Schematic flow charts for specific cases (e.g, wall at north and no wall at east, etc) could convey the RRAM-based reinforcement learning implementation, not as just waveforms similarly in Fig. 3. The flow charts might include 1) the initial conductance of the 32×32 in 'State' and 'Synapse', 2) how the 'penalty' and 'reward' are performed via Hebbian learning, and 3) how the conductance is either depressed or potentiated based on the maze (or Mars environment) geometry, etc.
-In 'State' building block in Fig. 2b, the indexing of the bottom-left RRAM cell might be typo? (i-1, j+1) →

(i-1, j-1)?

-Comparing the 9 index cells in each 'Synapse' and 'State' in Fig. 2b, why is the indexing different between 'State' and 'Synapse'?

-How many CMOS neurons are included in the system? If the total number of the CMOS neurons is just 8 (stated in line 173), how is each CMOS neuron interconnected/switched to each column of 'Synapse' and 'State'? For example, how is the one of the CMOS neurons (out of 8) is interconnected with ith col in 'Synapse' and (j-1)th col in 'State'?

I suggest that the author addresses these issues and revise the manuscript accordingly prior to the publication.

Reviewer #2 (Remarks to the Author):

The authors implemented a spiking neural network based on RRAM and FPGA, where RRAM stores the synaptic weights and internal states. The programming of the RRAM is based on an algorithm implemented in the FPGA. The neural network learns to navigate in a dynamically changing maze through Hebbian and reinforcement learning.

The documentation of the methods and results is easy to follow and the engineering work seems very solid. The learning algorithm is relatively simple but effective for the task described.

Implementing an algorithm on a novel hardware and making it work in an application requires much engineering effort. Unfortunately, IMHO, this does not fit to a top journal like Nature Communications, since there is not enough scientific novelty.

Reviewer #3 (Remarks to the Author):

The authors reported a new kind of recurrent neural network using In-memory SiOx RRAM arrays and neurons with Hebbian learning and homeostatic plasticity. Also, they demonstrated 1) the autonomous exploration of a continually evolving maze by hardware, and 2) the Mars rover navigation by software simulation as an example for applying this system to real world problem. All the contents including supplementary information and demonstration movies are well-organized and super clear.

Remarkable points in this manuscript is that SiOx-based RRAM arrays can be also used for their proposed recurrent neural networks. They previously proposed similar system using phase change memory in ISCAS2020 [A bio-inspired recurrent neural network with self-adaptive neurons and PCM synapses for solving reinforcement learning tasks], also in Frontiers in Neuroscience in 2021 [A Brain-

Inspired Homeostatic Neuron Based on Phase-Change Memories for Efficient Neuromorphic Computing]. In this manuscript, they replaced the memristor array from phase change memory to RRAM. They added software demonstration of the Mars rover navigation as a real world problem example in this manuscript, but it might be technically obvious that it should work fine even for the case since they basically replaced the penalty of reinforcement learning from "wall" in maze to "slope" in this problem. Since they utilized most of the concept from previous works, the replacement from phase change memory to RRAM might sound less technical progress. Also, as far as I investigated, the characteristics of the used RRAM looks general compared to the others. From power, performance, and area point of view, it would be better to include quantitative comparison analysis with conventional computing architecture to make the manuscript more convincing one.

I think one weakness to be solved for the proposed architecture would be scalability. As they also mentioned in line-293-297, the size of array needs to be increased as the size of grids are increased. This is due to straightforward one-to-one mapping between the maze grid and the array element. So, if I say it negatively, the proposed architecture seems to be simply recording the easiness of the routing path grid-by-grid in RRAM synapse and state.

Minor comments:

Note for Figure 1 and Figure 2 seems to be reversed in supplementary information document.

Reply to the reviewers' comments for the paper entitled "A SELF-ADAPTIVE HARDWARE WITH RESISTIVE SWITCHING SYNAPSES FOR EXPERIENCE-BASED NEUROCOMPUTING" submitted to Nature Communications.

Dear Reviewers,

Thank you very much for your valuable comments on our manuscript entitled "A self-adaptive hardware with resistive switching synapses for experience-based neurocomputing". All the feedbacks and comments have been very helpful for the revision and refining of the manuscript, as well as for improving the quality of our current research activity. We have provided additional work in order to cover all the open points and to clarify any source of doubt, misunderstanding and error arisen during the first review cycle. The revised parts are highlighted in the new version of the work while the point-by-point responses to the reviewers' comments are reported in the following of this document.

ANSWERS TO THE REVIEWERS' COMMENTS

Reviewer #1 (Remarks to the Author):

The manuscript reports a recurrent neural network using a digital system on chip integrated with in-memory resistive switching random-access memory array for experience based neurocomputing. Authors confirmed the feasibility of concept by experimentally demonstrate the autonomous exploration of a maze while shows the simulations for the Mars rover navigation. The proposed hardware is capable to process Hebbian learning and homeostatic plasticity. However, prior to the publication, the author needs to address and clarify the following issues. Other than below points, this manuscript clearly demonstrated the concept and confirmed the feasibility of their process.

We thank the reviewer for the useful suggestions that have helped us to improve the quality of this manuscript. We have taken into consideration all the comments provided during this review process. We provide now a version of the manuscript which addresses all the reviewer's observations and introduces further theoretical studies and experimental results.

In the following, we provide a point-by-point answer to each one of the comments. We also provide precise references to the points in which the main manuscript and the supplementary material have been accordingly updated.

1. Authors need to clearly explain about the major advantages of the Hebbian learning-based approach compared to conventional reinforcement algorithms for the maze problem such as Q-learning and multi-armed bandit problems. Can the authors compare the maze-solving performance based on various approaches to support the author's proposed bio-inspired hardware can outperform the performance or is comparable to these widely used software algorithms?

We thank the reviewer for this comment. We agree with the reviewer that the discussion about the major advantages of bio-inspired approach to reinforcement learning with respect to conventional approaches is a key topic to discuss in detail.

We have now improved our work in order to clarify this aspect, proposing a focused comparison with deep learning approaches and multi-armed bandit problems.

Furthermore, we provide now in the manuscript a new theoretical framework for the bio-inspired approach starting from the fundamental equations of reinforcement learning. This analytical framework highlights the benefits of bio-inspired hardware with respect to the software-based state-of-the-art techniques in terms of accuracy, resilience and power consumption.

In the following, in order to accomplish with the reviewer observations in terms of comparison with the state-of-the-art, we discuss in detail the following topics:

- I. The source of inspiration which has led us to deepen the bio-inspired approach capabilities for reinforcement learning tasks.

- II. The theoretical background of reinforcement learning.
- III. The comparison between bio-inspired and deep learning approaches.
- IV. The novel theoretical framework, based on the equations of reinforcement learning, which highlights the benefits of the neuromorphic approach with respect to the standard techniques.

Note that, in order to ease the reading of this answer, we divide into paragraphs each one of the aforementioned points. Most of the references are taken from the manuscript reference list, while others are added in the final part of this answer.

Source of inspiration

The dichotomy between deep learning and bio-inspired approaches is one of the most relevant discussions in the field of artificial intelligence since the late 80s. In particular, the deep learning approaches offer significant accuracy and stability in complex tasks, for instance showing superhuman abilities in playing games [20]. On the other hand, the bio-inspired approach has been investigated for mainly three reasons, namely: (i) the resilience capability of organisms for enabling adaptation to multiple situations; (ii) the low-power operation of the brain; (iii) the fast, unsupervised learning procedure. In particular, these two last points are the major advantages with respect to deep learning approaches, which generally need slow and energetic-hungry training procedures in order to achieve high accuracy and efficiency [21]. Furthermore, artificial algorithm shows great stability but lacks resilience to adapt to continually changing situations. This dichotomy was summed up since the early stages of artificial intelligence history with the famous sentence “stability-plasticity dilemma” [40].

This dichotomy is also what drives our study. By proposing a brain-inspired neural network hardware, we would like, first of all, to demonstrate the possibility of joining state-of-the-art accuracy and stability (matching Q-learning performances for autonomous navigation) but providing, at the same time, low-power operation and resilient capabilities. We have now clarified this point in Section “Introduction” of the manuscript.

In order to do so, as correctly pointed out by the reviewer, we must provide a more exhaustive comparison with respect to the state-of-the-art concerning deep learning techniques for reinforcement learning. We are going to analyse this topic in the next sections of this answer.

Theory of reinforcement learning

Generally, Reinforcement learning solves a particular kind of problem where decision making is sequential, and the goal is long-term, such as game playing, robotics, resource management, and autonomous navigation.

Reinforcement learning for autonomous navigation is a broad scientific field. The Bellman equation, developed for solving problems that can be represented by sub-problems (such as repetitive movements in an environment), focuses on the relevance of the memory of the past events experienced by the agent [1X]. In formula:

$$V(s) = \max_a (R(s, a) + \alpha V(s')),$$

Where $V(s)$ represents the maximum “value” of the agent of staying in a particular state “s” of the environment to get to the final objective as a consequence of an action “a”. Note that $V(s)$ is the maximum value of an expression containing the reward function $R(s,a)$, i.e. the reward (or the penalty) after making a decision, and the value $V(s')$, i.e. the value (weighted by a fitting factor α) of the new state that the agent is going to experience.

This equation works perfectly if the environment is known and if it can be modelled by a mathematical function. However, the world is stochastic, and the “R” and “V” functions generally show probability tendencies rather than pure analytical evolutions, thus arising the need of more elaborated models for describing the autonomous navigation of an agent.

To this aim, the Markov Decision Process introduces a probability function $P(s',a,s)$ which weights the value $V(s)$ as a function of the probability of the quality of a certain position s for moving towards another state s' [5,6]. In equation:

$$V(s) = \max_a (R(s, a) + \alpha \sum_{s'} P(s, a, s') V(s')),$$

In other words, The Markov decision process provides a mathematical framework for the decision-making procedure in situations where the outcomes are partly stochastic. As a consequence, the Q-learning technique for reinforcement learning poses the idea of assessing the quality of an action to a state rather than representing the value of the state itself. In formula:

$$Q(s) = R(s, a) + \alpha \sum_{s'} P(s, a, s') \max_{a'} Q(s', a'),$$

Where $\max_{a'} Q(s', a')$ is the maximum of all the possible $V(s')$. It means that a quality of a certain position $Q(s)$ is also dependent on the quality of the nearest states s' .

The last step to cover is related to the online calculation of the Q-values with respect to the modifications of the environment over time. The temporal difference (TD) of the quality function is what gives the name to this approach to reinforcement learning, in equation:

$$TD_t(a, s) = \beta (Q_t(a, s) - Q_{t-1}(a, s)),$$

Where β is the inverse of the learning rate of the current Q value with respect to the previous Q value ($Q_{t-1}(a, s)$).

These models can map the behaviour of the agent developing a decision-based policy by exploiting the interaction with the environment and taking a decision whose effect, in turn, constitutes part of the experience of the agent [9]. We have now reported the standard framework of reinforcement learning in section “Introduction” of the manuscript.

In the next section of this answer, we benchmark the neuromorphic approach with respect to the standard approaches.

Comparison of standard and neuromorphic approaches for reinforcement learning

Deep learning techniques using standard Von Neumann processors enable accurate autonomous navigation but require great power consumption and long time for making training algorithms

effective. One of the objectives of this manuscript, on the other hand, is the demonstration that it is possible to keep the high accuracy providing, at the same time, lower power consumption and better resilience.

For instance, there have been evidences that if a problem is formulated as a partially observable Markov decision process within the actor-critic framework for the definition of the reward and penalty policies, it is possible to define a deep learning training approach capable of optimizing complex navigation tasks, such as navigation among obstacles [64]. By means of a recurrent deterministic policy gradient algorithm (called RDPG), Wang et al. demonstrated that it is possible to reach the 99.33% success in the navigation procedure, which is comparable with the accuracy percentage we obtain in Fig. 4(f) for the navigation in the “moving maze”. However, note that the training procedure for the deep learning approach requires, in the optimized fast-RDPG, 3000 training episodes, which is a far higher number with respect to what our neuromorphic recurrent network needs thanks to the homeostatic STDP, Fig. 4(a,b). Furthermore, standard processors require data transmission back and forth the DRAM, highlighting the Von Neumann bottleneck affecting CMOS platforms [22,23,65].

Note also that deep Q-learning techniques suffer from unstable learning under some conditions of bias overestimation which requires a mutual training of a multi-layer-perceptron (MLP) network and a correct setting of the learning rate [66]. This could affect the effectiveness of the training algorithm when the system has to map the environment autonomously, requiring a network of several layers with the Adam optimizer applied for stochastic optimization [67]. All these features assure high accuracy in, at least, 1000 episodes for each trial. Contrarily, the bio-inspired learning procedure has two main advantages, namely: (i) it can learn a policy map from scratch, (ii) it relies on training-free in-situ hardware computation. This approach improves a lot the time efficiency, Fig. 4(b), and the energy consumption, Fig. 4(c), while keeping high the accuracy, Fig. 4(f), (refer also to Supplementary Fig. 9). Furthermore, the STDP does not require particular methods for stochastic optimization [36].

There are several applications in which reinforcement learning for autonomous navigation and goal disclosure is relevant. For instance, the problem of finding a lost gamma source in an irradiation room without human presence can be solved by combining convolutional neural networks (CNNs) and Q-learning [2X]. Note that in this case, even if the stimuli coming from the external world is different with respect to what we propose in Subsection “The event-based architecture”, the behavior of the agent is pretty the same and it offers a convergence similar to what we depict in Fig. 4 of the manuscript. Furthermore, note that Fathi et al. used 231,000 matrixes as dataset for each case study in order to make effective the CNN training. Each matrix is proposed to the network every single epoch and the synaptic weights of the whole network (consisting of 3 convolutional layers, 2 fully connected layers and the output) are adjusted in order to reduce the loss function. Considering the number of the synaptic weights (which are, at least, in the order of tens of thousands), the calculations and the necessary epochs for convergence (at least 15), it is clearly visible that the cost in term of time and power consumption is very high. Contrarily, the bio-inspired learning procedure has two main advantages, namely: (i) it does not require a specific creation of a dataset, since it can learn a policy map by scratch (first source of power and time saving), (ii) it relies on specific in-situ RRAM-based hardware computation of specific synaptic weights and not on a pre-compiled supervised training algorithm This approach improves a lot the time efficiency, Fig. 4(b), and the energy consumption, Fig. 4(c), while keeping high the accuracy, Fig. 4(f), and resilience, (refer also to Supplementary Fig. 9).

The Markov decision process, Q-learning, TD(λ) and deep learning are not the only topics to which the scientific community refers to for modelling and designing reinforcement learning algorithms. For instance, the multi-bandit problem is often taken as benchmark. The multi-armed bandit problem deals with an agent that attempts to make decisions as a consequence of previous experiences but, at the same time, it needs to acquire new knowledge for the next decision-making events. In order to cope with this framework, several works have proposed the use of recurrent neural networks (RNN) for enhancing the re-use of past information [68]. In the same conceptual context, there have been several attempts in building “meta-learners”, i.e. training a system on a distribution of similar tasks, in the hope of generalization capability when novel tasks are targeted [69,70]. However, even considering these meta-approaches, several CNN-based training algorithms are anyway demonstrated to be necessary to provide the system with an optimum policy map for the required navigation task, thus falling again in the power and time bottleneck described before.

All this topic has been reported in the text of the paper, more precisely in “Discussion” Section.

Theoretical framework for bio-inspired learning – Analysis of the benefits

Consider now the analytical representation of the fundamental building block of the autonomous learning, the “Q function”, in formula:

$$Q(s) = R(s, a) + \alpha \sum_{s'} P(s, a, s') \max_{a'} Q(s', a'),$$

In the following, we analyse the same equation by the RRAM-based bio-inspired point of view proposed in this work.

- The reward function $R(s,a)$ of the hardware is mapped by the homeostatic reaction described in Fig. 2(a-b) and in Fig. 3(i): the environment gives penalties and rewards which directly affect the quality “Q” of a position “s” by acting on the “state” RRAM devices. If a penalty (hitting a wall) occurs, or a reward (goal of the maze) is found, the firing neuron threshold is modulated, Fig. 3(g,h). Note that the firing neuron is the one which overcomes its threshold first, i.e. $(I_{out} - I_{th}) > 0$, in which I_{out} is the post-synaptic current, and I_{th} is the equivalent homeostatic threshold of that neuron. Thus, keeping constant the read voltage and being the neuronal current dependent on the synaptic elements, the reward function can be written as a function of the conductance values only, i.e. $R(G_{syn}, G_{state})$.
- The learning factor α is not required in the neuromorphic approach, since it is a parameter related to the deep learning procedure. However, we introduce here a generic fitting factor β for the modulation of the quality factor equation.
- The probability function “P”, which describes the probability of the quality of a certain position s for moving towards another state s', is dependent on the homeostatic-based STDP mechanism. Thus, it depends on the synaptic evolution, trial after trial, of the synaptic connection between state s and s', Fig.3(j), in formula:

$$P = P(G_{syn}(s, s'), G_{state}(s), \frac{\partial G_{syn}(s, s')}{\partial t}), [36].$$

- The value $\max_{a'} Q(s', a')$ is the maximum of all the possible values $Q(s')$ of the states “s” the agent could explore after an action a' . This feature is mapped by means of the synaptic-based movement, since the Q factor of each position is modelled over time, trail after trial, by the plastic modulation of the synaptic connections, Fig. 4(e). Thus, it is possible to rewrite this contribution as $\max_{s'} Q(G_{syn}(s', s), G_{state}(s'), \frac{\partial G_{syn}(s', s)}{\partial t})$, where the inverted “s” parameters stand for the possibility, depending on $R(s')$, of going on exploring or coming back to the previous state, Fig. 2(a).

Thus, the neuromorphic Q-learning equation, that we now call Q_N , can be re-written (for each state “s”) in this way:

$$Q_N(s) = R(G_{syn}(s, s'), G_{state}(s)) + \beta \sum_{s'} P(G_{syn}(s, s'), G_{state}(s), \frac{\partial G_{syn}(s, s')}{\partial t}) \max_{s'} Q(G_{syn}(s', s), G_{state}(s'), \frac{\partial G_{syn}(s', s)}{\partial t}).$$

Following the same procedure, it is possible to describe the TD(λ), Eq. III of the manuscript:

$$TD_{N,t}(a, s) = \rho(Q_{N,t}(G_{syn}, G_{state}) - Q_{N,t-1}(G_{syn}, G_{state})),$$

Where ρ is a fitting parameter. Note that we have re-written all the reinforcement learning equations in terms of memory-based circuitual parameters. The advantages of these equations are double fold since: (i) time and power consuming data transfers between CPU and DRAM are avoided; (ii) the system can rely on pure synaptic adaptation in order to carry out accurate computation. By a theoretical point of view, this is the most relevant achievement introduced by this work since it highlights the intrinsic benefit of bio-inspired in situ-computation with respect to the software-based state-of-the-art.

All this discussion has been reported in Subsection “Theoretical modelling for in-memory reinforcement learning”.

References proposed only in this document but not in the manuscript

[1X] Kirk, D.E. Optimal Control Theory (An Introduction). Dover Publications, INC. Mineola, New York (2004).

[2X] Fathi A., Masoudi S.F. Combining CNN and Q-learning for increasing the accuracy of lost gamma source finding. Sci Rep. **12**, 2644 (2022).

2. The author needs to specify which learning methods of the Hebbian learning has been employed in their neurocomputing system, such as long- or short-term potentiation, STDP, SRDP, or LIF. Also, the corresponding Hebbian responses of the memristor cells should also be experimentally demonstrated.

We thank the reviewer for arising this relevant point. Previously, we had not specified the type of Hebbian learning mechanism in the main text of the manuscript.

As the reviewer has suggested, we have now corrected this missing point which had previously led to a not complete clarity of the whole architecture. The neuro-plausible model we have taken into consideration is the spike-timing-dependent plasticity (STDP) with homeostatic integrate & fire neurons, i.e., neurons capable of fire modulation as a function of their fire rate (this behaviour is also known as synaptic scaling). We have provided a detailed explanation of the protocol in Fig. 2 and Fig. 3, also adding a brief explanation over the relevant role provided by the noise activity typical of the STDP protocol (refer to Subsection “The event-based architecture” and to “In-memory computing for autonomous navigation”) [36]. Note that Fig. 2(a) also provides a high-level representation of the leading ideas underling the bio-inspired sources of inspiration of this work.

3. The author implemented a hardware-based RNN reinforcement learning (RL) platform. However, it is not clear that how the conventional software RNN and RL algorithms have been implemented using the RRAM crossbar. Schematic representation of the software RNN/RL in conjunction with the hardware implementation could highlight the hardware interpretation more clearly, similarly in Fig. 1a,b in ref1.

ref1) Milano, G., Pedretti, G., Montano, K. et al. In materia reservoir computing with a fully memristive architecture based on self-organizing nanowire networks. *Nat. Mater.* (2021). <https://doi.org/10.1038/s41563-021-01099-9>

We thank the reviewer for this useful comment.

We have now provided a high-level representation of the leading ideas of our work, Fig. 2(a), and detailed flow-chart descriptions, Fig. 2(b), in order to clarify the main functionalities of our work. Furthermore, following the advice of the reviewer, we have provided a parallelism between the theoretical analysis and the effective hardware realization of our system, Fig. 2(c,d), as done in ref1 [27]. Note also that Fig. 2(c) and Fig. 2(d) have been properly modified in order to improve the quality of both theoretical and hardware descriptions. We have also provided further information about this topic in Subsection “The event-based architecture”.

4. The working principle and the system structure are not easily understandable in ‘The event-based architecture’ section and Fig. 2. The explanation of the working principle is dispersed and repeated in the manuscript too broadly, from Result to Discussion sections.

We thank the reviewer for the useful suggestions provided to improve the manuscript.

We agree that the previous version of Fig. 2 was not clear enough to explain the event-based architecture. We have now improved the figure providing a high-level flow chart which introduces the theoretical framework to enable a better comprehension of the overall hardware (a,b). We have also properly modified the other displayed items in Fig. 2 to improve the clarity of the presented concepts. In parallel, we have modified the corresponding caption and the text of the manuscript in order to improve the readability of the text.

Furthermore, we have now re-organized the paper. In particular, we have clarified “The event-based architecture” section and conveyed the information related to the signal-level description of

the hardware in the other dedicated Subsections of “Results” (e.g., “In-memory computing for autonomous navigation”).

Following the same approach, we have gathered the most relevant features of the hardware in another subsection (“Exploration, optimization and recall”) while leaving the most relevant outcomes with respect to the state-of-the art (especially concerning the deep learning approach) in section “Discussion”. Moreover, in order to better contextualize the achieved results, we have now introduced in parallel a novel theoretical framework to analyse the benefits of bio-inspired in-situ computation.

In the next answers, we focus point by point to each suggestion arisen by the reviewer for improving the quality and clarity of our manuscript.

The related questions are:

-What is the total dimension (colxrow) of the fabricated RRAM system and how is it divided into the building blocks such as the ‘Synapse’ and ‘State’?

We thank the reviewer for highlighting this point that we had not discussed in the first version of the manuscript.

We have reported in section “methods” and in the supplementary material (more precisely in Subsections “Discussion over the electrical characterization of the devices”, and “Architecture of the integrated hardware”) the most relevant steps for the development of the arrays. However, note that we cannot discuss all the details of the RRAM devices and of the peripheral circuitry implemented in the arrays, due to non-disclosure agreements.

Each one of the fabricated chips contains several arrays of RRAM devices with different technological features. As you can see in the first image attached to this answer, every state array is separated from the synaptic array and each of them has a dedicated direct-memory-access (DMA) circuit addressable by means of the pads reported in the figure, also proposed as Supplementary Fig. 4. For simplicity, given the high number of available devices, the state and synaptic arrays have been addressed using different arrays with same dimensions, but this choice is arbitrary: for instance, changing the conceptual architecture, it could have been also possible to use the same array (i.e. sharing the same DMA).

Furthermore, note that the array is built following a general-purpose setup and offers both fully connected architectures (for the implementation of matrix-vector-multiplication) as well as stand-alone addressable devices for single-ended functionality and electrical characterization.

As you can see in the first image here attached, there have been several arrays integrated but, in order to ease the bonding on the pads, only some of them have been effectively used during the experiments.

In particular, each one of the chips you can see in the second figure attached to this answer (Supplementary Fig. 3(b)) hosts a total of 96 kb bonded devices. Of course, given the high number of available arrays, further devices could be easily accessed by providing more bonding wires to the package of the integrated circuits (until 1 Mb). Note also that the maximum dimension of the array that can be accessed using only one DMA is around 16kb (128x128), while the smallest fully

connected array addressable by the hosting board and the experimental setup shown in Supplementary Fig. 3(a) has a dimension of 8x8.

Finally, note that the high reconfigurability of these arrays gives the possibility of choosing different top-level architectures for taking advantage of different features of the devices, depending on the application and target. We have discussed this topic, along with the scalability of the presented hardware, in the Subsection “Reconfigurability of the hardware”. In the same subsection we also cite further applications which fit very well the architecture proposed in this work (e.g., exploration of archaeological sites or mines to reduce hazards for humans).

-Fig. 2a, 2b are not sufficient to demonstrate the working principle. Schematic flow charts for specific cases (e.g, wall at north and no wall at east, etc) could convey the RRAM-based reinforcement learning implementation, not as just waveforms similarly in Fig. 3. The flow charts might include 1) the initial conductance of the 32x32 in ‘State’ and ‘Synapse’, 2) how the ‘penalty’ and ‘reward’ are performed via Hebbian learning, and 3) how the conductance is either depressed or potentiated based on the maze (or Mars environment) geometry, etc.

We thank the reviewer for arising this point. We have now improved the clarity of the manuscript following this advice.

In particular, we have introduced a new Fig. 2 proposing a detailed flow chart with dedicated explanations of the working principle of the hardware. We have also better addressed the Hebbian learning topic for the movement of the agent and the synaptic scaling for the memory mechanism exploiting the RRAM-based homeostasis. Furthermore, we have also inserted in Fig. 2(b) two colour maps for describing the initial conditions of the “synaptic” and “state” arrays.

In order to better contextualize all the topics proposed in this work, we propose now in Fig.5 a detailed explanation for the Mars Rover Navigation with respect to other relevant topics such as the hardware scalability and reconfigurability (refer also to Subsection “Reconfigurability of the hardware” in section “Results”).

-In ‘State’ building block in Fig. 2b, the indexing of the bottom-left RRAM cell might be typo? $(i-1, j+1) \rightarrow (i-1, j-1)$?

We thank the reviewer for highlighting this typo. The figure indexing is now correct.

-Comparing the 9 index cells in each ‘Synapse’ and ‘State’ in Fig. 2b, why is the indexing different between ‘State’ and ‘Synapse’?

We thank the reviewer for the comment. In particular, keeping into consideration a generic “state” position (i,j) , the “synaptic” devices connecting that position to the nearest neurons are accessed by providing the address of the current position (i,j) to each one of the available synaptic arrays. Note that the eight synaptic arrays represent the eight cardinal directions. On the other hand, the “state” devices map the physical positions of the nearest neurons with respect to the current position. In order to avoid misunderstandings with this referencing convention chosen to manage the addresses of the memory devices, we have now better clarified the strategy in the new version of Fig. 2, in the corresponding caption, and in Subsection “The event-based architecture”.

-How many CMOS neurons are included in the system? If the total number of the CMOS neurons is just 8 (stated in line 173), how is each CMOS neuron interconnected/switched to each column of ‘Synapse’ and ‘State’? For example, how is the one of the CMOS neurons (out of 8) is interconnected with i th col in ‘Synapse’ and $(j-1)$ th col in ‘State’?

We thank the reviewer for the comment. The connections between the memory devices and the neurons are managed as a function of the current state of the agent by using the direct-memory-access (DMA) peripheral circuitry. Thus, at every movement, as now better highlighted in Fig. 2(d), not only the address of the neuron itself is changed, but also the commands of the peripheral circuitry.

This is why we decided to keep dedicated and precise addresses of synaptic and state RRAM elements, with the former related to the connections among neurons, and the latter related to neuronal features. At top level, all this information is managed by the master of the system, the SoC Xilinx 7000, by dedicated hardware (VHDL) codes which also manage the addresses to send to the pads of the RRAM arrays by means of the DMAs, Supplementary Fig. 4. Some general-purpose decoders are also directly implemented in the peripheral circuitry in order to make possible all the interconnections of the memory arrays, while keeping low the number of CMOS

neurons, in this case 8. Due to non-disclosure agreements, we cannot provide further information about the peripheral circuitry of the RRAM arrays.

We have now clarified the figures and the text of the manuscript in order to ease the comprehension of the hardware setup.

I suggest that the author addresses these issues and revise the manuscript accordingly prior to the publication.

We would like to thank again the reviewer for all the useful observations provided during this review process: every discussion, suggestion and advice has been very useful for improving the quality, clarity and readability of the manuscript.

ANSWERS TO THE REVIEWERS' COMMENTS

Reviewer #2 (Remarks to the Author):

The authors implemented a spiking neural network based on RRAM and FPGA, where RRAM stores the synaptic weights and internal states. The programming of the RRAM is based on an algorithm implemented in the FPGA. The neural network learns to navigate in a dynamically changing maze through Hebbian and reinforcement learning.

We thank the reviewer for the comment. In the following, we provide a complete point by point answer to the arisen observations and we describe the improvements we have introduced in our work following the reviewers' and editor's suggestions.

Note that the manuscript provides now several additional points and improved quality along with the effectiveness of the experimental results. We also introduce a novel theoretical framework which conceptually analyses the benefits of the RRAM-based in-situ computation.

The documentation of the methods and results is easy to follow and the engineering work seems very solid. The learning algorithm is relatively simple but effective for the task described.

We thank the reviewer for arising the quality, the clarity and the effectiveness of the results of our work with respect to the bio-inspired recurrent neural network used for autonomous navigation.

Implementing an algorithm on a novel hardware and making it work in an application requires much engineering effort. Unfortunately, IMHO, this does not fit to a top journal like Nature Communications, since there is not enough scientific novelty.

We thank the reviewer for the comment.

We believe that the work we propose in this manuscript is not only an engineering effort, since it shows relevant conceptual advancements in the field of artificial intelligence and autonomous navigation (e.g., in the new version of the manuscript we demonstrate both theoretically and experimentally a potential solution for the historical "stability-plasticity dilemma", [40]).

In the revised manuscript, we propose a novel mathematical framework which highlights the benefits introduced by the bio-inspired approach for autonomous navigation. This theoretical discussion is one of the most relevant scientific novelties introduced in this manuscript, since it justifies the relevance of the experimental outcomes (with respect to the state of the art) in terms of accuracy, robustness, resilience and power consumption (refer to Section "Discussion").

Since we have strongly improved our work after the first review cycle, we report here below the most relevant outcomes for which we believe this work is now worth of publication in Nature Communications.

1. In this point, we analyse the first novelty proposed in this work, i.e. a new theoretical framework for in-memory reinforcement learning. In order to do so, consider the most relevant equations of reinforcement learning theory reported in Section “Introduction”. In particular, the Q function, which is the building block of most of the reinforcement learning models, can be described by the following formula:

$$Q(s) = R(s, a) + \alpha \sum_{s'} P(s, a, s') \max_{a'} Q(s', a'),$$

We now analyse the same equation by the RRAM-based bio-inspired point of view proposed in this work.

- The reward function $R(s,a)$ of the hardware is mapped by the homeostatic reaction described in Fig. 2(a-b) and in Fig. 3(i): the environment gives penalties and rewards which directly affect the quality “Q” of a position “s” by acting on the “state” RRAM devices. If a penalty (hitting a wall) occurs, or a reward (goal of the maze) is found, the firing neuron threshold is modulated, Fig. 3(g,h). Note that the firing neuron is the one which overcomes its threshold first, i.e. $(I_{out} - I_{th}) > 0$, in which I_{out} is the post-synaptic current, and I_{th} is the equivalent homeostatic threshold of that neuron. Thus, keeping constant the read voltage and being the neuronal current dependent on the synaptic elements, the reward function can be written as a function of the conductance values only, i.e. $R(G_{syn}, G_{state})$.
- The learning factor α is not required in the neuromorphic approach, since it is a parameter related to the deep learning procedure. However, we introduce here a generic fitting factor β for the modulation of the quality factor equation.
- The probability function “P”, which describes the probability of the quality of a certain position s for moving towards another state s’, is dependent on the homeostatic-based STDP mechanism. Thus, it depends on the synaptic evolution, trial after trial, of the synaptic connection between state s and s’, Fig.3(j), in formula: $P = P(G_{syn}(s, s'), G_{state}(s), \frac{\partial G_{syn}(s, s')}{\partial t})$, [36].
- The value $\max_{a'} Q(s', a')$ is the maximum of all the possible values $Q(s')$ of the states “s” the agent could explore after an action a’. This feature is mapped by means of the synaptic-based movement, since the Q factor of each position is modelled over time, trial after trial, by the plastic modulation of the synaptic connections, Fig. 4(e). Thus, it is possible to rewrite this contribution as $\max_{s'} Q(G_{syn}(s', s), G_{state}(s'), \frac{\partial G_{syn}(s', s)}{\partial t})$, where the inverted “s” parameters stand for the possibility, depending on $R(s')$, of going on exploring or coming back to the previous state, Fig. 2(a).

Thus, the neuromorphic Q-learning equation, that we now call Q_N , can be re-written (for each state “s”) in this way:

$$Q_N(s) = R(G_{syn}(s, s'), G_{state}(s)) + \beta \sum_{s'} P(G_{syn}(s, s'), G_{state}(s), \frac{\partial G_{syn}(s, s')}{\partial t}) \max_{s'} Q(G_{syn}(s', s), G_{state}(s'), \frac{\partial G_{syn}(s', s)}{\partial t}).$$

Following the same procedure, it is possible to describe the TD(λ), Eq. III of the manuscript:

$$TD_{N,t}(a, s) = \rho(Q_{N,t}(G_{syn}, G_{state}) - Q_{N,t-1}(G_{syn}, G_{state})),$$

where ρ is a fitting parameter. Note that we have re-written all the reinforcement learning equations in terms of memory-based circuitual parameters. The advantages of these equations are double fold since: (i) time and power consuming data transfers between CPU and DRAM are avoided; (ii) the system can rely on pure synaptic adaptation in order to carry out accurate computation. By a theoretical point of view, this is the most relevant achievement introduced by this work since it highlights the intrinsic benefit of bio-inspired in situ-computation with respect to the software-based state-of-the-art.

All this discussion has been reported in Subsection “Theoretical modelling for in-memory reinforcement learning”.

2. We have now compared our hardware for autonomous navigation with the most relevant results of deep learning approaches in terms of accuracy, resilience and power consumption. This benchmark can be accessed in Section “Discussion” of the manuscript.

In all the cases, the bio-inspired recurrent neural network shows comparable results in terms of accuracy, but it also offers plasticity and resilience, thus offering a potential solution to the historical “stability-plasticity dilemma” of artificial intelligence [40]. This outcome is the second theoretical novelty we propose in this work.

3. Note that the advantage of implementing in-memory bio-inspired recurrent neural networks is double-fold since it offers stability, accuracy and resilience but, at the same time, it also provides a low-power approach for the definition of decision-making policy maps (refer to Subsection “Reconfigurability of the hardware”). In particular, the RRAM-based “synaptic” and “state” arrays provide a complete, physically implemented, policy map of the environment to explore.

Note that such learning activity can be transferred to other environments, optimizing the behavior of the agent in conditions of lower power consumption and area saving. In other words, the policy map, rather than being a pure representation of an environment, can be interpreted as a strategy to get to an objective: nevertheless, artificial intelligence refers to agents able to learn a concept rather than only a mechanism (refer again to section “Reconfigurability of the hardware” for further novel details introduced after this review process). In Fig. 5 (f-g), we now propose a practical example of this concept: an agent uses previous learnt information for optimizing the reward time in a totally new environment, also improving the power consumption and the resources in terms of area.

Thus, our hardware enables life-long learning, paving the way for resilient hardware able to understand concepts rather than applying pure algorithms. This is the third conceptual novelty introduced by this work.

4. Note also that such outcomes are also relevant in terms of scalability, since the hardware can infer abstract strategies and scale for high-dimensional problems by using the same hardware architecture.
5. Furthermore, note that in standard deep reinforcement learning procedures, all the information must be recorded in the DRAM, thus arising additional power consumption for the so called Von Neumann bottleneck [23]. The advantage of using in-memory computation by exploiting the benefits of RRAM devices is double-fold since (i) it does not require time and power consuming data transfer between CPU and DRAM, Fig. 2, and (ii) it avoids the use of complex deep learning techniques (refer to “Discussion” section in the manuscript).

Concerning this last point, note also that the RRAM devices of our hardware are affected by intrinsic stochasticity and variation. However, contrarily to supervised approaches based on backpropagation algorithms, such variabilities do not affect the overall accuracy of the hardware. In particular, note that we match the highest accuracy obtainable with state-of-the-art approaches with less power and area consumption (technical advancement of this work).

6. We demonstrated in hardware the resilience of an agent in a continually changing environment, Subsection “Exploration, optimization and recall”, Fig. 4.

This outcome is very relevant because bio-inspired homeostatic STDP avoids the need of time and power-hungry re-training procedures, as instead required by deep learning approaches (refer again to “Discussion” section) once a change in the policy map is necessary. This is both a conceptual and technical advancement which generally refers to the plastic capabilities of the bio-inspired network based on RRAM devices.

7. At the best of our knowledge, this is one of the first bio-inspired full-hardware approach to autonomous navigation using non-volatile memories. We demonstrate the effectiveness of our network not only for the dynamic maze experiment but even for more complex tasks such as the Mars Rover navigation (refer to “Mars Rover Navigation” Subsection).

Furthermore, the RRAM devices are gathered in arrays and each array has a dedicated direct-memory-access (DMA) circuit addressable by means of the pads reported in Supplementary Fig. 4. The arrays are built following a general-purpose architecture in order to be used for different objectives and they offer both fully connected arrays (for the implementation of matrix-vector-multiplication) as well as stand-alone addressable devices for single-ended functionalities and electrical characterization.

As we report in the “Reconfigurability of the hardware” Subsection, the setup is completely re-usable for different tasks, thanks to its modularity of bio-inspired algorithms. Thus, the presented event-based hardware is an engineering novelty with respect to the state-of-the-art (also refer to the “Introduction” and “Discussion” sections).

8. Note that, as we highlight in the supplementary Discussions and in Fig. 4(f) of the manuscript, the multilevel programming of the devices is not required to be precise due to the intrinsic stochasticity of bio-inspired learning mechanisms. This gives the possibility of

avoiding power-hungry equipment for assuring a correct functionality of the RRAM devices, thus highlighting a further improvement in terms of power management of the hardware with respect to the state-of-the-art of deep learning (refer to “Discussion” section).

All the previous points refer to the most relevant novelties introduced in this work, by both conceptual and experimental points of view. Due to the importance of all the outcomes proposed with respect to the state-of-the-art in terms of accuracy, resilience and power-management, we think that our work is worth of publication in this Journal.

ANSWERS TO THE REVIEWERS' COMMENTS

Reviewer #3 (Remarks to the Author):

The authors reported a new kind of recurrent neural network using In-memory SiO_x RRAM arrays and neurons with Hebbian learning and homeostatic plasticity. Also, they demonstrated 1) the autonomous exploration of a continually evolving maze by hardware, and 2) the Mars rover navigation by software simulation as an example for applying this system to real world problem. All the contents including supplementary information and demonstration movies are well-organized and super clear.

We would like to thank the reviewer for all the suggestions provided during this review process. In particular, all the provided observations have helped us to clarify some points of the manuscript, to avoid misunderstandings and to improve the overall quality of this work.

Furthermore, we would also like to thank the reviewer for highlighting the organization and clarity of this manuscript, which is a key goal for us in order to make this work understandable by a large audience of researchers coming from different scientific fields.

In the following of this review process, we provide a detailed point-by-point answer to each of the reviewer's comments. Of course, all the suggestions, observations and discussions are also reported in the new version of the manuscript which, furthermore, provides novel studies and insights with respect to before.

Remarkable points in this manuscript is that SiO_x-based RRAM arrays can be also used for their proposed recurrent neural networks. They previously proposed similar system using phase change memory in ISCAS2020 [A bio-inspired recurrent neural network with self-adaptive neurons and PCM synapses for solving reinforcement learning tasks], also in Frontiers in Neuroscience in 2021 [A Brain-Inspired Homeostatic Neuron Based on Phase-Change Memories for Efficient Neuromorphic Computing]. In this manuscript, they replaced the memristor array from phase change memory to RRAM. They added software demonstration of the Mars rover navigation as a real world problem example in this manuscript, but it might be technically obvious that it should work fine even for the case since they basically replaced the penalty of reinforcement learning from "wall" in maze to "slope" in this problem. Since they utilized most of the concept from previous works, the replacement from phase change memory to RRAM might sound less technical progress.

We thank the reviewer for the comment.

The observations provided by the reviewer in this comment are a consequence of some misunderstandings coming from the main body of the manuscript, which, in its first original version, did not propose an accurate explanation of the most relevant outcomes of this work with respect to previous studies and to the state-of-the-art.

In the following, we are going to solve this misunderstanding highlighting (i) the complete novelty of this work with respect to previous studies, and (ii) the theoretical and technical progresses with respect to the state-of-the-art in terms of accuracy, resilience, and power-saving (refer to the Section “Discussion” in the manuscript).

Note that we are going to propose here a point-by-point answer to the first part of the reviewer’s comments. After this, we will provide a subsection to describe the Mars Rover Navigation, also reported above: we will propose again the observations of the reviewer with respect to this topic in order to help the reader to follow our answer.

This work proposes a complete novel theoretical and experimental discussion over the autonomous navigation task. In the following points we start from a high-level perspective for then going into the detail of the reviewer’s observations.

1. The main novel outcome provided by this work is related to the results obtained in terms of accuracy and stability, also assuring plastic properties in terms of resilience and low-power adaptation. Such feature of the proposed hardware system proposes a potential solution to the historical “stability-plasticity dilemma” of artificial intelligence community (refer to “Introduction” and “Discussion” sections). Furthermore, in the revised manuscript we also demonstrate that the implemented hardware architecture enables full reconfigurability and scalability of the implemented system for a various range of tasks (refer to Subsection “Reconfigurability of the hardware”). All this discussion is also analysed and discussed by a theoretical point of view proposing a novel mathematical framework which highlights the benefits introduced by the bio-inspired approach for autonomous navigation. This theoretical discussion starting from the reinforcement learning theory (refer to Subsection “Theoretical modelling for in-memory reinforcement learning”) is one of the most relevant scientific novelties introduced in this manuscript, since it explains the relevance of the presented experimental outcomes with respect to the state-of-the-art in terms of accuracy, robustness, resilience and power consumption.

The research activity proposed in this work, of course, has started from some sources of inspirations used as starting points (refer to “Introduction” Section). However, as we are going to demonstrate step-by-step, all the outcomes we propose here are completely novel.

2. In the works reported by the reviewer (of which Frontiers 2021 is the extended version on Journal of the same work presented at ISCAS2020, with some different discussions, case studies and outcomes) the autonomous navigation problem was only an embryonal discussion within a different framework. In general, the environment was smaller, and the exploration task was easier since it was static in time, while here the environmental conditions (e.g., the walls of the maze or the craters and hills for the Mars rover navigation) can change from trial to trial to test the resilient capability of our novel hardware.

Furthermore, the type of recurrent neural network used in those works was theoretically and conceptually different, as well as the implemented hardware and the bio-inspired algorithm. Moreover, no theoretical framework was proposed there since the system was not ready to support the advanced capabilities in terms of resilience, accuracy and efficiency. Last but not least, the hardware was not scalable and reconfigurable, which are properties that we have now assured (refer to subsection “Reconfigurability of the hardware” or to the next

parts of this answer to the reviewer's comments). Thus, the change of the devices (the use of RRAM devices in place of the PCM devices) is not the most significant advancement but only one of the several novelties we investigated for this work.

We report in the following a detailed list of the most relevant novelties with respect to the works cited by the Reviewer:

- a. The bio-inspired recurrent neural network proposed in this work has a different architecture: the network presented in the two cited works was limited to few interconnected devices from the current position to the nearest neighbours. Thus, the hardware was very limited, and it was proposing elementary recurrent connections which could not assure the same accuracy we propose here, e.g., refer to Fig. 4.
- b. The hardware of the previous works was not scalable, and it needed M^2 neurons, in which M is the discrete size (number of steps) of a digitalized environment. On the other hand, in our new work only 8 neurons are needed, and the system is also thought to track analogue movements, with a sampling mechanism ruled by the clock only (see Subsection "Discussion of the experimental setup" in the supplementary material and Subsection "In-memory computing for autonomous navigation" in the manuscript).
- c. The choice of implementing only 8 neurons goes along with the efficient hardware architecture we fabricated for the "synaptic" and "state" RRAM arrays. This is relevant in terms of giga-operations per second and area (GOPs/mm²), which is one of the most important factors of merit for the power and area efficiency for the neural computation [30]. Since the RRAM devices are built in the back end of the line, the ideal area consumption is only the one of the selectors: thus, reducing the number of neurons is a key point to enable the scalability of the hardware itself.
- d. The tests proposed here are related to mazes in which the topological configuration, such as the walls, move in time, differently from the previous works where all the environments were static (the environment did not change in time). In order to solve this class of problems, conceptual and technological novelties are needed.
- e. In particular, in relation to the aforementioned topic, the resilient properties were not possible in previous works, since the old RNNs were not ready for the recall properties after environmental changes. In fact, in this work, the neuron can provide a modulation of the internal states once a penalty is applied to an old-rewarded position, as we state in Section "Exploration, optimization and recall" and in Figs. 3-4. Once a penalty is applied to an old-rewarded position, the internal state of that position is not abruptly modulated, but only partially, since it starts from a full reset condition: this is thanks to the physics of the RRAM devices, which is dependent on the previous resistive condition [46-48].

This is a fundamental behavior of our bio-inspired neural network, since it gives the possibility of re-writing the Q function, one of the fundamental equations of the reinforcement learning, in terms of only RRAM-based parameters, in formula:

$$Q_N(s) = R(G_{syn}(s, s'), G_{state}(s)) + \beta \sum_{s'} P(G_{syn}(s, s'), G_{state}(s), \frac{\partial G_{syn}(s, s')}{\partial t}) \max_{s'} Q(G_{syn}(s', s), G_{state}(s'), \frac{\partial G_{syn}(s', s)}{\partial t}).$$

We will describe in detail this formula in the next answers to the reviewer's comments. What is important to highlight here is that the Reward function "R", in the previous works, could not be physically mapped by memory-based parameters, thus making impossible the resilient capabilities and the recall properties we report in Fig. 4 of the manuscript. Further details about this topic are available in Subsection "Theoretical modelling for in-memory reinforcement learning".

- f. Note that in the previous works the overall results were coming from MC simulations after performing several case-studies on the small experimental setup. On the other hand, what we propose in this work is a complete theoretical and experimental framework for the real time demonstration of the optimized autonomous navigation (see also the provided experimental video).
- g. At the best of our knowledge, this is one of the first bio-inspired full-hardware approach to autonomous navigation using non-volatile memories. We demonstrate the effectiveness of our network not only for the dynamic maze experiment but even for more complex tasks such as the Mars Rover navigation (refer to "Mars Rover Navigation" Subsection).

Furthermore, the RRAM devices are gathered in arrays and each array has a dedicated direct-memory-access (DMA) circuit addressable by means of the pads reported in Supplementary Fig. 4. The arrays are built following a general-purpose architecture in order to be used for different objectives and they offer both fully connected arrays (for the implementation of matrix-vector-multiplication) as well as stand-alone addressable devices for single-ended functionality and electrical characterization.

Differently with respect to previous works, as we report in the "Reconfigurability of the hardware" Subsection, the setup is completely re-usable for different tasks, thanks to the modularity of bio-inspired algorithms.

- h. The novel bio-inspired recurrent neural network shows very high results in terms of accuracy, but it also offers plasticity and resilience, thus offering a potential solution to the historical "stability-plasticity dilemma" of artificial intelligence [40]. This outcome was not achievable in the previous works, due to the limited hardware resources and to the not complete recall capabilities of the networks.
- i. Note that the advantage of implementing in-memory bio-inspired recurrent neural networks is double-fold since it offers stability, accuracy and resilience but, at the same time, it also provides a low-power approach for the definition of decision-making policy maps (refer to Subsection "Reconfigurability of the hardware"). In

particular, the RRAM-based “synaptic” and “state” arrays provide a complete, physically implemented, policy map of the environment to explore.

Note that such learning activity, differently from previous works, can be transferred to other environments for transfer learning, optimizing the behavior of the agent. In other words, the policy map, rather than being a pure representation of an environment, can be interpreted as a strategy to get to an objective: nevertheless, artificial intelligence refers to agents able to learn a concept rather than only a mechanism (refer again to section “Reconfigurability of the hardware” for further novel details). In Fig. 5 (f-g), we now propose a practical example of this concept: an agent uses previous learnt information for optimizing the reward time in a totally new environment, also improving the power consumption and the resources in terms of area. Such property of transfer learning enables scalability of the hardware and it is completely novel with respect to the previous works.

We have now highlighted all these points throughout the manuscript.

They added software demonstration of the Mars rover navigation as a real world problem example in this manuscript, but it might be technically obvious that it should work fine even for the case since they basically replaced the penalty of reinforcement learning from "wall" in maze to "slope" in this problem.

We would like to thank the reviewer for highlighting this limitation of our work with respect to the Mars Rover Navigation: we have now provided further work about this topic in order to improve the quality of the manuscript.

The Mars Rover Navigation has been performed in several steps which involved experimental measurements. The first step regarded the definition of the environment from re-adapted HiRise images by individuating slopes and descents in order to build a proper description of the environment in the SoC; the second step regarded the definition of the experimental setup, while the third one was related to the effective experimental measurements. Finally, we took the experimental measurements for performing Monte Carlo simulations.

In the previous version of the manuscript, we used the “simulations” term for the aforementioned aspects: the modelling of the environment taken from HiRise and the final Monte Carlo simulations. We have now better explained this point in Section “Mars Rover Navigation” of the manuscript.

In order to overcome the limitations highlighted by the reviewer, we have now introduced a new concept in the manuscript that is related to the reconfigurability of the RRAM arrays. In particular, reconfigurability enables transfer learning from the current environment to further different environments in which autonomous navigation is required. This is a fundamental for two reasons:

- a) It enables continual learning throughout the agent life, which is a fundamental property of the agent in harsh situations (imagine, for instance a robot that explores caves for archaeological studies or a robot whose task is to find and deactivate bombs). As we now

explain in Fig.5 and in Subsection “Reconfigurability of the hardware”, this is obtained at low-power and it enables full scalability of the hardware.

- b) It overcomes the memory-based mapping of the environments. For instance, the agent learns how to behave not only considering physical changes (walls or slopes) but elaborating an abstract strategy.

The latter point here reported is fundamental for the Mars Rover Navigation, since the “concept learning” is what actually makes the system “more” bio-inspired. For instance, understanding how the shape of a penalty is, e.g., a circle, speeds up exploration and introduces versatility in the agent (refer again to subsection “Reconfigurability of the hardware” and to Fig. 5(f-g)).

Also, as far as I investigated, the characteristics of the used RRAM looks general compared to the others.

We would like to thank the reviewer for arising this point related to RRAM devices compared to PCM devices.

We analyse here the main differences between the two types of devices:

- 1) Both PCMs and RRAMs are built in the BEOL (backend-of-the-line), thus reducing the area consumption, which is, per each memory element, reduced to the selector only, Fig. 1. However, the switching mechanism of the devices is different: the RRAMs provide a resistive modulation by migration of ionic defects and Oxygen vacancies; on the other hand, PCMs show resistive switching mechanisms by thermo-based phase change of the GST material (alloy of Germanium, Antimony, Tellurium) [32].
- 2) Both the devices show multilevel behavior but different switching voltage conditions. The RRAM devices, in particular, need lower programming power and the multilevel resistive states are directly managed by the modulation of the gate voltage of the selector, which directly modifies the compliance current, Fig. 1.
- 3) The PCM devices suffer from resistance drift while the RRAM devices not [54].
- 4) The PCM suffer from high programming current, which slightly hinders the power performances of the hardware [32].
- 5) The power efficiency of the RRAM devices enables simpler electronical periphery with respect to PCM devices, since they can enable the design of easier DMAs (direct memory access circuitry managed by the SoC) and experimental setup (refer to the supplementary material).

We have now better highlighted this and further information about the devices in Sections “Introduction”, “Methods” and in the “Discussion over the electrical characterization of the devices” in the supplementary material.

From power, performance, and area point of view, it would be better to include quantitative comparison analysis with conventional computing architecture to make the manuscript more convincing one.

We thank the reviewer for this comment. We have now taken into consideration this suggestion and we have accordingly improved our manuscript.

We agree with the reviewer that the discussion about the major advantages of bio-inspired approach to reinforcement learning with respect to conventional approaches is a key topic to discuss in detail.

We have now improved our work in order to clarify this aspect, proposing focused comparisons with the state-of-the-art, especially with respect to deep learning approaches. We also provide a theoretical framework starting from the fundamental reinforcement learning equations for explaining the main benefits of bio-inspired learning in terms of accuracy, resilience, and power saving.

In order to provide a complete framework with respect to conventional computing architectures, we discuss in the following:

- 1) The source of inspiration which has led us to exploit the bio-inspired approach to solve decision-making navigation tasks in place of standard approaches.
- 2) The comparison of the bio-inspired approach proposed in this manuscript with respect to the state-of-the-art in terms of accuracy, resilience and power-efficiency.
- 3) Finally, in order to theoretically explain the benefits of the presented hardware, we discuss all the achieved results with respect to a novel theoretical framework we developed for in-memory reinforcement learning.

In order to ease the reading of this answer, we divide into paragraphs each of the aforementioned points.

Source of inspiration

The dichotomy between deep learning and bio-inspired approaches is one of the most relevant discussions in the field of artificial intelligence since the late 80s. In particular, the deep learning approaches offer significant accuracy and stability in complex tasks, for instance showing superhuman abilities in playing games [20]. On the other hand, the bio-inspired approach has been investigated for mainly three reasons, namely: (i) the resilience capability of organisms for enabling adaptation to multiple situations; (ii) the low-power operation of the brain; (iii) the fast, unsupervised learning procedure. In particular, these two last points are the major advantages with respect to deep learning approaches, which generally need slow and energetic-hungry training procedures in order to achieve high accuracy and efficiency [21]. Furthermore, artificial algorithm shows great stability but lacks resilience to adapt to continually changing situations. This dichotomy was summed up since the early stages of artificial intelligence history with the famous sentence “stability-plasticity dilemma” [40].

This dichotomy is also what drives our study. By proposing a brain-inspired neural network hardware, we would like, first of all, to demonstrate the possibility of joining state-of-the-art accuracy and stability (matching Q-learning performances for autonomous navigation) but providing, at the same time, low-power operation and resilient capabilities. We have now clarified this point in Section “Introduction” of the manuscript.

In order to do so, as correctly pointed out by the reviewer, we must provide a more exhaustive comparison with respect to the state-of-the-art concerning deep learning techniques for reinforcement learning. We are going to analyse this topic in the next sections of this answer.

Comparison of standard and neuromorphic approaches for reinforcement learning

Deep learning techniques using standard Von Neumann processors enable accurate autonomous navigation but require great power consumption and long time for making training algorithms effective. One of the objectives of this manuscript, on the other hand, is the demonstration that it is possible to keep the high accuracy providing, at the same time, lower power consumption and better resilience.

For instance, there have been evidences that if a problem is formulated as a partially observable Markov decision process within the actor-critic framework for the definition of the reward and penalty policies, it is possible to define a deep learning training approach capable of optimizing complex navigation tasks, such as navigation among obstacles [64]. By means of a recurrent deterministic policy gradient algorithm (called RDPG), Wang et al. demonstrated that it is possible to reach the 99.33% success in the navigation procedure, which is comparable with the accuracy percentage we obtain in Fig. 4(f) for the navigation in the “moving maze”. However, note that the training procedure for the deep learning approach requires, in the optimized fast-RDPG, 3000 training episodes, which is a far higher number with respect to what our neuromorphic recurrent network needs thanks to the homeostatic STDP, Fig. 4(a,b). Furthermore, standard processors require data transmission back and forth the DRAM, highlighting the Von Neumann bottleneck affecting CMOS platforms [22,23,65].

Note also that deep Q-learning techniques suffer from unstable learning under some conditions of bias overestimation which requires a mutual training of a multi-layer-perceptron (MLP) network and a correct setting of the learning rate [66]. This could affect the effectiveness of the training algorithm when the system has to map the environment autonomously, requiring a network of several layers with the Adam optimizer applied for stochastic optimization [67]. All these features assure high accuracy in, at least, 1000 episodes for each trial. Contrarily, the bio-inspired learning procedure has two main advantages, namely: (i) it can learn a policy map from scratch, (ii) it relies on training-free in-situ hardware computation. This approach improves a lot the time efficiency, Fig. 4(b), and the energy consumption, Fig. 4(c), while keeping high the accuracy, Fig. 4(f), (refer also to Supplementary Fig. 9). Furthermore, the STDP does not require particular methods for stochastic optimization [36].

There are several applications in which reinforcement learning for autonomous navigation and goal disclosure is relevant. For instance, the problem of finding a lost gamma source in an irradiation room without human presence can be solved by combining convolutional neural networks (CNNs) and Q-learning [2X]. Note that in this case, even if the stimuli coming from the external world is different with respect to what we propose in Subsection “The event-based architecture”, the behavior of the agent is pretty the same and it offers a convergence similar to what we depict in Fig. 4 of the manuscript. Furthermore, note that Fathi et al. used 231,000 matrixes as dataset for each case study in order to make effective the CNN training. Each matrix is proposed to the network every single epoch and the synaptic weights of the whole network (consisting of 3 convolutional layers, 2 fully connected layers and the output) are adjusted in order to reduce the loss function. Considering the number of the synaptic weights (which are, at least, in

the order of tens of thousands), the calculations and the necessary epochs for convergence (at least 15), it is clearly visible that the cost in term of time and power consumption is very high. Contrarily, the bio-inspired learning procedure has two main advantages, namely: (i) it does not require a specific creation of a dataset, since it can learn a policy map by scratch (first source of power and time saving), (ii) it relies on specific in-situ RRAM-based hardware computation of specific synaptic weights and not on a pre-compiled supervised training algorithm. This approach improves a lot the time efficiency, Fig. 4(b), and the energy consumption, Fig. 4(c), while keeping high the accuracy, Fig. 4(f), and resilience, (refer also to Supplementary Fig. 9).

The Markov decision process, Q-learning, TD(λ) and deep learning are not the only topics to which the scientific community refers to for modelling and designing reinforcement learning algorithms. For instance, the multi-bandit problem is often taken as benchmark. The multi-armed bandit problem deals with an agent that attempts to make decisions as a consequence of previous experiences but, at the same time, it needs to acquire new knowledge for the next decision-making events. In order to cope with this framework, several works have proposed the use of recurrent neural networks (RNN) for enhancing the re-use of past information [68]. In the same conceptual context, there have been several attempts in building “meta-learners”, i.e. training a system on a distribution of similar tasks, in the hope of generalization capability when novel tasks are targeted [69,70]. However, even considering these meta-approaches, several CNN-based training algorithms are anyway demonstrated to be necessary to provide the system with an optimum policy map for the required navigation task, thus falling again in the power and time bottleneck described before.

All this topic has been reported in the text of the paper, more precisely in “Discussion” Section.

Theoretical framework for bio-inspired learning – Analysis of the benefits

In Section “Introduction” we have now provided a theoretical framework for the standard approaches to reinforcement learning. In this context, consider now the analytical representation of one of the fundamental equations for the reinforcement learning, the “Q function”, in formula:

$$Q(s) = R(s, a) + \alpha \sum_{s'} P(s, a, s') \max_{a'} Q(s', a'),$$

In the following, we analyse the same equation by the memory-based bio-inspired point of view proposed in this work.

- The reward function $R(s,a)$ of the hardware is mapped by the homeostatic reaction described in Fig. 2(a-b) and in Fig. 3(i): the environment gives penalties and rewards which directly affect the quality “Q” of a position “s” by acting on the “state” RRAM devices. If a penalty (hitting a wall) occurs, or a reward (goal of the maze) is found, the firing neuron threshold is modulated, Fig. 3(g,h). Note that the firing neuron is the one which overcomes its threshold first, i.e. $(I_{out} - I_{th}) > 0$, in which I_{out} is the post-synaptic current, and I_{th} is the equivalent homeostatic threshold of that neuron. Thus, keeping constant the read voltage and being the neuronal current dependent on the synaptic elements, the reward function can be written as a function of the conductance values only, i.e. $R(G_{syn}, G_{state})$.

- The learning factor α is not required in the neuromorphic approach, since it is a parameter related to the deep learning procedure. However, we introduce here a generic fitting factor β for the modulation of the quality factor equation.
- The probability function “P”, which describes the probability of the quality of a certain position s for moving towards another state s' , is dependent on the homeostatic-based STDP mechanism. Thus, it depends on the synaptic evolution, trial after trial, of the synaptic connection between state s and s' , Fig.3(j), in formula:

$$P = P(G_{syn}(s, s'), G_{state}(s), \frac{\partial G_{syn}(s, s')}{\partial t}), [36].$$
- The value $max_{a'} Q(s', a')$ is the maximum of all the possible values $Q(s')$ of the states “s” the agent could explore after an action a' . This feature is mapped by means of the synaptic-based movement, since the Q factor of each position is modelled over time, trail after trial, by the plastic modulation of the synaptic connections, Fig. 4(e). Thus, it is possible to rewrite this contribution as $max_{s'} Q(G_{syn}(s', s), G_{state}(s'), \frac{\partial G_{syn}(s', s)}{\partial t})$, where the inverted “s” parameters stand for the possibility, depending on $R(s')$, of going on exploring or coming back to the previous state, Fig. 2(a).

Thus, the neuromorphic Q-learning equation, that we now call Q_N , can be re-written (for each state “s”) in this way:

$$Q_N(s) = R(G_{syn}(s, s'), G_{state}(s)) + \beta \sum_{s'} P(G_{syn}(s, s'), G_{state}(s), \frac{\partial G_{syn}(s, s')}{\partial t}) max_{s'} Q(G_{syn}(s', s), G_{state}(s'), \frac{\partial G_{syn}(s', s)}{\partial t}).$$

Following the same procedure, it is possible to describe the $TD(\lambda)$, Eq. III of the manuscript:

$$TD_{N,t}(a, s) = \rho(Q_{N,t}(G_{syn}, G_{state}) - Q_{N,t-1}(G_{syn}, G_{state})),$$

where ρ is a fitting parameter. Note that we have re-written all the reinforcement learning equations in terms of memory-based circuitual parameters. The advantages of these equations are double fold since: (i) time and power consuming data transfers between CPU and DRAM are avoided; (ii) the system can rely on pure synaptic adaptation in order to carry out accurate computation. By a theoretical point of view, this is the most relevant achievement introduced by this work since it highlights the intrinsic benefit of bio-inspired in situ-computation with respect to the software-based state-of-the-art.

All this discussion has been reported in Subsection “Theoretical modelling for in-memory reinforcement learning”.

References proposed only in this document but not in the manuscript

[1X] Kirk, D.E. Optimal Control Theory (An Introduction). Dover Publications, INC. Mineola, New York (2004).

[2X] Fathi A., Masoudi S.F. Combining CNN and Q-learning for increasing the accuracy of lost gamma source finding. Sci Rep. **12**, 2644 (2022).

I think one weakness to be solved for the proposed architecture would be scalability. As they also mentioned in line-293-297, the size of array needs to be increased as the size of grids are increased. This is due to straightforward one-to-one mapping between the maze grid and the array element. So, if I say it negatively, the proposed architecture seems to be simply recording the easiness of the routing path grid-by-grid in RRAM synapse and state.

We thank the reviewer for the significant suggestions provided for improving our work. We agree that scalability is a key topic to address in this paper and we are going to dedicate the following answer to this point.

In lines 293-297 of the previous manuscript, we erroneously induced the reader to a misunderstanding, since we did not explain in detail the context this observation was referred to. In particular, the memory size is linearly proportional to the “size of the environment” only for what concerns the first policy-free trials of exploration. On the other hand, such memory-position mapping can be the target for a various range of tasks. For instance, robots from “Boston Dynamics” have been used in archaeological areas to inspect hard-to-access sections of the ruins, to collect data and to alert people for safety and structural problems whenever some unexpected changes are detected [62]. Thus, in general, Robots can be useful for human beings in order to keep distance from hazards, as it was also demonstrated for mine exploration [3X].

Actually, considering the deep learning approaches for autonomous navigation, the initial trials of free exploration would require higher power and area consumption than the solution proposed in this paper, since a step-by-step record of the states is anyway necessary to create a consequent policy map. Furthermore, note also that standard neural networks are based on CMOS platforms and require time- and power- expensive training algorithms due to the continuous data transfer from the CPU to the DRAM and vice versa [23]. Thus, considering a pure memory-device mapping for accurate navigation in harsh environments, our solution appears promising with respect to the state-of-the-art. Consider also that the RRAMs devices are built in the backend-of-the-line, thus providing no area consumption further than the low size nmos selector, Fig 1. This is one of the most relevant improvements introduced by non-volatile memories with respect to standard artificial intelligence platforms and deep learning algorithms, which require more area and power for performing the same memory tasks [30].

However, besides these memory-position mapping applications, providing scalability to neural networks means that the system has to learn a concept rather than being a tracking mechanism. Now, we are going to tackle this topic in the framework of the Mars Rover navigation demonstrating that our recurrent neural network can infer abstract strategies and scale for high-dimensional problems.

In order to demonstrate the scalability of our system, we compare the exploration of a new environment using two different approaches, namely (i) the step-to-step mapping and (ii) the optimized exploration using information from previous trials within different environments. Such transfer-learning approach is based on two steps: (i) the RRAM-based development of a policy map from previous explorations; (ii) the modulation of the reward function (the penalty refers to larger regions of spaces, e.g., to circular areas). During the first step, Fig. 5(e), small sections of

the old policy map are dissected in order to record random shapes. The record is simply driven by the integrated current of all the RRAM devices included in the region of the memory under consideration, choosing only those sections which are far enough from the maximum and minimum boundaries (i.e. all LRS devices and all HRS devices). Once this procedure is iterated for different shape dimensions, the set of shapes is recorded in the FPGA and stochastically used as penalty function during the exploration of the new environment, Fig. 5(f). Such approach slightly improves the efficiency results, Fig. 5(g), and it avoids the physical device-position mapping (a single address is enough to represent a region of space).

Furthermore, note that the system is flexible because it can be easily reconfigured during operation. For instance, it would be also possible to re-write old, allocated memory arrays within the same trial in order to dynamically improve the RRAM memory efficiency over time. Such reconfigurability, which enables the use of the same hardware for different autonomous navigation tasks, goes in the direction of providing hardware-based computation while retaining the flexibility of a software approach, as it is done in reconfigurable FPGAs [63].

We thank again the reviewer for the important observation provided about scalability. We have now accordingly updated the manuscript in order to cover this aspect (refer to “Reconfigurability of the hardware”) and the supplementary material (“Discussion over the scalability of RRAM and CMOS approaches to reinforcement learning”).

References proposed only in this answer but not in the manuscript

[3X] <https://www.bostondynamics.com/resources/case-study/kidd-creek-mine>

Minor comments:

Note for Figure 1 and Figure 2 seems to be reversed in supplementary information document.

We thank the reviewer for the comment. We have now corrected the supplementary material and provided additional explanation and figures concerning the topics arisen during this review.

We would like to thank again the reviewer for all the observations provided during this review process: every suggestion and comment has been very useful and profitable for improving the overall quality of our work.

REVIEWER COMMENTS

Reviewer #1 (Remarks to the Author):

The author greatly provided additional information to address the reviewer's questions. Most of the revisions have focused on clarifying the working principle and hardware configuration of the proposed neuromorphic system. However, some parts of the reviewer's questions have still not been resolved, including the comparison between the hardware and software-based reinforcement learning. I would suggest the publication only if the authors can provide reasonable directions for the following and additional questions.

<Original Question 1>

Authors need to clearly explain about the major advantages of the Hebbian learning-based approach compared to conventional reinforcement algorithms for the maze problem such as Q-learning and multi-armed bandit problems. Can the authors compare the maze-solving performance based on various approaches to support the author's proposed bio-inspired hardware can outperform the performance or is comparable to these widely used software algorithms?

<Additional Comment>

The author additionally provided the superiority of the bio-inspired learning procedure as follows: Contrarily, the bio-inspired learning procedure has two main advantages, namely: (i) it can learn a policy map from scratch, (ii) it relies on training-free in-situ hardware computation. This approach improves a lot the time efficiency, Fig. 4(b), and the energy consumption, Fig. 4(c), while keeping high the accuracy, Fig. 4(f), (refer also to Supplementary Fig. 9). Furthermore, the STDP does not require particular methods for stochastic optimization

I assume that the "bio-inspired learning procedure" means the hardware-based neuromorphic chip, not the software-based bio-inspired programming. Despite this situation, in the case of (i), I believe that it is still possible for pure software Q-learning or general reinforcement learning to perform the maze task without any training dataset, which can also learn a policy map from scratch. Also, in the case of (ii), the improvement in the time efficiency is not clearly demonstrated (Fig. 4c~f); only the performances of the proposed hardware approach were demonstrated without the performances of the pure software Q-learning. If the calculation of the pure software situation is challenging, please refer to other studies that have reasonably provided alternative comparison standards to highlight the performance superiority of the proposed hardware system.

<Original Question 2>

The author needs to specify which learning methods of the Hebbian learning has been employed in their neurocomputing system, such as long- or short-term potentiation, STDP, SRDP, or LIF. Also, the corresponding Hebbian responses of the memristor cells should also be experimentally demonstrated.

<Additional Comment>

The author clearly provided the specific methods of Hebbian learning. In the case of STDP in this work,

the STDP behavior is emulated by a programming language. However, there are other memristor studies [1],[2] that have emulated the different STDP behavior, by sequentially applying the actual voltage pulse to the top electrode (PRE-neuron) and bottom electrode (POST-neuron) of one memristor, (similarly, middle sub-figure of Fig. 1a). Therefore, it is recommended that the middle sub-figure of Fig. 1a conveys the correct STDP implementation scheme that could be confused as the different STDP implementation.

<References>

[1] Panwar, Neeraj, Bipin Rajendran, and Udayan Ganguly. "Arbitrary spike time dependent plasticity (STDP) in memristor by analog waveform engineering." *IEEE Electron Device Letters* 38.6 (2017): 740-743.

[2] Lu, Ke, et al. "Diverse spike-timing-dependent plasticity based on multilevel HfO_x memristor for neuromorphic computing." *Applied Physics A* 124.6 (2018): 1-9.

<Original Question 3>

The author implemented a hardware-based RNN reinforcement learning (RL) platform. However, it is not clear that how the conventional software RNN and RL algorithms have been implemented using the RRAM crossbar. Schematic representation of the software RNN/RL in conjunction with the hardware implementation could highlight the hardware interpretation more clearly, similarly in Fig. 1a,b in ref1. ref1) Milano, G., Pedretti, G., Montano, K. et al. In materia reservoir computing with a fully memristive architecture based on self-organizing nanowire networks. *Nat. Mater.* (2021).

<https://doi.org/10.1038/s41563-021-01099-9>

<Additional Comment>

The revised "Theoretical modelling for in-memory reinforcement learning" Section provided a clear implementation of the reinforcement learning based on the circuit parameters. However, it is still not clear how the reinforcement parameters are computed from the circuit parameters. Can the author provide the mathematical relationship between the reinforcement and circuit parameters (e.g. $R = \text{functon}(G_{\text{syn}}, G_{\text{state}})$), or if it is challenging, provide the waveforms for representative reinforcement parameters (e.g. R, P, TDN,t) in the cases of the reward (final goal) and penalty (wall)?

<Original Question 4>

The working principle and the system structure are not easily understandable in 'The event-based architecture' section and Fig. 2. The explanation of the working principle is dispersed and repeated in the manuscript too broadly, from Result to Discussion sections.

<Additional Comment>

As author has addressed, the revised Fig. 2 improved the readability of the manuscript. To clearly demonstrate the methodology, please provide the detailed pulse geometry (width and amplitude) for the STDP, SET, and RESET processes. One further recommendation is to reconsider the main figure configuration since Fig. S8 (flow chart) is much more intuitive to understand the working principle than Fig. 2b, c.

<Original Question 5-2>

Fig. 2a, 2b are not sufficient to demonstrate the working principle. Schematic flow charts for specific cases (e.g, wall at north and no wall at east, etc) could convey the RRAM-based reinforcement learning implementation, not as just waveforms similarly in Fig. 3. The flow charts might include 1) the initial conductance of the 32×32 in 'State' and 'Synapse', 2) how the 'penalty' and 'reward' are performed via Hebbian learning, and 3) how the conductance is either depressed or potentiated based on the maze (or Mars environment) geometry, etc.

<Additional Comment>

Although author additionally provided the data for the 1) ~ 3) aspects, further methodology needs to be clearer. In Fig. 2b, how are the initial matrices defined (e.g. Gaussian distribution) and achieved (e.g. programming amplitude and width)?

<Additional Questions>

1. In the same page of Fig. 3, how are the integration signals discharged in the circuit structure?
2. In the next page of Fig. 3, what is the detailed methodology to generate the "Random spiking activity at low frequency", and why is the stochasticity provided in the system?
3. In the next page of Fig. 3, what's the specific situation of unwanted "rewarded position runs into a penalty"?

Reviewer #2 (Remarks to the Author):

The authors have made substantial improvements to address the previous comments.

Reviewer #3 (Remarks to the Author):

Thank you for the updates. I think the revised manuscript becomes much better than the initial version.

I now understand the author's point of view about novelty of this paper and difference from previous papers.

1. New questions arises about STDP. It is not clear to me about how STDP is implemented in this system. I suggest that the author describe the details in this paper because STDP learning is one of the most important key enablers for the proposed system.

For instance, the questions and unclear points to me are

A. In reference paper[36], Fig.1 describes how to implement on-chip STDP learning. Is this circuitry (Fig.1-(a)) or slightly modified circuitry also used in this paper? If not, how the spike activities on pre-

and post-neurons can be monitored in this system? How does the system detect either pre- or post-neuron fires earlier or later?

B. This is related to question-A, but how do you provide pulses on pre-synaptic neurons?

- Are the pulses provided as digital Poisson spike trains from FPGA first? Then, do the pulses trigger voltage-modulated two continuous pulses shown in Fig-1.(c) and (d) by chip-internal pulse generator?

- Are the pulses provided as simple digital High/Low signals to select and activate one row? How many pulses are provided? How much pulse duration are used per one pulse?

- There is a statement: "Random spiking activity at low frequency is also inserted in the system by means of a linear feedback shift register (LFSR)". How do you provide it to current integrator? Is it through different RRAM cell?

- In supplemental document Fig-7, two descriptions: "Reset (i,j) synapse array N" and "Set (x,y) SiOx RRAM synapse connected to the spiking neuron" can be found in "External" red boxes. This confuses me because the procedures sounds deterministic operation based on penalty or reward not using STDP. Could you clarify the relation between STDP learning and this deterministic descriptions?

2. In DISCUSSION section, the author refers to [64] as a DNN example about navigation task, then introduces that "3000" is the training episodes for the task. I think the size of navigation space in the reference paper looks much larger than the author's paper, so it is not fair comparison and probably misleads readers.

3. Regarding scalability in "Mars rover navigation" section, it's interesting approach to utilize past experience against new environment. As the authors describes, the RRAM array already stores obstacle shapes in its memory through the previous exploring. To reuse it as abstracted data is reasonable approach to improve performance in new environment.

However, there are some unclear points in this section.

- What's the difference between center and right figures in Fig-7(f)? What is the red rectangle in right figure? Why the red rectangles are missing in center figure?

- I might misunderstand the author's idea, but since the system doesn't know 1) whether the agent receives a penalty or not in new environment (there might be no obstacles in the area), 2) how many and where the agent receives a penalty, the system need to prepare same size of RRAM memory anyway before the rover starts exploring in different space on Mar.

- One address can express some size of grids, but it would be challenging to mix different grid resolution in the area, such as normal(original) resolution for rover exploring and lower resolution for penalty area.

Minor comments:

- $Q(s, a)$ and $Q(a, s)$ are mixed in equation-II and III. Is it intentionally swapped?

- It would be helpful for reviewers if there are line numbers in revised manuscript.

Reply to the reviewers' comments for the paper entitled "A SELF-ADAPTIVE HARDWARE WITH RESISTIVE SWITCHING SYNAPSES FOR EXPERIENCE-BASED NEUROCOMPUTING" submitted to Nature Communications.

Dear Reviewers,

We would like to thank you for all the feedbacks provided with respect to our manuscript entitled "A self-adaptive hardware with resistive switching synapses for experience-based neurocomputing". During this period, we have taken into consideration every suggestion and observation arisen during the review process: all the comments have been very helpful for improving the manuscript and for solving the remaining open points. We have also provided further qualitative and quantitative details in order to clarify and enhance the most relevant outcomes and novelties proposed in this work. In the following of this document, we report the point-by-point responses to the reviewers' comments, while the revised parts are highlighted in the new versions of the submitted documents.

REVIEWERS' COMMENTS

Reviewer #1 (Remarks to the Author):

The author greatly provided additional information to address the reviewer's questions. Most of the revisions have focused on clarifying the working principle and hardware configuration of the proposed neuromorphic system.

We thank the reviewer for arising the quality and completeness of the additional information related to our bio-inspired system in terms of working principle and hardware configuration. Furthermore, we would also like to thank the reviewer for all the observations arisen during these review cycles: every discussion has been fruitful to clarify the unclear points of the manuscript, to avoid misunderstandings and to improve the overall readability of this work.

However, some parts of the reviewer's questions have still not been resolved, including the comparison between the hardware and software-based reinforcement learning. I would suggest the publication only if the authors can provide reasonable directions for the following and additional questions.

We thank the reviewer for the comment.

Along with this submission, we have provided additional work in order to study qualitatively and quantitatively the comparison between the hardware-based and software-based approaches to reinforcement learning.

In particular, we now provide a comparison with **Q-learning software-based codes (based on official Python libraries for reinforcement learning)** for comparing the results of standard software-based environmental exploration with respect to our hardware solution.

As we report in the following of this document and in the new version of the manuscript, our solution overcomes the results obtained with standard software approaches. Note also that we have provided further information in the section "discussion" with respect to the state-of-the-art literature.

In order to provide a clear description of the refinement of our work, in the following of this document we provide a detailed point-by-point answer to each one of the reviewer's comments.

<Original Question 1>

Authors need to clearly explain about the major advantages of the Hebbian learning-based approach compared to conventional reinforcement algorithms for the maze problem such as Q-learning and multi-armed bandit problems. Can the authors compare the maze-solving performance based on various approaches to support the author's proposed bio-inspired hardware can outperform the performance or is comparable to these widely used software algorithms?

The author additionally provided the superiority of the bio-inspired learning procedure as follows:

Contrarily, the bio-inspired learning procedure has two main advantages, namely: (i) it can learn a policy map from scratch, (ii) it relies on training-free in-situ hardware computation. This approach improves a lot the time efficiency, Fig. 4(b), and the energy consumption, Fig. 4(c), while keeping high the accuracy, Fig. 4(f), (refer also to Supplementary Fig. 9). Furthermore, the STDP does not require particular methods for stochastic optimization

I assume that the “bio-inspired learning procedure” means the hardware-based neuromorphic chip, not the software-based bio-inspired programming.

We thank the reviewer for highlighting this point.

As the reviewer correctly points out, by means of the expression “bio-inspired learning procedure” we were referring to our “hardware neuromorphic chip”.

In the previous version of the manuscript the comparison with the software-based literature was mainly provided in section “Discussion” of the paper. We have now deeply improved section “Discussion” in order to cope with all the observations highlighted by the reviewer, to propose further comparisons with respect to the literature and to the deep-learning Python-based outcomes (especially in terms of accuracy and power consumption).

In particular, we now provide a full comparison of our bio-inspired hardware with respect to state-of-the-art software approach (designed in Python environment).

Note that the software-based approaches, as we report in Sections “Introduction” and “Discussion”, are run on conventional computers, where the processing unit is separated from the memory, thus arising the so called “Von Neumann bottleneck”. We will analyse this topic in terms of power/energy consumption under the same benchmarking conditions, thus providing a fair comparison between our computing hardware (i.e., the recurrent neural network with RRAM devices) and the state-of-the-art, Fig. 4(f,g).

We will also analyse the advantages of bio-inspired computing with respect to the conventional algorithms in terms of resilience (refer to supplementary section “Additional insights over the theoretical modelling of reinforcement learning bio-inspired networks”).

In order to avoid misunderstandings and to improve the readability of the manuscript, we have now clarified all those parts which refer to software or hardware approaches, respectively.

Despite this situation, in the case of (i), I believe that it is still possible for pure software Q-learning or general reinforcement learning to perform the maze task without any training dataset, which can also learn a policy map from scratch. Also, in the case of (ii), the improvement in the time efficiency is not clearly demonstrated (Fig. 4c~f); only the

performances of the proposed hardware approach were demonstrated without the performances of the pure software Q-learning.

We would like to thank the reviewer for arising this point that we did not properly discuss in the previous version of the manuscript. We provide now a qualitative and quantitative comparison of our results with respect to software-based reinforcement learning algorithms developed in Python environment by referring to official artificial intelligence libraries.

As the reviewer correctly points out, the policy map creation without previous knowledge of the environment is indeed possible even with software-based Q-learning, as we report now in Sections “Exploration, optimization and recall” and “Discussion”. However, we demonstrate that our bio-inspired hardware outperforms the maze task even in such situation under the same benchmarking conditions.

In the following, (i) we demonstrate that our hardware overcomes software-based solutions implementing policy map-free learning, and (ii) we propose a very detailed panorama of the most relevant works about autonomous navigation. Note that the benchmark is done with respect to the same task and condition (i.e., the same environment to explore).

Comparison with software-based reinforcement learning

In sections “Exploration, optimization and recall” and “Discussion” we report now a fair comparison with a Python-based library for reinforcement learning (you can access the homepage of the library and all the main codes clicking here ([“https://pypi.org/project/pyqlearning/”](https://pypi.org/project/pyqlearning/)) [57]).

In particular, this Python library is modular and can be used for implementing Reinforcement Learning by means of Q-Learning algorithms, Deep Q-Networks, and Multi-agent deep networks.

We link here ([“https://github.com/accel-brain/accel-brain-code/blob/master/Reinforcement-Learning/demo/search_maze_by_deep_q_network.ipynb”](https://github.com/accel-brain/accel-brain-code/blob/master/Reinforcement-Learning/demo/search_maze_by_deep_q_network.ipynb)) the main source of the codes used for the comparison. As you can see, the default maze exploration reported in this Python library is performed in a 2D environment and the shape of the maze is comparable (a bit smaller, i.e. easier) to the one presented in Fig. 4(a) of the manuscript. However, in order to be fair in the comparison, we are going to benchmark the two approaches (bio-inspired and Python-based) using the environment reported in the “GitHub codes” folder attached to this submission, Figure R1.

Figure R1. Maze used for benchmarking the proposed bio-inspired hardware with respect to the state-of-the-art approaches.

In the following we are going to compare our bio-inspired approach with respect to the Python-based solution. We will first compare the algorithms starting from a qualitative analysis about the model-free reinforcement learning (which deals with the learning activity of a policy map from scratch) for then providing a quantitative analysis in terms of accuracy of the results, power/area efficiency, reliability, and resilience for real life applications.

- *Model-free learning: comparison of bio-inspired hardware with respect to conventional approaches*
 - Model-free methods are crucial when the main goal to provide by means of the neural computation is the exploration of a completely new environment where previous information is not available [62].

As we describe in Sections “The event-based architecture”, “Exploration, optimization and recall” and in Figs. 2-4, our hardware system is based on unsupervised bio-inspired algorithms such as homeostatic “spike-timing-dependent plasticity”, STDP. This choice triggers the possibility of performing “on-line” learning, i.e., real-time learning without needing training datasets or previous labelled information, as we now better describe in Fig. 2(a) with precise reference to the effective hardware realization we propose.

This is in contrast with the so called “supervised learning”, the conventional learning procedure where training datasets are needed. The bio-inspired approach usually pays this feature in terms of lower accuracy [40] but in this work we demonstrate that, by proper development of the explorative algorithm, it is possible to boost the accuracy of the bio-inspired exploration to overcome the state-of-the-art results obtained with conventional approaches.

Furthermore, note that the conventional approaches require initial explorative trials to develop a policy from environmental information that is frequently sparse, noisy and delayed. This procedure can be tricky for deep learning-based algorithms since they rely on training algorithms, such as the backpropagation, which are not plastic and resilient [57]. In order to cope with this situation, conventional neural networks for reinforcement learning rely on specific trainings that change the loss function at each iteration: this is usually performed by using complex deep Convolutional Neural Networks (CNNs), as we report in Section “Discussion” [57,64].

In the following we analyse the quantitative figures of merit for both the reinforcement learning algorithms, conventional and bio-inspired.

- *Accuracy*
 - In order to provide a fair benchmark, we tested the accuracy for a given 2D maze exploration (we provide the file describing the maze under test in the attached repository “GitHub codes”). The standard approach of the conventional free-model

reinforcement learning was developed using the Python codes reported here: "https://github.com/accel-brain/accel-brain-code/blob/master/Reinforcement-Learning/demo/search_maze_by_deep_q_network.ipynb" [57].

Several state-of-the-art algorithms have been tested for providing a fair analysis, all giving comparable outcomes. *For simplicity and clarity of the exposition, we report in the following only the best outcome of the simulated state-of-the-art algorithm (deep Q-learning). All the approaches can be compared by accessing the provided link (for the state-of-the-art solutions) and the repository "GitHub codes" (for the behavioral description of our bio-inspired approach).*

As you can see in the plot reported here below, Figure R2, our approach overcomes the state-of-the-art free-learning algorithm in terms of accuracy. The slight better results of the bio-inspired approach with respect to the pure software-based are due to the more plastic and resilient algorithm to find the solution, which leads to a faster convergence to the optimum result (no training is needed and the system directly reacts as a function of the current experience). In fact, as we report in section "Exploration, optimization and recall", our hardware system learns on-line, and it becomes a master of the problem relying on the intrinsic plasticity for the continual adaptation to the environment.

The comparison has been now reported in Fig. 4(f), where we propose two color maps expressing the average accuracy, one for the bio-inspired approach and the other for the software-based deep Q-learning. The accuracy is calculated as a function of the number of trials per experiment and of the number of steps allowed per single trial considering the environment reported in Figure R1. This choice has been done for performing a fair comparison: our hardware is custom-based on RRAM synapses, while the software-based is run on standard processors. Thus, we decided to perform various set of experiments, each repeated 200 times, with different number of trials each (to test the recall property). This number of trials per experiment is also a function of the number of steps that the agent is allowed to do per each trial (y axis). Thus, this figure of merit comes out to be only algorithm related.

Furthermore, note also that the standard neural networks suffer from the delay between taking actions and receiving rewards, which can also be hundreds of timesteps long. In particular, this is daunting when related to the direct association between inputs and targets typical of supervised learning, requiring complex models of convolutional neural networks to numerically find the best combination of parameters [57]. Thus, the standard approaches to reinforcement learning enables free-policy learning by reinforcement, but this is paid in terms of cost of the resources (very deep networks are needed) and accuracy.

Figure R2. Colour maps of the exploration accuracy as a function of the number of trials per experiment and of the steps per trial for both the Deep Q-learning approach (state-of-the-art) and the bio-inspired solution proposed in this work, respectively. Note that the bio-inspired hardware assures better accuracy results thanks to the intrinsic resilience and RRAM-based in-memory computing.

- *Power efficiency and area efficiency*

- The power and area efficiencies are also studied for benchmarking our solution with respect to the software-based state-of-the-art. Since it is hard to perform a benchmark between our approach and the standard software-based due to the diversity of the hardware and of the algorithms, we have carried out a study for assuring a fair comparison.

In particular, as you can see from the plot reported here below, Figure R3, we compute the number of memory elements (such as RRAMs for our approach or single memory elements for standard processors) that are needed for carrying out an exploration at a certain accuracy (99%) averaged over 200 experiments. Note also that the environment under examination is the same reported in Figure R1, without environmental changes. The number of computing elements needed for the computation is directly proportional to the area consumption and to the power needed to achieve the target accuracy.

As you can see from Figure R3, now reported in Fig. 4(g), our approach overcomes the state-of-the-art algorithm in terms of area efficiency considering different sizes of the environment under examination. This is due to the fact that the explorative trials of the state-of-the-art approaches aim to develop a policy map from environmental information that is frequently sparse, noisy and delayed, while training procedures are basically supervised and static. In particular, in order to cope with this problem, a convolutional neural network (CNN) is needed to assure a good explorative behaviour limiting the disadvantages of the backpropagation [57]. Keeping into consideration the best CNN (i.e. with the fewest number of synaptic weights assuring

the target accuracy – the MobileNetV2 –) more than 5 M of parameters are needed with almost 1 Billion of Multiply and accumulate operations considering the environmental configuration proposed in Figure R1 [58]. On the other hand, the bio-inspired approach proposes a better solution in terms of computing architecture (plasticity is assured thanks to the STDP algorithm) and exploitation of the resources.

Considering this last point, note that the RRAM-based computation has two further advantages, namely: (i) thanks to the in-memory computation, no data transfer between the processor and the conventional memory is needed; (ii) the bio-inspired algorithm only excites event-based cells, due to the current computation, thus no constant operation of millions of synaptic weights is necessary (furtherly improving the power efficiency of our solution).

Also considering larger environments, the estimation of our simulations highlights that the benefits of the bio-inspired approach are evident, since the power and area consumptions scale linearly with respect to the state-of-the-art approaches once the size of the environment increases (biggest RRAM-array and deeper neural networks are needed for both the approaches).

Thus, the advantage of the RRAM-based bio-inspired hardware is two-fold since (i) the bio-inspired computation requires a lower number of synaptic weights, (ii) the use of non-volatile memories built in the back end of the line reduces the power consumption, which, on standard technological platforms is affected by the Von Neumann bottleneck (refer to section “Introduction”).

Figure R3. Comparison in terms of memory computing elements between the deep Q-learning procedure and the bio-inspired solution at increasing sizes of the environment to explore. Note that the power consumption is also furtherly improved in the bio-inspired solution thanks to the use of RRAM memory devices built in the back end of the line, which avoids the Von Neumann bottleneck typical of standard computing platforms.

- *Reliability*

- Both the approaches, hardware bio-inspired and software Q-learning, provide great results in terms of reliability and robustness. In particular, once the policy map is created and no environmental change occurs, none of the approaches highlight trial failures and the reward is always found.

- *Resilience*

- In order to test the resilience of the software-based reinforcement learning, we studied the impact of a continually changing environment on the performance of the reinforcement learning computation. In particular, in order to generalize this topic, we propose here a simple theoretical approach for both the bio-inspired network and the standard software-based.

The bio-inspired network, as we report in Section “Exploration, optimization and recall”, is plastic, in the sense that it maps the fire activity, as well as the penalties and rewards, by means of the homeostatic STDP, Eq. IV. This point directly affects the behaviour of the agent, since the bio-inspired paradigm modifies on-line the synaptic connections correspondingly to the current configuration of the environment, as shown in Fig. 3. Thus, an environmental change does not affect the correctness of the computation, provided that the environmental evolution is not too fast, Supplementary Fig. 9.

On the other *hand*, *conventional software algorithms based on Q-learning techniques*, generally rely on deep neural networks which require a big number of training data. Thus, if the data distribution changes, as it happens in reinforcement learning for adapting the agent’s behaviour to a changing environment, the computation can be problematic since data samples are not independent to one another. In particular, Q-learning requires to map the state (M_{state}), the action (M_{action}) and the reward (M_{reward}) of each point of the maze in a matrix form which has the dimensions of the environment. In formula:

$$M = \begin{bmatrix} m_{1,1,k} & \cdots & m_{1,N,k} \\ \vdots & \ddots & \vdots \\ m_{N,1,k} & \cdots & m_{N,N,k} \end{bmatrix},$$

In which “N” represents the side dimension of the environment and $k = 1, 2, 3$, the state, the action and the reward information, respectively. Thus, the formula for defining the Q-value of a state can be described as follows:

$$Q(s, a) = Q(M_{state}, M_{action}, M_{reward})$$

The goal of deep Q-learning is to minimize the cost function of the neural computation at every iteration “i” of each trial. The cost function can be written in this form:

$$\varepsilon_{Loss} = \frac{1}{2} (y_i - Q_i(s', a', \gamma))^2$$

Where “y_i” is the target to achieve (maximum reward per each position) and Q_i(s', a', γ) is the current Q-value matrix.

In this context, the Q-learning takes advantage of a convolutional neural network which looks for the best combination of synaptic weights γ in order to minimize the cost function (i.e., the CNN looks for that combination of synaptic weights which maximizes the Q values). In particular, the output of the neural computation depends on matrix M, since this matrix describes the overall characteristics of the environment to explore. Thus, calling y'_i the output of the neural network, the best combination of synaptic weights is computed using this formula:

$$y'_i = F_{activation}[(M_{state}, M_{action}, M_{reward}) * W_{\gamma} + b],$$

Where “b” is the bias of the neural computation and F_{activation} is the neuronal activation function (such as the sigmoid or the ReLu). Thus, by means of convolutional steps (* operator), the CNN is able to find the best combination of synaptic weights γ in order to optimize the behaviour of the neural network for the reinforcement learning task.

This point means that, in contrast with the bio-inspired approach where the computation relies on the live experience of the agent, for standard Q-learning the calculation of the best synaptic weights depends on the current configuration of the environment which directly affects the “M” matrices.

Thus, if after some trials the configuration of the environment changes, the synaptic weights carried out by the convolutional operation (which can be in the order of millions [58]) are not optimized for the new topology, driving the user to a full re-training of the network. This behaviour well resumes the stability-plasticity dilemma of standard neural networks, as reported in section “Introduction” [40].

It is possible to optimize the neural network adaptation (see here (“<https://github.com/accel-brain/accel-brain-code/blob/master/Accel-Brain-Base/README.md>”) some optimization techniques), however without reaching the plastic features of bio-inspired neuromorphic networks. Furthermore, expensive (for time and power) re-trainings of the convolutional neural network are necessary for enabling more plastic behaviours in standard software-based Q-learning.

We can conclude that the bio-inspired solution is intrinsically more prone to adaptation and resilience with respect to standard approaches.

This discussion has been reported in the supplementary section “Comparison of the resilient properties between bio-inspired and deep learning approaches”.

Thus: it is possible to provide a learning by scratch even with conventional algorithms but with lower efficiency with respect to our solution. On the other hand, considering the power management and the resilience, our work clearly overcomes the state-of-the-art.

Note that, in order to comply with all the requests of the reviewer, in section “Discussion” we also provide a complete comparison with the “state-of-the-art” literature, of which we present in the following answer a brief summary, also considering the changes introduced after this review.

If the calculation of the pure software situation is challenging, please refer to other studies that have reasonably provided alternative comparison standards to highlight the performance superiority of the proposed hardware system.

We thank the reviewer for arising this relevant point: as we have just discussed in the previous answer, we have now improved the manuscript discussing a fair comparison of our solution with respect to pure software Python-based Q-learning, deep Q-learning and Multi-agent Deep Q-Network (refer to section “Exploration, optimization and recall” and to section “Discussion”).

In the following, we provide a brief overview over the comparison of our work with respect to several studies in the literature, as requested by the reviewer. All these outcomes are reported and furtherly highlighted in section “Discussion” and in the Supplementary information of this work.

Comparison with state-of-the-art literature

In section “Discussion” and in the supplementary “Discussion over the RRAM and CMOS approaches to reinforcement learning” we propose a study over the main features of reinforcement learning based on deep neural networks [57,64]. Since deep learning lacks plastic capabilities for adaptation, we have investigated how this feature can be targeted by additional algorithms (generally paying in complexity and time/area-consumption [40, 58]).

While deep training algorithms are generally “supervised”, bio-inspired homeostatic STDP is “unsupervised”, which means that it does not need a training algorithm since it directly learns from the “lived experiences” [54]. Furthermore, this is obtained on-line (i.e., in real time, during the movement of the agent) and at low power, emulating the brain functionalities [38], while backpropagation requires several iterations per single training procedure.

Note that bio-inspired learning is intrinsically prone to resilience and adaptation, since a change of the environment does not require a full re-training (as already discussed in the previous answer of this document).

Note also that deep Q-learning techniques suffer from unstable learning under some conditions of bias overestimation which requires a mutual training of a multi-layer-perceptron (MLP) network

and a correct setting of the learning rate [66]. This could affect the effectiveness of the training algorithm when the system has to map the environment autonomously, requiring a network of several layers with the Adam optimizer applied for stochastic optimization [67]. All these features assure high accuracy but, at the same time, require significant effort for the training procedure. Contrarily, the bio-inspired learning has two main advantages, namely: (i) it can learn a policy map from scratch without relying on backpropagation training, (ii) it relies on in-situ hardware computation thanks to the RRAM devices. This approach improves a lot the time efficiency, Fig. 4(b), and the energy consumption, Fig. 4(c), while keeping high the accuracy, Fig. 4(f), (refer also to Supplementary Fig. 9). Furthermore, the STDP does not require particular methods for stochastic optimization [36].

There are several applications in which reinforcement learning for autonomous navigation and goal disclosure is relevant. For instance, the problem of finding a lost gamma source in an irradiation room without human presence can be solved by combining convolutional neural networks (CNNs) and Q-learning [57]. Note that in this case, even if the stimuli coming from the external world is different with respect to what we propose in Subsection “The event-based architecture”, the behaviour of the agent is pretty the same and it offers a convergence similar to what we depict in Fig. 4 of the manuscript.

Furthermore, note that Fathi et al. used 231,000 matrices as dataset for each case study in order to make effective the CNN training. Each matrix is proposed to the network every single epoch and the synaptic weights of the whole network (consisting of 3 convolutional layers, 2 fully connected layers and the output) are adjusted in order to reduce the loss function. Considering the number of the synaptic weights (which are, at least, in the order of hundreds of thousands), the calculations and the necessary epochs for convergence (at least 15), it is evident that the cost in terms of time and power consumption is very high.

The Markov decision process, Q-learning, $TD(\lambda)$ and deep learning are not the only topics to which the scientific community refers to for modelling and designing reinforcement learning algorithms.

For instance, the multi-bandit problem is often taken as benchmark. The multi-armed bandit problem deals with an agent that attempts to make decisions as a consequence of previous experiences but, at the same time, it needs to acquire new knowledge for the next decision-making events. In order to cope with this framework, several works have proposed the use of recurrent neural networks (RNN) for enhancing the re-use of past information [68].

In the same conceptual context, there have been several attempts in building “meta-learners”, i.e. training a system on a distribution of similar tasks, in the hope of generalization capability when novel tasks are targeted [69,70].

However, even considering these meta-approaches, several CNN-based training algorithms are anyway demonstrated to be necessary to provide the system with an optimum policy map for the required navigation task, thus falling again in the power and time bottleneck described before.

<Original Question 2>

The author needs to specify which learning methods of the Hebbian learning has been employed in their neurocomputing system, such as long- or short-term potentiation, STDP, SRDP, or LIF. Also, the corresponding Hebbian responses of the memristor cells should also be experimentally demonstrated.

The author clearly provided the specific methods of Hebbian learning. In the case of STDP in this work, the STDP behavior is emulated by a programming language. However, there are other memristor studies [1],[2] that have emulated the different STDP behavior, by sequentially applying the actual voltage pulse to the top electrode (PRE-neuron) and bottom electrode (POST-neuron) of one memristor, (similarly, middle sub-figure of Fig. 1a). Therefore, it is recommended that the middle sub-figure of Fig. 1a conveys the correct STDP implementation scheme that could be confused as the different STDP implementation.

<References>

[1] Panwar, Neeraj, Bipin Rajendran, and Udayan Ganguly. "Arbitrary spike time dependent plasticity (STDP) in memristor by analog waveform engineering." *IEEE Electron Device Letters* 38.6 (2017): 740-743.

[2] Lu, Ke, et al. "Diverse spike-timing-dependent plasticity based on multilevel HfOx memristor for neuromorphic computing." *Applied Physics A* 124.6 (2018): 1-9.

We would like to thank the reviewer for highlighting this point related to the STDP, which is one of the most relevant topics proposed in this paper for achieving efficient reinforcement learning.

In particular, we have now improved Fig.1 (where we propose a description of the signals involved in the STDP paradigm), Fig.2 (where we report a schematic for the STDP hardware implementation), Fig. 4 and Section "The event-based architecture", presenting both theoretical analysis and experimental demonstrations. We have also provided a supplementary section "Technical overview over STDP learning paradigm" in order to avoid any kind of misunderstanding related to the implementation of the STDP.

In particular, the STDP paradigm implemented in our network is not an emulation performed by a programming language but a real in-memory implementation.

As in [X1, X2], we are applying voltage pulses to the synaptic elements in order to provide potentiation and depression, respectively. However, differently from [X1, X2] we are following the STDP approach reported in the new Fig. 2(a), which does not involve only the top electrode and bottom electrode of a specific non-volatile memory cell but also the nmos selector in series to the RRAM. The pre-neuronal signal (which stands for the current position occupied by the agent) excites the gate of the selector of the synaptic RRAM element by sending a burst of rectangular pulses (2.2V of amplitude with duration 700ns). At the same time, the Top Electrode of the synaptic element is biased at a read voltage (which can range between 50 mV to 150 mV), thus driving the selector in ohmic state. The current, which depends on the state of the RRAM synapse,

Fig. 1(b), is integrated, as also reported in Fig. 2(b), and then compared to the internal threshold of the post-neuron, which is ruled by the corresponding “state” device, Fig. 2(a) and Fig. 2(d). If the threshold is overcome, a programming signal arises and directly potentiates the synaptic element acting on the top electrode of the synapse connecting the current position with the next firing neuron. At the same time, the neuronal threshold is correspondingly updated in order to keep trace of the fire excitability of the new spiking neuron (homeostatic mechanism), Fig. 2(a-d).

What we have just described takes into consideration a configuration in which the pre-neuron (the current position of the agent) excites the post-neuron, which eventually fires. However, biological studies demonstrate that the post-neuron could also fire before the pre-neuron, thus causing depression of the synaptic connection, middle picture in Fig. 2(a) [25]. This “time-dependence” (which, nevertheless, gives the name to the spike-timing-dependent plasticity), is mapped in hardware by using LFSRs (linear feedback shift registers) which randomly select neurons of the network to give rise to uncorrelated spiking activities. Note that the biological firing activity has been modelled as in Fig. 2(a), where also the refractory period and the depression signal are shown. In particular, when such depression signals happen in the nearest positions to the current one, the connecting synaptic element is reset (i.e., depressed) thanks to the negative polarity of the top electrode, as already demonstrated in previous works [36]. Furthermore, the reset signal starts with a “refractory period” of $1\mu\text{s}$, as it happens in biological neurons [36]. Note that the generation of the pulsed programming signal can be managed internally by the hardware system or, conversely, by the pulser generator (refer to Supplementary Fig. 7-8) for debugging purposes and very precise programming of the synaptic weights.

A precise definition of the synaptic weight requires not strictly necessary additional power consumption. In fact, differently from what reported in [X1, X2], the STDP algorithm can be also operated in a bistable way (i.e., switching the synapse from LRS to HRS and vice versa): in this way, the exponential model of the STDP is digitalized and the overall system requires less power consumption and area. The digitized STDP is not affecting the accuracy of the neural computation, as discussed in several works demonstrating the low-power neuromorphic computation based on the STDP algorithm [37,54]. On the other hand, the multilevel programming of the state device is fundamental for the homeostatic definition of the “state” device since it sets the internal threshold of each position and keeps trace of the neuronal excitability.

Finally, note that the STDP functionalities are also described in the codes reported in the folder “GitHub codes”.

[X1] Panwar, Neeraj, Bipin Rajendran, and Udayan Ganguly. "Arbitrary spike time dependent plasticity (STDP) in memristor by analog waveform engineering." **IEEE Electron Device Letters** **38.6** (2017): 740-743.

[X2] Lu, Ke, et al. "Diverse spike-timing-dependent plasticity based on multilevel HfOx memristor for neuromorphic computing." **Applied Physics A** **124.6** (2018): 1-9.

<Original Question 3>

The author implemented a hardware-based RNN reinforcement learning (RL) platform. However, it is not clear that how the conventional software RNN and RL algorithms have been implemented using the RRAM crossbar. Schematic representation of the software RNN/RL in conjunction with the hardware implementation could highlight the hardware interpretation more clearly, similarly in Fig. 1a,b in ref1. ref1) Milano, G., Pedretti, G., Montano, K. et al. In materia reservoir computing with a fully memristive architecture based on self-organizing nanowire networks. Nat. Mater. (2021). <https://doi.org/10.1038/s41563-021-01099-9>

The revised “Theoretical modelling for in-memory reinforcement learning” Section provided a clear implementation of the reinforcement learning based on the circuit parameters.

We would like to thank the reviewer for highlighting the clarity of the reinforcement learning theoretical description provided in the revised Section “Theoretical modelling for in-memory reinforcement learning”. This theoretical modelling, in particular, is a key point for demonstrating the most relevant features of bio-inspired in-memory computing for artificial intelligence.

However, it is still not clear how the reinforcement parameters are computed from the circuit parameters. Can the author provide the mathematical relationship between the reinforcement and circuit parameters (e.g. $R = \text{functon}(G_{\text{syn}}, G_{\text{state}})$), or if it is challenging, provide the waveforms for representative reinforcement parameters (e.g. R, P, TDN,t) in the cases of the reward (final goal) and penalty (wall)?

We thank the reviewer for suggesting this relevant improvement related to the modelling of the in-memory reinforcement learning.

In the following we provide a mathematical relationship between the reinforcement and circuit parameters and the corresponding waveforms of the learning activity. In particular, the reference we provide in terms of waveforms of the learning activity covers a key role, since the stochasticity of the process due to the variations of the RRAM elements can be tracked on average only.

All the reported improvements have been mainly described in the novel Section of the supplementary information “Additional insights over the theoretical modelling of bio-inspired networks for reinforcement learning”. In this way, we can respect the editorial requests for the maximum number of words of the main manuscript. However, we have also provided references to this topic in Sections “Theoretical modelling for in-memory reinforcement learning” and “Methods”.

Reward and penalty function

The reward function acts on both the synaptic connection linking the positions “ $s - s'$ ” and the internal state “ s ”. In particular, when the agent undergoes a reward, the internal state is driven to

high resistive state, in order to lower the threshold, Fig. 3(h), while the connecting synaptic element undergoes potentiation due to usual STDP evolution, Fig. 3(j). Conversely, if a penalty occurs, the internal state is brought to LRS abruptly, thus causing an increase of the threshold, Fig. 3(g) while the connecting synaptic element is depressed, Fig. 3(j). The depression of the synaptic element could be avoided, since the depression would have been anyway provided to these positions by the stochastic noise applied to the system with the LFSR registers used for accessing random neurons (refer to subsection “In-memory computing for autonomous navigation”). Thus, the reward function, for both state and synaptic elements, can be expressed by the following formulas:

$$R = \begin{cases} \frac{d\phi}{dt} = Ae^{\frac{-E_A}{kT(z)}} \left(\frac{\phi_{MAX} - \phi}{\phi_{MAX}} \right), \text{ for every synapse receiving a reward and state receiving a penalty} \\ \frac{d\Delta}{dt} = Ae^{\frac{-E_A}{kT(z)}}, \text{ for every synapse receiving a penalty and state receiving a reward} \\ G_{syn}(s, s') = \max\left(\frac{d\phi}{dt}\right), \text{ LRS for all the rewarded synapses – Binary STDP} \\ G_{syn}(s, s') = \max\left(\frac{d\Delta}{dt}\right), \text{ HRS for all the synapses undergoing penalty – Binary STDP} \end{cases} \quad (S_1)$$

Where:

- ϕ is the conductive filament diameter that grows inside the RRAM Silicon oxide following the Arrhenius Law during each set. In particular, the migration in the oxide of ionic defects is what rules the modulation of the resistive state of the RRAM [3X]. In fact, the final multilevel resistance of the device depends on the number of defects brought in the device from the top electrode reservoir. Note also that the ionic defects are driven by the electric field due to the applied voltage pulses to the top electrode and thus to the risen temperature “T” along the cross-section of the device “z”. The temperature profile can be obtained solving the 1-D steady state Fourier equation, which depends on the current density J which crosses the device [3X]. The current J is chosen by setting the gate voltage V_G , Fig. 1, of the selector. Different values of J give rise to different values of T and to different values of R.

Thus: the temperature dependence is directly related to the voltage applied to the top electrode of the RRAM device, Fig. 1(b), and to the gate of the RRAM selector, Fig. 1(c). We cannot provide further details about the physics of the built devices due to non-disclosure agreements.

- ϕ_{MAX} : Maximum reachable filament diameter. We cannot provide more details about this value for non-disclosure agreements.
- Δ : the migration of the charges due to an inversed electric field generates a gap in the oxide, which increases the resistance. For this RESET process the same physical laws apply as for the conductive filament ϕ , but with opposite polarity.

Note that the physics for the formation and disruption of the conductive filament is the same (only the driver of the electric field changes polarity, i.e., positive or negative voltages to the top electrode).

Furthermore, note that the analog increase or decrease of the resistance is directly proportional to the gradual creation and disruption of the conductive filament. Thus, the formulas are also useful to describe the homeostatic multilevel mechanism, Fig. 3(f).

Synaptic evolution trial after trial

Those synapses which do not undergo the reward function, i.e., penalty or rewards, show a pure STDP tendency. For this reason, the proper description is time dependent.

The average synaptic evolution in time can be expressed by the following formula:

$$\frac{dG_{syn}}{dt} = AR_{PN}(G_{MAX} + G_{MIN} - 2G_{syn}) + C(G_{MAX} - G_{syn})(G_{syn} - \alpha NG_{MIN})(P - N)R_P \quad (S_II)$$

Where:

- A, C, α are fitting parameters.
- G_{syn} is the average conductance value.
- G_{MAX} and G_{MIN} are the maximum and minimum values of the synaptic conductance.
- P is a figure of merit for the density of pre-neuronal spikes at the 1T1R gates of the network.
- R_P is the frequency of pre-neuronal spikes at a specific 1T1R gate.
- R_{PN} refers to the uncorrelated neuronal fire rate activity which could rise to random fire activity (refer to section “In-memory computing for autonomous navigation”).
- N is a figure of merit for the density of stochastic depression signal which can counteract the synaptic potentiation, on average. This is very relevant for matching the bio-inspired algorithm [36], as we report in Subsection “In-memory computing for autonomous navigation”.
- The first part of the equation basically reports the condition for which, if no correlated excitation happens ($R_P = 0$), i.e., the agent is not exploring the environment, the system does not evolve towards any valuable state (something like white noise condition) and the average synaptic evolution remains undefined.
- The second part of the equation describes the actual behaviour of the STDP during the reinforcement learning exploration: if the burst of the current neuronal activity is significant ($P > N$), then, on average, the synaptic elements of the walked paths tend to have a lower resistive value. On the other hand, if we increase the stochasticity of the algorithm ($P < N$) the system evolves towards a high resistive state (the exploration is not effective anymore). Thus, the stochasticity is relevant only to increase the performance when it is very localized in the network (for increasing the randomness of the explorative trials).
- This model is very general and can be also extended to pure unsupervised learning activities (e.g., for image learning, as we report in the folder “GitHub codes”).

Probability function

The probability function “P”, which describes the probability of the quality of a certain position “s” for moving towards another state “s’”, is dependent on the homeostatic-based STDP mechanism. Thus, since everything is related to the RRAM evolution in terms of resistance, the probability

function is directly dependent on the physical parameters described in Eq. S_I. Furthermore, the probability function depends on the synaptic evolution, trial after trial, of the synaptic connection between state s and s' , Eq. S_II and Fig.3(j).

Moreover, note that the assessment of the probability function is dependent on the current configuration of the system: the bio-inspired algorithm reacts as a function of it, and it provides the best behaviour for getting the final reward, Fig. 3.

$TD_{N,t}$

Since the formula depends only on Q factors and since the Q factor is completely determined by the aforementioned equations, also $TD_{N,t}(a, s)$ is completely determined.

We have now improved both the manuscript and the supplementary material, as reported here above.

[3X] Ambrogio S., et al. Analytical Modeling of Oxide-Based Bipolar Resistive Memories and Complementary Resistive Switches. **Transactions on Electron Devices**, **61**, 2378-2386, 2014.

<Original Question 4>

The working principle and the system structure are not easily understandable in 'The event-based architecture' section and Fig. 2. The explanation of the working principle is dispersed and repeated in the manuscript too broadly, from Result to Discussion sections.

As author has addressed, the revised Fig. 2 improved the readability of the manuscript. To clearly demonstrate the methodology, please provide the detailed pulse geometry (width and amplitude) for the STDP, SET, and RESET processes. One further recommendation is to reconsider the main figure configuration since Fig. S8 (flow chart) is much more intuitive to understand the working principle than Fig. 2b, c.

We thank the reviewer for highlighting this point.

We have now provided the detailed STDP pulse geometry (SET, RESET) in Fig. 2(a), and in the manuscript, Subsection "The event-based architecture". In particular, we have also better reported the used programming pulses for the homeostatic multilevel programming of the internal states with respect to the experimental results shown in Fig. 1.

We have also modified Fig. 2(b) in order to make it resembling Fig. S8.

In particular, Fig. 2(b) has been now designed to provide a clear explanation also for the functional block diagram of Fig. 2(c). As suggested, we have decided to draw inspiration from Supplementary Fig.8 for this part of the image but not for all the other sub-figures (d,e). This choice has been taken because Nature Communications has a wide audience coming from

different scientific fields, while Supplementary Fig. 8 has a lot of technical details related to the effective hardware realization. Thus, we have preferred to highlight the main conceptual working principles in Fig. 2, that nevertheless introduces the reinforcement learning topic which is the main reference of this work.

Last but not least, we have also improved Supplementary Figure 8 following all the advice provided during this review process.

<Original Question 5-2>

Fig. 2a, 2b are not sufficient to demonstrate the working principle. Schematic flow charts for specific cases (e.g, wall at north and no wall at east, etc) could convey the RRAM-based reinforcement learning implementation, not as just waveforms similarly in Fig. 3. The flow charts might include 1) the initial conductance of the 32×32 in 'State' and 'Synapse', 2) how the 'penalty' and 'reward' are performed via Hebbian learning, and 3) how the conductance is either depressed or potentiated based on the maze (or Mars environment) geometry, etc.

Although author additionally provided the data for the 1) ~ 3) aspects, further methodology needs to be clearer. In Fig. 2b, how are the initial matrices defined (e.g. Gaussian distribution) and achieved (e.g. programming amplitude and width)?

Thank you for highlighting this point. We have now specified in Fig. 2(b) the distribution of the initial matrices, both in the sub-figure and in the caption. Furthermore, we have clarified the concept behind the initial programming of the weights, also in relation to the device characterization reported in Fig. 1 (you can also refer to section "Methods").

In particular, note that any distribution of the synaptic and internal states would be fine, since the system does not depend on the initial conditions but on the evolution of the reinforcement learning algorithm, as specified in Section "The event-based architecture". However, it is preferable to prepare moderately high resistive internal states in order to decrease the neuronal integration time. In fact, at each step, the post-synaptic currents are integrated by the nearest neurons, identified with the cardinal positions, eventually leading to firing activities [52]. This behaviour is similar to what is observed in bio-inspired winner-take-all (WTA) networks, where the output neurons compete with each other to specialize on different tasks [53,54].

The programming of the devices was performed in this way: (i) First, we took the arrays and we formed all the RRAM devices (refer to Methods section) using a top electrode voltage of 3V, as reported in Supplementary Fig. 2; (ii) After this step we provided set/reset programming pulses to all the cells in order to study the yield of the arrays; (iii) then, we applied a programming with compliance current $I_C = 54 \mu\text{A}$, refer to Fig. 1(d), for the synaptic matrix; on the other hand, for the V_{TH} state matrix, we applied a stop voltage $|V_{\text{STOP}}| = 1.1\text{V}$, as we report in Fig. 1(g); (iv) finally, we activate the LFSR registers in order to select random positions and provide random HRS ($|V_{\text{STOP}}| = 1.1\text{V}$) on the synaptic matrix, and random LRS ($I_C = 54 \mu\text{A}$) on the V_{TH} state matrix. Thus, the final distributions for both synaptic and state matrices are slightly bimodal, as reported in Fig. 2(b).

We have now accordingly updated the manuscript with all this information.

1. In the same page of Fig. 3, how are the integration signals discharged in the circuit structure?

We thank the reviewer for arising this point. The integration signals are discharged after each fire event by switching on a transistor in parallel to the capacitor used for integration, Fig. 2(a). When off, the resistance of the discharging transistor is in the order of the hundreds of $M\Omega$, whereas when it is turned on it is in the order of tens of Ω (the time constant is much faster, the integrating capacitor is easily discharged, and the neurons are ready for the next computation).

We have also experimentally investigated a further method to apply the discharging signal (generally it is frequent to find in literature the definition of “inhibitory synapses” for such circuits, in parallel to the effective biological observations). This further method is mainly digital, and it works as reported in the following: (i) the readout current from the synaptic elements feeds transimpedance amplifiers (OTAs) which translate the signal into the voltage domain; (ii) then, such voltage is sampled by an analog-to-digital converter and (iii) digitally integrated. Finally, (iv) the value is used in the computation with respect to the neuronal threshold and zeroed after the eventual firing activities.

Even if this method is less bio-inspired due to the ADC and digital management of the information, it is anyway useful for debugging purposes, as we report in the supplementary information.

2. In the next page of Fig. 3, what is the detailed methodology to generate the “Random spiking activity at low frequency”, and why is the stochasticity provided in the system?

We agree with the reviewer that this part of the manuscript was not clear enough. We have now provided a complete and clear explanation about this point, starting from the new version of Fig. 2.

Stochasticity is provided to the system for emulating the biological Hebbian Learning (STDP). In particular, stochastic signals have been demonstrated to play a key role in human brain and thus in neuromorphic engineering for performing a correct synaptic computation (especially for the low-frequency depression of the synaptic elements) [36,57]. We have now better specified this sentence in Section “In-memory computing for autonomous navigation” to avoid any kind of misunderstanding.

We generate random spiking activity by sending programming pulses to random neurons selected by means of linear feedback shift registers (LFSRs) implemented in the programmable logic (FPGA) of the SoC. LFSRs provide pseudorandom outputs if the initial seed is constant in time while, if the initial seed changes in time, the LFSRs can be used as pure random number generator.

Once the LFSRs generate random positions, the hardware system sends “reset” (negative) programming pulses to the top electrode of the selected devices superimposed to the bias voltage, Fig.2(a) [37]. This operation is made possible by properly managing the direct memory access (DMA) of the memory arrays. It is also possible to send “Set” programming pulses after selecting

random neurons, but we did not observe effective benefits in doing so. We have now provided a better explanation of this topic in section “In-memory computing for autonomous navigation” and in section “Methods”.

3. In the next page of Fig. 3, what’s the specific situation of unwanted “rewarded position runs into a penalty”?

We thank the reviewer for arising this sentence which was not clear enough. This expression, in particular, was referring to a situation in which, after finding the target of the maze, the environmental configuration changed and, considering a particular position, the agent consequently got a penalty in place of a reward. We have now modified this phrase in order to avoid any kind of misunderstanding.

We would like to thank again the reviewer for all the comments provided during this review process: every suggestion, consideration, and all the observations have been very useful and profitable for improving the overall quality of this work.

REVIEWER COMMENTS

Reviewer #2 (Remarks to the Author):

The authors have made substantial improvements to address the previous comments.

We would like to thank the reviewer for his/her contributions in making this work worth of publication in Nature Communications.

During this review, we have furtherly improved our manuscript considering all the suggestions we received. In particular, we have described in detail the outcomes of our solution with respect to software-based approaches. We have also optimized the test and the figures of both the main text and the supplementary information to highlight the obtained results.

REVIEWER COMMENTS

Reviewer #3 (Remarks to the Author):

Thank you for the updates. I think the revised manuscript becomes much better than the initial version. I now understand the author's point of view about novelty of this paper and difference from previous papers.

We thank the reviewer for all the useful suggestions provided during the review process: every observation has helped us to improve the quality, the clarity and the readability of this manuscript. Now, we propose a new version of the manuscript which addresses all the reviewer's observations and introduces further theoretical studies and experimental results in order to cover the last open points.

In the following, we report a point-by-point answer to each one of the comments. We also provide precise references to the points in which the main manuscript and the supplementary material have been accordingly updated.

1. New questions arises about STDP. It is not clear to me about how STDP is implemented in this system. I suggest that the author describe the details in this paper because STDP learning is one of the most important key enablers for the proposed system.

We would like to thank the reviewer for this relevant suggestion. We agree that the previous version of the manuscript was not reporting enough technical details with respect to the homeostatic STDP implementation.

We have now provided further qualitative and quantitative information about this topic and covered step-by-step all the open points highlighted by the reviewer.

In particular, we have updated Fig. 2(a), where we report now the hardware realization of the STDP learning paradigm with the corresponding electric signals. Furthermore, we have improved Sections "Introduction", "In-memory computing for autonomous navigation" and Fig. 1 to provide a better overview over the bio-inspired reinforcement learning based on homeostatic STDP.

Finally, we propose a detailed comparison with respect to the state-of-the-art by precise reference to standard deep learning libraries in Python framework used for conventional reinforcement learning in 2D environments, Fig. 4(f,g). We have also inserted further insights and analytical analysis about STDP in the supplementary information, section "Technical overview over STDP learning paradigm".

For instance, the questions and unclear points to me are

A. In reference paper[36], Fig.1 describes how to implement on-chip STDP learning. Is this circuitry (Fig.1-(a)) or slightly modified circuitry also used in this paper? If not, how the spike activities on pre- and post-neurons can be monitored in this system? How does the system detect either pre- or post- neuron fires earlier or later?

We would like to thank the reviewer for arising this point with respect to the STDP hardware implementation. We agree that this topic is crucial for a complete understanding of the presented work.

In the following of this answer, we are going to make a comparison with respect to previous implementations of the STDP paradigm [36] and to provide a detailed description about the STDP circuital structure presented in this work, now better reported in Fig. 2(a).

All the additional information here described is reported in the manuscript and/or in the supplementary material.

Detailed comparison of STDP implementations

The STDP structure reported in [36] shares with the current STDP implementation the basic idea of time-dependent plasticity and some of the behavioural features reported in Section “The event-based architecture”. However, it does not reflect the effective hardware realization. In the following, in order to avoid any kind of misunderstanding, we are going to summarize the main differences and similarities between the two approaches:

- The implementation of STDP in [36] did not take into consideration a modulable neuronal internal threshold, as instead we have proposed in this work. The spike-frequency control mechanism, by means of a further “state” RRAM device, modifies the integration time and introduces new behavioural features such as the homeostatic plasticity. In particular, as you can see in Section “In-memory computing for autonomous navigation” and “Exploration, optimization and recall”, the homeostatic feature is fundamental for increasing the computational accuracy.
- The STDP circuit in [36] was a proof of concept with several pre-neurons, represented by the spike activity at the gates of the selectors of the 1T1R cells (i.e., the synapses), with only one post-neuron. In particular, this configuration was used to demonstrate the possibility of implementing a “pattern learning activity”. In this experiment, an image (pattern) was presented at the input of the system (alternated with uncorrelated information) by means of pixel-voltage spikes. As a consequence, the synaptic elements re-arranged their resistive status accordingly to the STDP paradigm for recording without supervision the aforementioned pattern (“bright pixels” were translated into LRS, while “black pixels” into HRS).

In the presented work, instead, the STDP is working in a different way: the current position of the agent is recorded in the SoC which sends spike voltage signals ($V_{\text{GATE}} = 2.2 \text{ V}$), Fig. 2(a), to the input of all the 1T1R RRAM devices which are connecting the current position to the 8 nearest post-neurons that represent the 8 principal cardinal directions, Fig. 2(c). Note that, once one post-neuron reaches its internal threshold (managed by the corresponding “state” device, Fig. 2(d)), that neuron fires and it sends a signal backward to the top electrode of the synaptic signal, Fig. 2(a). Note that this feedback mechanism is used to set the synapse as a function of the fire activity of the post-neuron. Thus, when the agent moves from one position to another one, the linking synaptic element is potentiated (i.e., the RRAM connecting the two positions is set to a low resistive state, LRS). Note that such

feedback-based update of the synaptic element, referring to Fig. 2(a), is similar to what reported in [36] but with three main differences: (i) the dependence on the homeostatic control voltage; (ii) the management of the depression signal by means of additional logic (such as the LFSRs registers), as we are going to discuss in one of the following point of this answer; (iii) the condition for which the potentiation is triggered with respect to the effective hardware realization.

Related to this last consideration, note that the programming of the synaptic devices can be performed in two ways: (i) using the implemented circuits on the boards by means of the SoC Zync7000, or (ii) commanding the pulser and the oscilloscope (best option for debugging purposes), Supplementary Fig. 7.

Note that, during the potentiation phase, it is also possible to modulate the V_{GATE} in order to have a precise LRS value, Fig.1. However, in order to simplify the procedure, the STDP can be exploited in a binary way, since no loss of the accuracy is detected [36]. We have now improved Fig. 1, sections “The event-based architecture”, “In-memory computing for autonomous navigation” and “Methods” in order to better describe the STDP implementation. We have also updated the supplementary information introducing a new section, which is named “Technical overview over STDP learning paradigm”.

- The STDP is a stochastic paradigm where also random spikes have a relevant role in the computation [57]. In [36] such “noise” was introduced just after the refractory period which is a time interval in which the post-neuron is inactive as a consequence of a previous fire activity. In this manuscript the leading idea is the same, but the hardware realization is different. Thus, we are going to first analyse the basic concept and then we will treat in detail the effective hardware realization reported in this work.

In particular, keeping into consideration a succession of epochs (the “epoch” is a fixed time slot where a neuronal excitation happens) uncorrelated signals are fundamental to achieve random depression. Synaptic depression takes place when the post-neuronal spiking activity happens before the pre-neuronal spike activity, Fig. 2(a): in [36] after a fire activity and the successive refractory period, the post-neuron sent a negative signal to the top electrode of the activated synapse. If this happened in parallel with the presence of uncorrelated random spike activity at the gate of the 1T1R synapses (in the next computation epoch) it caused the depression of the corresponding synaptic linking elements. In this context, as we are going to discuss in the following, the stability of the system is assured only when the density of the uncorrelated “noise” signal “N” is not too high.

Such approach has been improved in the current work. In particular, we generate random spiking activity by means of linear feedback shift registers (LFSRs) implemented in the programmable logic of the SoC. LFSRs provide pseudorandom outputs if the initial seed is constant in time while, if the initial seed changes in time, the LFSRs can be described as pure random number generators.

In this manuscript, such LFSRs are used to provide random numbers for accessing random positions of the memory arrays and sending reset spikes superimposed to the read voltage at the top electrode of the synaptic devices, Fig. 2(a). The reset signal is constituted by the refractory period (which is in the order of 1 μ s) plus the reset signal. Note that the refractory period has been programmed to be longer than the average integration time in order to assure the best accuracy. Thus, when a reset occurs, there must be such random signal on the top electrode of the RRAM while, at the same moment, the pre-neuronal spiking activity is stimulating the gate of the same synapse: the reset of the cell occurs when the post-neuron has randomly fired before the pre-neuron (due to the fact the reset signal is anticipated by the refractory period, Fig. 2(a)). This concept has been also clarified in Section “The event-based architecture”.

The stochastic signal is fundamental to enable random explorative trials and to increase the overall accuracy, since it induces the reinforcement learning algorithm to discover/inhibit new places of the environment. In order to improve this aspect, it is also possible to randomly provide “set” signal after selecting random neurons (the choice between set or reset pulses is easily managed by a further LFSR which has only two states, one associated to the reset signal and the other to the set, respectively). However, note that the implementation of random set signals besides the reset ones does not introduce significant improvements in the overall explorative accuracy.

Note also that the stochasticity is relevant only if the density of noise inserted in the stochastic computation is kept low, as demonstrated in [36]. If the density of the uncorrelated signal gets higher, as we now describe in the new section of the supplementary information “Additional insights over the theoretical modelling of bio-inspired networks for reinforcement learning”, the system gets unstable, and no good exploration can be carried out by the agent.

- Thus: the concept of the STDP between this work and [36] is pretty similar but the large-scale hardware realization we propose in this work, the homeostatic mechanism applied to the neurons and the novel management of the stochastic signals for the random depression make this approach more accurate and efficient.
- The analytical model described in [36] related to the synaptic evolution of the STDP is here re-adapted, as we now describe in the supplementary Section “Additional insights over the theoretical modelling of bio-inspired networks for reinforcement learning”.

Additional details about homeostatic-STDP

The homeostatic-based STDP is an independent paradigm for expressing the learning evolution of the agent, position after position. Thus, the FPGA is the master of the reinforcement learning system, but the effective potentiation and depression also depend on the intrinsic stochasticity of the paradigm. Furthermore, note that the definition of each internal threshold by means of the

“state” synaptic device randomly characterizes the neuronal firing activity, substantially adding a further degree of freedom with respect to a pure STDP-based implementation. In order to provide a holistic overview about the whole learning paradigm here implemented, we have improved sections “The event-based architecture”, “In-memory computing for autonomous navigation” “Methods” and Fig. 2. Furthermore, we have also inserted a further section in the supplementary information for describing additional ways for the implementation and hardware debug of the STDP paradigm. This section can be found under the title “Technical overview over STDP learning paradigm”.

B. This is related to question-A, but how do you provide pulses on pre-synaptic neurons?

We thank the reviewer for arising this point: we have now clarified this aspect in Fig. 2(a) and in Section “The event-based architecture”.

The pre-synaptic spikes (with amplitude 2.2V and average duration 700 ns) are managed by the SoC (FPGA) and they identify the current position occupied by the agent. Such spikes are then sent to the gate of the 1T1R synaptic devices, Fig. 2(a), in order to integrate the consequent post-synaptic current (refer also to Fig. 1(b)) in the 8 post-synaptic neurons.

The presented system is fully connected and reciprocal, in the sense that, looking at Fig. 2(c), each neuron is connected towards its nearest neighbours by means of two symmetrical synaptic elements, to make possible the movement of the agent back and forth.

Note also that, for debugging purposes of the memory elements, we can study all the signals by means of a pulser (this procedure has been very useful for the collection of the signals depicted, for instance, in Fig. 3). A more detailed explanation about how we measure and debug the system is reported in Supplementary Fig. 6-8.

- Are the pulses provided as digital Poisson spike trains from FPGA first? Then, do the pulses trigger voltage-modulated two continuous pulses shown in Fig-1.(c) and (d) by chip-internal pulse generator?

Thank you for arising this relevant point.

The signals of the pre-synaptic neurons can be provided by the FPGA or by the external equipment, depending on the type of measurement or debug that must be performed on the system. In particular, they can be externally managed by means of a TTI TGA 12104 pulser generator and, at the same time, they can be monitored with the oscilloscope (this procedure has been very useful for the collection of the signals depicted, for instance, in Fig. 3 and for enabling the characterization of the memory devices). A more detailed explanation about how we measure and debug the system is reported in the supplementary material, Supplementary Fig. 6-8, where we also report the way in which we choose the programming voltages by means of the Matlab interface. In particular, note that the accurate multilevel programming of the memory “state” devices reported in Fig. 1 can be assured only relying on the external equipment.

When using the FPGA as master during the standard operative condition of the system, we can provide to the gate of the 1T1R synaptic elements Poisson spike trains, as correctly pointed out by the reviewer (this can be achieved by internal implementation of a specific function). However, for simplicity, it is also possible to send clocked signal to the synaptic elements, which is simpler by a computational point of view, Fig. 2(a). Note also that the voltage level can be easily managed by using reconfigurable level shifters, which is very useful considering the wide range of electrical studies to be performed with respect to the non-volatile memory devices.

Thus, we do not have any pulse generator embedded in the chip since the programming of the memory elements, as well as the management of the pre- and post- neuronal signals, can be done by means of the hardware system (SoC, memory arrays and hosting PCBs, as we report in Supplementary Fig. 3-4); as previously described, the read/program signal management can be also done by means of the pulser generator (or even by the parameter analyser) and measured with the Oscilloscope, as reported in Supplementary Figures 7-8.

- Are the pulses provided as simple digital High/Low signals to select and activate one row? How many pulses are provided? How much pulse duration are used per one pulse?

Yes, the signals used for selecting the row and columns of the synaptic arrays are digital and they follow a predetermined procedure in order to manage the DMA (direct memory access) circuit co-integrated with the memory arrays. All the signals are sent by the SoC (FPGA and programmable logic), and they can be properly modulated in order to get the wanted voltage level. Due to non-disclosure agreements about the internal structure of the DMA circuitry, we cannot provide more details about this topic (refer to Fig. 2(c) for the architectural description).

On the other hand, with respect to the RRAM elements, we provide all the programming signals in Fig.1. In particular, note that the accurate multilevel programming of the memory "state" devices reported in Fig. 1 can be assured only relying on the external equipment (if not, a bit higher variability of the resistive states must be taken into account, however without affecting the overall computed explorative accuracy).

We have also inserted further details about the pulse duration of the signal in Fig. 2(a): note that we can provide pulse durations low to tens of ns. However, the duration of the signals reported throughout the manuscript and the supplementary information have been mainly chosen with respect to the clock frequency designed for the digital core in order to assure a correct interface with the SoC and, eventually, with the programming and debugging equipment.

- There is a statement: "Random spiking activity at low frequency is also inserted in the system by means of a linear feedback shift register (LFSR)". How do you provide it to current integrator? Is it through different RRAM cell?

We agree with the reviewer that the sentence reported in the previous version of the manuscript was not clear enough. We have now provided a complete and clear explanation about this point, starting from the new version of Fig.2. In the following, we are going to briefly summarize the inspiration, the conceptual evolution and the effective hardware realization of the "LFSR topic".

Stochasticity is provided to the system for emulating the biological Hebbian Learning (STDP). In particular, stochastic signals have been demonstrated to play a key role in human brain and thus in neuromorphic engineering for performing a correct synaptic computation (especially for the low-frequency depression of the synaptic elements) [36,57]. We have now better specified this sentence to avoid any misunderstanding in Section "In-memory computing for autonomous navigation".

We generate random spike activity by sending programming pulses to random neurons selected by means of linear feedback shift registers (LFSRs) implemented in the programmable logic of the SoC. LFSRs provide pseudorandom outputs if the initial seed is constant in time while, if the initial seed changes in time, the LFSRs can be used as pure random number generators.

Once the LFSRs generate the coordinates of random positions, the hardware system sends "reset" (negative) programming pulses to the top electrode of the selected devices, superimposing the programming signal to the bias voltage, Fig.2(a) [37]. If this happens in parallel with the excitement of the gates of the synaptic devices by means of the pre-neuronal spiking activity, a reset event of the memory cell occurs, accordingly to the STDP paradigm, Fig. 2(a). This operation is made possible by a proper use of the direct memory access (DMA). The system can also send "Set" programming pulses after selecting random neurons, but we did not measure effective benefits in doing so. We have now provided a better explanation in Section "In-memory computing for autonomous navigation" and in Section "Methods".

Thus, the LFSR is not needed to provide any outcome to the integrator and there is no need of further RRAM elements with respect to the description of the system provided in Fig.2.

- In supplemental document Fig-7, two descriptions: "Reset (i,j) synapse array N" and "Set (x,y) SiOx RRAM synapse connected to the spiking neuron" can be found in "External" red boxes. This confuses me because the procedures sounds deterministic operation based on penalty or reward not using STDP. Could you clarify the relation between STDP learning and this deterministic descriptions?

Thank you for arising this point. The source of misunderstanding comes out from a not correct description of the image in the previous version of the supplementary material. We have now modified the figure in order to improve the clarity and readability of this part of our work and to overcome all the errors.

The "external" box reported by the reviewer was highlighting the possibility of performing the debug of the multilevel programming of the "state" devices by receiving/sending the information off-chip (from the pulser and to the oscilloscope). This procedure is dependent on a complex system, where the pulser and the oscilloscope are managed by Matlab (which is also connected to the PCB board and thus to the SoC, the FPGA and the arrays). In this context, the "box" depicted in the old figure was only related to the possibility of "external" management of the memory devices, by using the system interaction depicted in Supplementary Figure 7.

Thus, the external red boxes do not have any conceptual meaning. We are sorry for not having specified this in a better way in the previous version of the manuscript.

For solving this issue, we have now improved the Figure under examination (Supplementary Fig. 8) and the corresponding supplementary section “Programming for signal management”. In particular, Supplementary Fig. 8 is reporting the global functionalities of the system by a “flow” point of view (the analog-based functionalities of the system are simplified and reported secondarily with respect to the main topic of the image which is the management of the top-level signals of the bio-inspired system).

In particular, the movement of the agent is mapped by means of the STDP and of the homeostatic mechanism, which are intrinsically stochastic, as described in Section “In-memory computing for autonomous navigation”. What is deterministic, instead, is the environment (in the sense that if the walls of the maze, as well as the target, do not change in time, then the agent is always going to experience penalties and rewards in the same positions).

In other words, the environmental configuration can be deterministic and externally defined (i.e., chosen by the user or simply related to a situation), while the bio-inspired procedure to become a master of the system stochastically evolves towards the best situation (the final task of the agent is to build a policy map in a continually changing environment without any pre-compiled information and only relying on the past experiences).

2. In DISCUSSION section, the author refers to [64] as a DNN example about navigation task, then introduces that "3000" is the training episodes for the task. I think the size of navigation space in the reference paper looks much larger than the author's paper, so it is not fair comparison and probably misleads readers.

We thank the reviewer for arising this point: we agree about the need of providing a better framework for the comparison of our solution with respect to standard approaches.

The parallelism with the reported work was mainly referred to Fig. 6 and Fig. 10 of [64], which present a similar cardinality of the environment with the one proposed in our work. However, we agree that the case study is different: even if the size of the environment could be compared with the Mars Rover Navigation Test, it has anyway different goals (e.g., air navigation among buildings).

For this reason, in the following, (i) we demonstrate that our hardware overcomes software-based solutions implementing policy map-free learning (*the comparison is carried out using the official Python environment for software-based reinforcement learning*), and (ii) we propose a very detailed panorama of the most relevant works about autonomous navigation. Note that the benchmark with respect to the standard approaches is done considering the same task (i.e., the same environment to explore).

All the considerations here described are also reported in the main manuscript and in the supplementary information (we will give the details step-by-step in the following of this answer).

Comparison with software-based reinforcement learning

In sections “Exploration, optimization and recall” and “Discussion” we report now a fair comparison with a Python-based library for reinforcement learning. You can access the homepage of the library and all the main codes clicking here (“<https://pypi.org/project/pyqlearning/>”) [57]. In particular, this Python library is modular and can be used for implementing Reinforcement Learning by means of Q-Learning algorithms, Deep Q-Networks, and Multi-agent deep networks.

We link here (“https://github.com/accel-brain/accel-brain-code/blob/master/Reinforcement-Learning/demo/search_maze_by_deep_q_network.ipynb”) the main source of the codes used for the comparison. As you can see, the default maze exploration reported in this Python library is performed in a 2D environment and the shape of the maze is comparable (a bit smaller, i.e. easier) to the one presented in Fig. 4(a) of the manuscript. However, in order to be fair in the comparison, we are going to benchmark the two approaches (bio-inspired and Python-based) using the environment reported in the “GitHub codes” folder attached to this submission, Fig. R1.

Figure R1. Maze used for benchmarking the proposed bio-inspired hardware with respect to the state-of-the-art approaches.

In the following we are going to compare our bio-inspired approach with respect to the Python-based solution. We will first compare the algorithms starting from a qualitative analysis about the model-free reinforcement learning (which deals with the learning activity of a policy map from scratch) for then providing a quantitative analysis in terms of accuracy of the results, power/area efficiency, reliability, and resilience for real life applications.

- *Model-free learning: comparison of bio-inspired hardware with respect to conventional approaches*
 - Model-free methods are crucial when the main goal to provide by means of the neural computation is the exploration of a completely new environment where previous information is not available [62].

As we describe in Sections “The event-based architecture”, “Exploration, optimization and recall” and in Figs. 2-4, our hardware system is based on unsupervised bio-inspired algorithms such as homeostatic “spike-timing-dependent plasticity”, STDP. This choice triggers the possibility of performing “on-line” learning, i.e., real-time learning without needing training datasets or previous labelled information, as we now better describe in Fig. 2(a) with precise reference to the effective hardware realization we propose.

This is in contrast with the so called “supervised learning”, the conventional learning procedure where training datasets are needed. The bio-inspired approach usually pays this feature in terms of lower accuracy [40] but in this work we demonstrate

that, by proper development of the explorative algorithm, it is possible to boost the accuracy of the bio-inspired exploration to overcome the state-of-the-art results obtained with conventional approaches.

Furthermore, note that the conventional approaches require initial explorative trials to develop a policy from environmental information that is frequently sparse, noisy and delayed. This procedure can be tricky for deep learning-based algorithms since they rely on training algorithms, such as the backpropagation, which are not plastic and resilient [57]. In order to cope with this situation, conventional neural networks for reinforcement learning rely on specific trainings that change the loss function at each iteration: this is usually performed by using complex deep Convolutional Neural Networks (CNNs), as we report in Section “Discussion” [57,64].

In the following we analyse the quantitative figures of merit for both the reinforcement learning algorithms, conventional and bio-inspired.

- *Accuracy*

- In order to provide a fair benchmark, we tested the accuracy for a given 2D maze exploration (we provide the file describing the maze under test in the attached repository “GitHub codes”). The standard approach of the conventional free-model reinforcement learning was developed using the Python codes reported here (“https://github.com/accel-brain/accel-brain-code/blob/master/Reinforcement-Learning/demo/search_maze_by_deep_q_network.ipynb”) [57].

Several state-of-the-art algorithms have been tested for providing a fair analysis, all giving comparable outcomes. *For simplicity and clarity of the exposition, we report in the following only the best outcome of the simulated state-of-the-art algorithm (deep Q-learning). All the approaches can be compared by accessing the provided link (for the state-of-the-art solutions) and the repository “GitHub codes” (for the behavioral description of our bio-inspired approach).*

As you can see in the plot reported here below, Figure R2, our approach overcomes the state-of-the-art free-learning algorithm in terms of accuracy. The slight better results of the bio-inspired approach with respect to the pure software-based are due to the more plastic and resilient algorithm to find the solution, which leads to a faster convergence to the optimum result (no training is needed and the system directly reacts as a function of the current experience). In fact, as we report in section “Exploration, optimization and recall”, our hardware system learns on-line, and it becomes a master of the problem relying on the intrinsic plasticity for the continual adaptation to the environment.

The comparison has been now reported in Fig. 4(f), where we propose two color maps expressing the average accuracy, one for the bio-inspired approach and the

other for the software-based deep Q-learning. The accuracy is calculated as a function of the number of trials per experiment and of the number of steps allowed per single trial considering the environment reported in Figure R1. This choice has been done for performing a fair comparison: our hardware is custom-based on RRAM synapses, while the software-based is run on standard processors. Thus, we decided to perform various set of experiments, each repeated 200 times, with different number of trials each (to test the recall property). This number of trials per experiment is also a function of the number of steps that the agent is allowed to do per each trial (y axis). Thus, this figure of merit comes out to be only algorithm related.

Furthermore, note also that the standard neural networks suffer from the delay between taking actions and receiving rewards, which can also be hundreds of timesteps long. In particular, this is daunting when related to the direct association between inputs and targets typical of supervised learning, requiring complex models of convolutional neural networks to numerically find the best combination of parameters [57]. Thus, the standard approaches to reinforcement learning enables free-policy learning by reinforcement, but this is paid in terms of cost of the resources (very deep networks are needed) and accuracy.

Figure R2. Colour maps of the exploration accuracy as a function of the number of trials per experiment and of the steps per trial for both the Deep Q-learning approach (state-of-the-art) and the bio-inspired solution proposed in this work, respectively. Note that the bio-inspired hardware assures better accuracy results thanks to the intrinsic resilience and RRAM-based in-memory computing.

- *Power efficiency and area efficiency*
 - The power and area efficiencies are also studied for benchmarking our solution with respect to the software-based state-of-the-art. Since it is hard to perform a benchmark between our approach and the standard software-based due to the diversity of the hardware and of the algorithms, we have carried out a study for assuring a fair comparison.

In particular, as you can see from the plot reported here below, Figure R3, we compute the number of memory elements (such as RRAMs for our approach or single memory elements for standard processors) that are needed for carrying out an exploration at a certain accuracy (99%) averaged over 200 experiments. Note also that the environment under examination is the same reported in Figure R1, without environmental changes. The number of computing elements needed for the computation is directly proportional to the area consumption and to the power needed to achieve the target accuracy.

As you can see from Figure R3, now reported in Fig. 4(g), our approach overcomes the state-of-the-art algorithm in terms of area efficiency considering different sizes of the environment under examination. This is due to the fact that the explorative trials of the state-of-the-art approaches aim to develop a policy map from environmental information that is frequently sparse, noisy and delayed, while training procedures are basically supervised and static. In particular, in order to cope with this problem, a convolutional neural network (CNN) is needed to assure a good explorative behaviour limiting the disadvantages of the backpropagation [57]. Keeping into consideration the best CNN (i.e. with the fewest number of synaptic weights assuring the target accuracy – the MobileNetV2 –) more than 5 M of parameters are needed with almost 1 Billion of Multiply and accumulate operations considering the environmental configuration proposed in Figure R1 [58]. On the other hand, the bio-inspired approach proposes a better solution in terms of computing architecture (plasticity is assured thanks to the STDP algorithm) and exploitation of the resources.

Considering this last point, note that the RRAM-based computation has two further advantages, namely: (i) thanks to the in-memory computation, no data transfer between the processor and the conventional memory is needed; (ii) the bio-inspired algorithm only excites event-based cells, due to the current computation, thus no constant operation of millions of synaptic weights is necessary (furtherly improving the power efficiency of our solution).

Also considering larger environments, the estimation of our simulations highlights that the benefits of the bio-inspired approach are evident, since the power and area consumptions scale linearly with respect to the state-of-the-art approaches once the size of the environment increases (biggest RRAM-array and deeper neural networks are needed for both the approaches).

Thus, the advantage of the RRAM-based bio-inspired hardware is two-fold since (i) the bio-inspired computation requires a lower number of synaptic weights, (ii) the use of non-volatile memories built in the back end of the line dramatically reduces the power consumption, which, on standard technological platforms is affected by the Von Neumann bottleneck (refer to section “Introduction”).

Figure R3. Comparison in terms of memory computing elements between the deep Q-learning procedure and the bio-inspired solution at increasing sizes of the environment to explore. Note that the power consumption is also furtherly improved in the bio-inspired solution thanks to the use of RRAM memory devices built in the back end of the line, which avoids the Von Neuman bottleneck typical of standard computing platforms.

- *Reliability*
 - Both the approaches, hardware bio-inspired and software Q-learning, provide great results in terms of reliability and robustness. In particular, once the policy map is created and no environmental change occurs, none of the approaches highlight trial failures and the reward is always found.

- *Resilience*
 - In order to test the resilience of the software-based reinforcement learning, we studied the impact of a continually changing environment on the performance of the reinforcement learning computation. In particular, to generalize this topic, we propose here a simple theoretical approach for both the bio-inspired network and the standard software-based.

The bio-inspired network, as we report in Section “Exploration, optimization and recall”, is plastic, in the sense that it maps the fire activity, as well as the penalties and rewards, by means of the homeostatic STDP, Eq. IV. This directly affects the behaviour of the agent, since the bio-inspired paradigm modifies on-line the synaptic connections correspondingly to the current configuration of the environment, as shown in Fig. 3. Thus, an environmental change does not affect the correctness of the computation, provided that the environmental evolution is not too fast, Supplementary Fig. 9.

On the other *hand*, *conventional software algorithms based on Q-learning techniques*, generally rely on deep neural networks which require a big number of training data. Thus, if the data distribution changes, as it happens in reinforcement learning for adapting the agent's behaviour to a changing environment, the computation can be problematic since data samples are not independent to one another. In particular, Q-learning requires to map the state (M_{state}), the action (M_{action}) and the reward (M_{reward}) of each point of the maze in a matrix form which has the dimensions of the environment. In formula:

$$M = \begin{bmatrix} m_{1,1,k} & \cdots & m_{1,N,k} \\ \vdots & \ddots & \vdots \\ m_{N,1,k} & \cdots & m_{N,N,k} \end{bmatrix},$$

In which "N" represents the side dimension of the environment and $k = 1, 2, 3$, the state, the action and the reward information, respectively. Thus, the formula for defining the Q-value of a state can be described as follows:

$$Q(s, a) = Q(M_{state}, M_{action}, M_{reward})$$

The goal of deep Q-learning is to minimize the cost function of the neural computation at every iteration "i" of each trial. The cost function can be written in this form:

$$\epsilon_{Loss} = \frac{1}{2} (y_i - Q_i(s', a', \gamma))^2$$

Where "y_i" is the target to achieve (maximum reward per each position) and $Q_i(s', a', \gamma)$ is the current Q-value matrix.

In this context, the Q-learning takes advantage of a convolutional neural network which looks for the best combination of synaptic weights γ in order to minimize the cost function (i.e., the CNN looks for that combination of synaptic weights which maximizes the Q values). In particular, the output of the neural computation depends on matrix M, since this matrix describes the overall characteristics of the environment to explore. Thus, calling y'_i the output of the neural network, the best combination of synaptic weights is computed using this formula:

$$y'_i = F_{activation}[(M_{state}, M_{action}, M_{reward}) * W_\gamma + b],$$

Where "b" is the bias of the neural computation and $F_{activation}$ is the neuronal activation function (such as the sigmoid or the ReLu). Thus, by means of convolutional steps (* operator), the CNN is able to find the best combination of

synaptic weights γ in order to optimize the behaviour of the neural network for the reinforcement learning task.

This point means that, in contrast with the bio-inspired approach where the computation relies on the live experience of the agent, for standard Q-learning the calculation of the best synaptic weights depends on the current configuration of the environment which directly affects the “M” matrices.

Thus, if after some trials the configuration of the environment changes, the synaptic weights carried out by the convolutional operation (which can be in the order of millions [58]) are not optimized for the new topology, driving the user to a full re-training of the network. This behaviour well resumes the stability-plasticity dilemma of standard neural networks, as reported in section “Introduction” [40].

It is possible to optimize the neural network adaptation (see here “<https://github.com/accel-brain/accel-brain-code/blob/master/Accel-Brain-Base/README.md>” some optimization techniques), however without reaching the plastic features of bio-inspired neuromorphic networks. Furthermore, expensive (for time and power) re-trainings of the convolutional neural network are necessary for enabling more plastic behaviours in standard software-based Q-learning.

We can conclude that the bio-inspired solution is intrinsically more prone to adaptation and resilience with respect to standard approaches.

This discussion has been reported in the supplementary section “Comparison of the resilient properties between bio-inspired and deep learning approaches”.

Thus: it is possible to provide a learning by scratch even with conventional algorithms but with lower accuracy with respect to our solution. On the other hand, considering the power efficiency and the resilience, our work clearly overcomes the state-of-the-art.

3. Regarding scalability in "Mars rover navigation" section, it's interesting approach to utilize past experience against new environment. As the authors describes, the RRAM array already stores obstacle shapes in its memory through the previous exploring. To reuse it as abstracted data is reasonable approach to improve performance in new environment.

We thank the reviewer for arising the goodness of the bio-inspired approach in terms of scalability. In the following, we are going to provide more details about the hardware management with respect to the reconfigurability of the resources in order to improve the quality of the manuscript and to avoid any kind of misunderstanding.

Furthermore, we are also going to investigate a precise comparison between conventional and bio-inspired approaches to reinforcement learning with respect to the scalability of the hardware.

However, there are some unclear points in this section.

- What's the difference between center and right figures in Fig-7(f)? What is the red rectangle in right figure? Why the red rectangles are missing in center figure?

We thank the reviewer for arising this topic. In this answer, in order to provide an organized framework, we are going to better describe the scalability initially proposed in our work. Starting from the provided analysis, in the next answers (as well as in Sections "Reconfigurability of the hardware" and supplementary "Additional insights over the scalability topic"), we are going to investigate and propose a furtherly optimized solution which enhances the scalable features of the bio-inspired hardware for reinforcement learning.

The Mars Rover navigation, that is the framework in which we investigate the scalability of the hardware, is depicted in Fig. 5. Within this figure, Fig. 5(f) shows the exploration of a new environment taking into consideration different sizes of pre-recorded penalty shapes: once the agent receives a penalty, a generic area of the environment is prohibited, thus avoiding a biunivocal memory-position mapping of the environment (this information can be stored in separated registers).

The red rectangles in the right sub-figure are just simple graphical representations of the procedure proposed for boosting the scalability of the hardware. In particular, comparing the right sub-figure with the central one, the different sizes of the red rectangles deal with the possibility of having different sizes of inhibited regions depending on various types of past penalty experiences.

The past experiences create a repository of shapes (different in dimension and configuration) which are recorded into the FPGA and stochastically used as penalty function during the exploration of the new environment, Fig. 5(f). Note also that this part of memory can be virtual, since it does not need to be physical represented in hardware: a numerical function is enough for expressing a region that is not convenient to explore. Such approach slightly improves the efficiency results, Fig. 5(g), and it avoids the physical device-position mapping (a single address is enough to represent a region of space).

Of course, note that there is a trade-off between the dimension of the inhibited region after getting a penalty and the efficiency of the system: if the inhibited region is big, then much area is saved but it is possible to have lower accuracy than expected in Fig. 5(g), since the best optimum path to get to the reward could remain "hidden" among the forbidden regions. We discussed this point in section "Reconfigurability of the hardware".

In order to avoid misunderstanding, we have now provided better explanation in section "Reconfigurability of the hardware" and in the caption of Fig. 5. Furthermore, we are going to propose a more accurate study related to the reconfigurability and scalability of the hardware.

- I might misunderstand the author's idea, but since the system doesn't know 1) whether the agent receives a penalty or not in new environment (there might be no obstacles in the area), 2) how many and where the agent receives a penalty, the system need to prepare same size of RRAM memory anyway before the rover starts exploring in different space on Mar.

Thank you for arising this topic, which has given us the possibility of enhancing the quality of our work and to overcome the remaining open points. We have now provided a full discussion which covers both qualitative and quantitative analysis about the management of the hardware architecture with respect to the scalability.

Since we have a limited amount of space in the main manuscript due to editorial rules, part of this discussion has been also reported in the supplementary information, precisely in “Additional insights over the scalability topic”.

We are going to answer to this point step by step:

1. We will first provide an overview about the scalability issue and the adopted solution to improve the area efficiency.
2. We will then deal with the topic related to how the system can know whether the agent is going to receive a penalty or not in a new environment. In particular, we are going to propose three different approaches to overcome the issue highlighted by the reviewer. The optimum solution which completely overcomes the arisen limitation is the third one (that is a direct consequence of the first two observations).
3. We finally demonstrate that the area/power consumption of the bio-inspired approach is much more efficient with respect to the state-of-the-art when the environments scales to higher dimensions.

As the reviewer correctly points out, the inhibition of certain region of space for computational effort saving when the environment scales to higher cardinality, rises once the agent gets a penalty. In this case, the “new type” of penalty simply inhibits a wider region of space, Fig. 5(f), and not only those points in which the agent touches a “forbidden boundary”, Fig. 4(d). Note that the shape of the inhibited region can vary depending on the type of abstraction is wanted to be performed, Fig. 5(e), and on the experience of the agent, Fig. 5(f).

The part of the memory in which we store the shapes coming from previous experience trials is not needed to be completely hardwired since a numerical function is enough to describe a big region of space (it is possible to provide simple memory addresses expressing a region that is not convenient to explore).

This means that it is possible to create a sort of “virtual memory” also used to reference parts of space that do not physically exist on hardware, basically giving rise to area saving. Furthermore, such information can be easily stored in separated registers, since it is not always needed for the computation, furtherly improving the scalability and the management of the resources.

In the same way, as the reviewer correctly points out, if there are no obstacles, the agent cannot get any penalty and thus no inhibited regions would be allocated, giving rise to a situation in which the scalability issue is not solved at all. However, as we are going to discuss, this can be easily overcome, provided that a good management of the resources is taken into consideration.

Solutions for improving the scalability properties of the bio-inspired hardware

As we report now in Section “Reconfigurability of the hardware”, in the caption of Fig. 5, and in supplementary “Additional insights over the scalability topic”, there are three solutions for the management of the computing architecture when the computing resources are limited.

1. *Rough solution* – It is possible to reduce the computational effort by providing lower sampling of the positions covered by the agent, meaning that a single step of the agent covers more physical space of the environment. This behaviour of the agent can be simply triggered by a CMOS-based counter which counts, during a first limited trial, the number of firing activities from penalty-to-penalty event, thus giving a rough estimation about the granularity of the sampling that can be applied to the research algorithm.

However, this solution can be critical in terms of safety and reliability, since the system takes advantage of a rough estimation, and it is not fully reactive to the actual configuration of the environment.

2. *Medium solution* - The second way is related to the “paging” that is done in current processors. The main goal of this procedure is to bring and compress part of the information stored in the RAM to the Hard Disk Drive, thus improving the scalability of the computing resources. Such choice can be also implemented in RRAM-based systems, but it is not an optimum solution. Basically, the paging enables the use of the processor unit for the current computation while keeping the previous details stored in other CMOS-based memory units. This method has a major drawback related to the Von-Neumann bottleneck since data must be fetched not only between the computing unit and the DRAM memory, but also towards the main memory (hard disk), thus arising significant power consumption and time latency. Furthermore, the standard non-volatile memory elements are not built in the back-end-of-the-line, raising a problem in terms of area efficiency with respect to the bio-inspired hardware proposed in this work.

Thus, even taking into consideration an abstraction of the stored information, the Von Neumann bottleneck limits the scalability of the resources. The problems related to the Von Neumann bottleneck are the ones that are currently hindering scalability in standard processors used for conventional deep learning algorithms [23].

3. *Best solution* - The third approach is based on the second one, but it proposes a further advancement which makes the bio-inspired approach completely scalable and far more efficient than the state-of-the-art even in the case in which penalties do not occur. Basically, it is possible to take advantage of the RRAM-based matrices for relying on the in-memory computation and avoiding the Von-Neumann bottleneck.

In particular, the memory array can be continually exploited by the bio-inspired computation until the complete memory resources are fully allocated. Then, when the memory array is completed, the direction of movement, the number of steps and some further information (such as the rewards) can be saved in separated registers (which can be also RRAM-

based) as pure coordinates (strongly limiting the data fetch). Thus, the RRAM computing units are practically ready again to perform the explorative trials, to abstract the previous maps of exploration and to reference the stored coordinates to the effective number of “refreshes” of the RRAM memory arrays.

The same approach could be exploited both when the agent undergoes penalties and when it does not. In every case, the relevant points can be safely compressed by means of coordinates and mathematical representations of large areas, and the bio-inspired exploration can go on assuring the highest possible accuracy following the approach described in Fig. 5(e,f) of the manuscript.

Thus, this third solution overcomes all the discussed points, and it also offers a new strategy to get hardware scalability.

We have now inserted the solution of this topic in the main manuscript, Section “Reconfigurability of the hardware” and in Fig. 5 (where we contextualize the algorithms to get scalability in bio-inspired systems based on non-volatile memories). All the other discussions have been reported in the supplementary information, Section “Additional insights over the scalability topic”.

To conclude, we propose here a last consideration related to the state-of-the-art: the scalability is much more complicated to assure (in terms of power and area resources) in standard hardware with respect to the RRAM-based solution presented in this work.

Comparison with the state-of-the-art in terms of scalability properties

In sections “Exploration, Optimization and Recall”, “Reconfigurability of the hardware” and “Discussion” we report now a comparison of our hardware with respect to a Python-based library for reinforcement learning (you can access the homepage of the library and all the main codes clicking here - <https://pypi.org/project/pyqlearning/> -). The benchmark is the same for both the approaches and it deals with the same environmental exploration, Fig. 4(f,g).

Note that all the test cases proposed in this work are fundamental not only for discussing theoretical issues on reinforcement learning but also for practical situations. For instance, the point-by-point mapping of the environment is a key topic for robots that must be used in archaeological areas to inspect hard-to-access sections of the ruins and to alert people for safety and structural problems whenever some unexpected changes are detected [62].

As we report in Fig. 4(g), the area/power resources needed for carrying out a correct neural computation are much more optimized for the bio-inspired approach: if the environment to explore gets bigger, the area and power resources for the bio-inspired solution, keeping constant the target accuracy, are 10x more efficient than what is obtained by exploiting standard reinforcement learning approaches [57, 58].

For simplicity, we report here below the plot under examination (Fig. R3).

Figure R3. Comparison in terms of memory computing elements between the deep Q-learning procedure and the bio-inspired solution at increasing sizes of the environment to explore. Note that the power consumption is also furtherly improved in the bio-inspired solution thanks to the use of RRAM memory devices built in the back end of the line, which avoids the Von Neuman bottleneck typical of standard computing platforms.

- One address can express some size of grids, but it would be challenging to mix different grid resolution in the area, such as normal(original) resolution for rover exploring and lower resolution for penalty area.

Thank you for arising this observation: in the previous version of the manuscript this point was not reported clearly enough. We are now going to provide a more accurate framework about the topic.

As we have just discussed in the previous answer, in parallel to the abstraction of the penalty shapes for “transfer learning” reported in Fig. 5(f), we also propose a complete scalable solution for the bio-inspired system. This solution enables complete re-programmability of the RRAM hardware supported by local savings of specific information which assure no accuracy loss.

In particular, combining these approaches, as now reported in section “Reconfigurability of the hardware”, there is no need of mixing different resolutions inside the RRAM array (the local information is stored by proper addresses, coordinates and mathematical representations in separated registers).

In the following of this answer, we are going (i) to recap the proposed solution which avoids the mix of different grid resolution areas and (ii) to introduce further benefits of the bio-inspired approach using RRAM devices.

The memory array can be continually exploited by the bio-inspired computation until the complete memory resources are fully allocated. Then, when the memory array is completed, the direction of movement, the number of steps and some further information (such as the rewards) can be saved in separated registers (which can be also RRAM-based) as pure coordinates (strongly limiting the data fetch). Thus, the RRAM computing units are practically ready again to perform the explorative trials, to abstract the previous maps of exploration and to reference the stored coordinates to the effective number of “refreshes” of the RRAM memory arrays. The same could be done when the agent undergoes penalties (and applying the same technique to get the time improvement, as

depicted in Fig. 5(g)). Hence, the penalty points can be safely compressed by means of coordinates and mathematical representations of large areas and the bio-inspired exploration can go on assuring the highest possible accuracy.

Thus, combining these approaches is possible to provide scalability and increased accuracy without needing different resolutions in the RRAM memory arrays, since all the information is stored in additional memory elements.

Furthermore, the aforementioned procedure highlights all the main benefits of RRAM-based bio-inspired computing, namely:

1. The RRAM-based approach shows significant efficiency in terms of area and power resources overcoming the state-of-the-art, as reported in Fig. 4(g).
2. The bio-inspired system also provides higher accuracy with respect to standard solutions, Fig. 4(f).
3. The reconfigurability of the RRAM-based hardware enables the improvement of the one-to-one matching between environmental positions and memory devices keeping high the accuracy and providing a reliable way for abstracting the size of the memory.
4. The RRAM-based solution not only overcomes the Von Neumann bottleneck, but it takes also into consideration the most relevant benefits of CMOS-based hardware management. Thus, merging the two approaches, our bio-inspired system goes towards the definition on a large scale of an efficient, cheap, and scalable computing architecture.

We would like to thank again the reviewer for all the interesting topics arisen about the scalability of the hardware: this discussion has been very useful to improve the quality of our work.

Minor comments:

Q(s, a) and Q(a, s) are mixed in equation-II and III. Is it intentionally swapped?

Thank you for arising the typo, we have now corrected the manuscript.

- It would be helpful for reviewers if there are line numbers in revised manuscript.

We thank the reviewer for this suggestion. We have now provided the revised manuscript and the supplementary information (highlighted versions) with line numbers.

We would like to thank again the reviewer for all the useful observations provided during this review process: every discussion, suggestion and all the advice have been very useful for improving the clarity of the manuscript and for enhancing the quality of our work.

REVIEWERS' COMMENTS

Reviewer #1 (Remarks to the Author):

The author provided a significant analysis for power/energy consumption. Figure. R3 shows a great comparison between the software and hardware. I have just one minor comment.

1. The RL matrix (M) can be trained via CNN and MLP. Is it then possible to implement such CNN- or MLP-based RL algorithms by using only RRAM crossbars for matrix multiplication?

Reviewer #3 (Remarks to the Author):

The manuscript has been made substantial improvements. Most of my comments have been addressed with additional analysis and discussions, revised figures, and supplemental document.

**Reply to the reviewers' comments for the paper entitled "A SELF-ADAPTIVE
HARDWARE WITH RESISTIVE SWITCHING SYNAPSES FOR EXPERIENCE-BASED
NEUROCOMPUTING" submitted to Nature Communications.**

Dear Reviewers,

We would like to thank you for the great contribution you gave during this review process with respect to our manuscript entitled "A self-adaptive hardware with resistive switching synapses for experience-based neurocomputing". All the suggestions and discussions have been very helpful for improving the overall quality of this work, finally making it worth of publication in "Nature Communications". In the following of this document, we report the point-by-point responses to the last reviewers' comments: the improvements and minor corrections are directly presented in the final version of the manuscript and in the supplementary information.

REVIEWERS' COMMENTS

Reviewer #1 (Remarks to the Author):

The author provided a significant analysis for power/energy consumption. Figure. R3 shows a great comparison between the software and hardware. I have just one minor comment.

We would like to thank the reviewer for his/her contribution in making this work worth of publication in Nature Communications. All the suggestions, comments and insights provided during the review process have been very helpful for enhancing the overall quality of our manuscript.

In the following, we finally provide a final answer to the last minor comment.

1. The RL matrix (M) can be trained via CNN and MLP. Is it then possible to implement such CNN- or MLP-based RL algorithms by using only RRAM crossbars for matrix multiplication?

We thank the reviewer for arising this point.

As correctly suggested in this comment, a possible improvement of the standard approach would require the exploitation of the RRAM-based matrix-vector multiplication for computing the M matrix. In this case, the multiply and accumulate operations of standard processors can be significantly reduced. However, as stated in Fig. 4(f) and in Fig. 4(g), the bio-inspired approach intrinsically provides more computing performance and less power consumption. Furthermore, expensive (for time and power) re-trainings of the convolutional neural network would be anyway necessary for enabling more plastic behaviours in standard software-based Q-learning.

We have discussed this point in Section “Comparison of the resilient properties between bio-inspired and deep learning approaches” of the supplementary information.

We would like to thank again the reviewer for all the useful observations provided during this review process: every discussion, suggestion and all the advice have been very useful for improving the clarity of the manuscript and for enhancing the quality of our work.

REVIEWER COMMENTS

Reviewer #3 (Remarks to the Author):

The manuscript has been made substantial improvements. Most of my comments have been addressed with additional analysis and discussions, revised figures, and supplemental document.

We would like to thank the reviewer for the observations provided during these review cycles which have led this work to be worth of publication in "Nature Communications": all the discussions, suggestions and insights have been fruitful for enhancing the readability of the manuscript and for improving the overall quality of this work.